# Provable Policy Gradient for Robust Average-Reward MDPs Beyond Rectangularity

**Qiuhao Wang** [* 1] **Yuqi Zha** [* 2] **Chin Pang Ho** [2] **Marek Petrik** [3]

## Abstract

Robust Markov Decision Processes (MDPs) offer a promising framework for computing reliable policies under model uncertainty. While policy gradient methods have gained increasing popularity in robust discounted MDPs, their application to the average-reward criterion remains largely unexplored. This paper proposes a Robust Projected Policy Gradient (RP2G), the first generic policy gradient method for robust average-reward MDPs (RAMDPs) that is applicable beyond the typical rectangularity assumption on transition ambiguity. In contrast to existing robust policy gradient algorithms, RP2G incorporates an adaptive decreasing tolerance mechanism for efficient policy updates at each iteration. We also present a comprehensive convergence analysis of RP2G for solving ergodic tabular RAMDPs. Furthermore, we establish the first study of the inner worst-case transition evaluation problem in RAMDPs, proposing two gradient-based algorithms tailored for rectangular and general ambiguity sets, each with provable convergence guarantees. Numerical experiments confirm the global convergence of our new algorithm and demonstrate its superior performance.

## 1. Introduction

Markov Decision Processes (MDPs) (Puterman, 2014) provide a powerful framework for sequential decision-making, with applications such as game solving (Mnih, 2013), healthcare (Shechter et al., 2008), and finance (Bäuerle & Rieder,

---

*Equal contribution [1]Fintech Innovation Center, Research Institute for Digital Economy and Interdisciplinary Sciences, Southwestern University of Finance and Economics [2]Department of Data Science, City University of Hong Kong [3]Department of Computer Science, University of New Hampshire. Correspondence to: Chin Pang Ho <clint.ho@cityu.edu.hk>, Marek Petrik <mpetrik@cs.unh.edu>.

*Proceedings of the 42$^{nd}$ International Conference on Machine Learning*, Vancouver, Canada. PMLR 267, 2025. Copyright 2025 by the author(s).

2011). However, applying MDPs to real-world problems often faces the challenge of model uncertainty, particularly in transition dynamics, which are rarely known precisely. To mitigate the impact of model errors, robust MDPs (Iyengar, 2005; Nilim & El Ghaoui, 2005) offer a compelling solution by assuming that uncertain parameters lie within a predefined *ambiguity set*. These robust MDPs aim to identify policies that optimize performance under the worst-case scenario within the ambiguity set.

The majority of existing research on robust MDPs focuses on the *discounted setting*, where future costs are discounted by a factor $\gamma \in (0, 1)$. Under this setting, robust MDPs become computationally tractable by imposing certain rectangularity assumptions, such as $(s, a)$-rectangularity (Iyengar, 2005; Nilim & El Ghaoui, 2005), $s$-rectangularity (Le Tallec, 2007; Wiesemann et al., 2013), $k$-rectangular (Mannor et al., 2016), and $r$-rectangular (Goyal & Grand-Clement, 2023). Significant progress has been made in both value-based methods that rely on the Bellman equation (Iyengar, 2005; Nilim & El Ghaoui, 2005; Kaufman & Schaefer, 2013; Ho et al., 2021; Panaganti & Kalathil, 2021) and gradient-based methods that directly optimize the policy (Wang & Zou, 2022; Li et al., 2022; Wang et al., 2023a; Kumar et al., 2024a; Lin et al., 2024; Wang et al., 2024a). Despite these advances, research on robust discounted MDPs beyond structured rectangular ambiguity sets is still scarce, with only a few notable exceptions (Li et al., 2023), as solving robust discounted MDPs with general ambiguity sets is NP-hard (Wiesemann et al., 2013).

While much of the existing work focuses on robust discounted MDPs, many real-world systems that primarily focus on the steady-state behavior, such as queueing control and scheduling automatic guided vehicles (Kober et al., 2013), may still yield policies that perform poorly over the long term (Wang et al., 2024b). Meanwhile, as discounted factor $\gamma$ approaches one, solution methods tend to converge more slowly, increasing computational costs (Grand-Clement et al., 2023). We refer interested reader to Appendix E.5 for more detailed numerical illustration. As such, optimizing the long-term average cost is preferable to the total discounted cost for these applications.

To overcome the limitations of the discounted setting, recent

research has focused on robust MDPs with the average-reward criterion (Tewari & Bartlett, 2007; Lim et al., 2013; Grand-Clement et al., 2023; Wang et al., 2023d;c; 2024b; Sun et al., 2024). While policy gradient methods have been widely employed in standard reinforcement learning due to their empirical success and flexibility for problems in complex environments, and have been effectively extended to robust discounted MDPs, the development of robust policy gradient for average-reward settings with optimality guarantees remains largely unexplored in the literature. To this end, we aim to develop a computationally tractable algorithm for solving robust average-reward MDPs (RAMDPs) with theoretical convergence guarantees. The challenges and our major contributions are summarized as follows.

Our first contribution is *Robust Projected Policy Gradient* (RP2G), a novel generic policy gradient scheme for solving RAMDPs. While RP2G retains the policy gradient updates used in nominal average-reward MDPs (AMDPs), it incorporates an additional inner subroutine for evaluating worst-case transitions. To reduce the computational cost of optimally solving the inner subroutine, RP2G incorporates a decreasing tolerance sequence for the inner subroutine, ensuring convergence even in the general ambiguity setting.

Our second contribution establishes the global convergence of RP2G to the optimal policy, assuming an oracle for solving the inner subroutine. While this result aligns with findings for nominal AMDPs (Kumar et al., 2024b), the robust setting introduces unique challenges, including the non-differentiability and non-convexity of the robust return (Razaviyayn et al., 2020). We address these challenges by leveraging the Moreau envelope as a differentiable surrogate for our convergence analysis.

Our third contribution is the two proposed specialized gradient-based algorithms to solve the inner problem: one for rectangular ambiguity sets based on projected gradient ascent, and another for general ambiguity sets leveraging a novel projected Langevin dynamics update. Both algorithms are supported by convergence and optimality guarantees. To our knowledge, this is the first study addressing the inner worst-case evaluation problem for RAMDPs.

**Notation.** Boldface lowercase letters and uppercase letters are used to denote vectors and matrices, respectively. The symbol $e$ denotes a vector of all ones of the size appropriate to the context and the symbol $e$ denotes the Euler's number. The set $\mathbb{R}$ represents the set of real numbers, and the set $\mathbb{R}_+$ represents the set of non-negative real numbers. The probability simplex in $\mathbb{R}_+^S$ is denoted as $\Delta^S$. For vectors, we use $\|\cdot\|$ to denote the $l_2$-norm.

## 1.1. Related Work

**Average-reward MDPs.** Early research on average-reward MDPs focused on fundamental characterizations of the model and its properties (Bertsekas, 2012; Puterman, 2014). Many existing methods consider model-free approaches in tabular settings (Abounadi et al., 2001; Yang et al., 2016; Wan et al., 2021; Avrachenkov & Borkar, 2022; Wan & Sutton, 2022; Chae et al., 2024; Yang et al., 2024). Function approximation techniques have also been studied for AMDPs(Marbach & Tsitsiklis, 2001; Abbasi-Yadkori et al., 2019; Wei et al., 2021; Zhang et al., 2021; Chen et al., 2023; Wu et al., 2022; Zhang & Xie, 2023). Parametrization methods are another popular approach (Liao et al., 2022; Wang et al., 2022; 2023b; Bai et al., 2024), along with gradient-based methods (Murthy & Srikant, 2023; Grosof et al., 2024; Kumar et al., 2024b). Despite these advancements, addressing the robust setting introduces additional challenges, which we focus on in this work.

**Robust average-reward MDPs.** Research on robust average-cost MDPs is limited (Tewari & Bartlett, 2007; Lim et al., 2013; Grand-Clement et al., 2023; Wang et al., 2023d;c; 2024b; Sun et al., 2024), with no prior work on gradient-based algorithms. While a concurrent work by (Sun et al., 2024) extends mirror descent for RAMDPs, their approach is restricted to $(s, a)$-rectangular ambiguity sets and requires exact worst-case transition evaluations, leading to high computational costs. In contrast, RP2G ensures global convergence for general compact, convex ambiguity sets and reduces computational cost via a decreasing adaptive tolerance for the worst-case transition evaluation.

## 2. Preliminaries

### 2.1. Average-Reward Markov Decision Processes

A nominal infinite-horizon average-reward MDP is defined by the tuple $\mathcal{M} = \langle \mathcal{S}, \mathcal{A}, \boldsymbol{p}, \boldsymbol{c}, \boldsymbol{\rho} \rangle$, where $\mathcal{S} = \{1, 2, \ldots, S\}$ and $\mathcal{A} = \{1, 2, \ldots, A\}$ represent the finite sets of states and actions, respectively. The initial state is chosen randomly according to the distribution $\boldsymbol{\rho} \in \Delta^S$. The probability distribution of transiting from a current state $s$ to a next state $s'$ after taking an action $a$ is denoted as a vector $\boldsymbol{p}_{sa} := (p_{sas'})_{s' \in \mathcal{S}} \in \Delta^S$, which is part of the transition kernel $\boldsymbol{p} := (\boldsymbol{p}_{sa})_{s \in \mathcal{S}, a \in \mathcal{A}} \in (\Delta^S)^{S \times A}$. The instantaneous cost of this transition is denoted by $c_{sas'}$ (or equivalently, a reward $r_{sas'} = -c_{sas'}$). We assume $c_{sas'} \in [0, 1]$ for all $s, s' \in \mathcal{S}$ and $a \in \mathcal{A}$, as translating or scaling costs does not affect the set of optimal policies (Puterman, 2014).

We focus on stationary randomized policies due to their practical simplicity (Sutton & Barto, 2018; Zhang et al., 2022). A stationary randomized policy $\boldsymbol{\pi} := (\boldsymbol{\pi}_s)_{s \in \mathcal{S}}$, where $\boldsymbol{\pi}_s \in \Delta^A$, specifies the probabilities over actions $a \in \mathcal{A}$ for each state $s \in \mathcal{S}$. Under this policy, the action $a$

is selected with probability $\pi_{sa}$ whenever the AMDP is in state $s \in \mathcal{S}$. The set of all stationary randomized policies is denoted by $\Pi = (\Delta^A)^S$.

The long-term average cost $J_{\boldsymbol{\rho}}(\boldsymbol{\pi}, \boldsymbol{p})$ for a given policy $\boldsymbol{\pi}$ and transition kernel $\boldsymbol{p}$ is defined as

$$J_{\boldsymbol{\rho}}(\boldsymbol{\pi}, \boldsymbol{p}) := \lim_{T \to \infty} \frac{1}{T} \mathbb{E}_{\boldsymbol{\pi}, \boldsymbol{p}, s_0 \sim \rho} \left[ \sum_{t=0}^{T-1} c_{s_t a_t s_{t+1}} \right]. \quad (1)$$

Here, $\mathbb{E}_{\boldsymbol{\pi}, \boldsymbol{p}, s_0 \sim \rho}$ denotes the expectation with respect to a stochastic process where the action $a_t$ is selected according to the policy $\boldsymbol{\pi}_{s_t}$, the next state $s_{t+1}$ evolves according to the transition kernel $\boldsymbol{p}_{s_t a_t}$, and the initial state $s_0$ is drawn from the initial distribution $\boldsymbol{\rho} \in \Delta^S$. For time-homogeneous MDPs with a finite state space and bounded costs, the limit in (1) is guaranteed to exist (Puterman, 2014).

In this work, we restrict our attention to the ergodic setting, which is formally stated through the following assumption:

**Assumption 2.1.** The MDP $\mathcal{M}$ is ergodic, *i.e.*, for any policy $\boldsymbol{\pi}$ and kernel $\boldsymbol{p}$, the Markov chain $\{s_t\}_{t \geq 0}$ is irreducible and aperiodic.

The assumption of ergodicity is standard in average-reward MDPs (Gong & Wang, 2020; Wei et al., 2020; Pesquerel & Maillard, 2022; Bai et al., 2024; Cheng et al., 2024; Ganesh et al., 2024; Wu et al., 2024). Under ergodicity, the average cost objective is independent of the initial distribution $\boldsymbol{\rho}$ for any feasible $\boldsymbol{\pi}$ and $\boldsymbol{p}$ (see, for example, (Puterman, 2014, Section 8)). Hence, we can redefine the long-term average cost by overloading the notation $J$ as:

$$J_{\boldsymbol{\rho}}(\boldsymbol{\pi}, \boldsymbol{p}) = J(\boldsymbol{\pi}, \boldsymbol{p}) := \mathbb{E}_{s \sim \boldsymbol{d}^{\boldsymbol{\pi}, \boldsymbol{p}}, a \sim \boldsymbol{\pi}_s, s' \sim \boldsymbol{p}_{sa}} [c_{sas'}], \quad (2)$$

where $\boldsymbol{d}^{\boldsymbol{\pi}, \boldsymbol{p}} \in \Delta^S$ is the stationary state distribution induced by $\boldsymbol{\pi}$ and $\boldsymbol{p}$, formally defined as:

$$d_s^{\boldsymbol{\pi}, \boldsymbol{p}} := \lim_{T \to \infty} \frac{1}{T} \mathbb{E}_{\boldsymbol{\pi}, \boldsymbol{p}} \left[ \sum_{t=0}^{T-1} \mathbf{1} \{s_t = s\} \right]. \quad (3)$$

It is well-established that the stationary distribution is unique under ergodicity (Norris, 1998; Meyn & Tweedie, 2012; Gagniuc, 2017) and independent of $\boldsymbol{\rho}$ as well (Puterman, 2014). The goal of an AMDP is to find a policy $\boldsymbol{\pi}^\star$ minimizing the long-run average cost:

$$\boldsymbol{\pi}^\star = \arg\min_{\boldsymbol{\pi} \in \Pi} J(\boldsymbol{\pi}, \boldsymbol{p}).$$

The above stationary and Markovian policy $\boldsymbol{\pi}^\star$ is guarantee to be optimal, even when considering the broader class of all possible policies, including history-dependent and non-stationary ones (Puterman, 2014).

## 2.2. Differential Value Functions

In the average-reward setting, we introduce the following differential functions, analogous to the value and action-value functions in standard MDPs. These functions quantify the accumulated deviations from steady-state performance and serve as key elements in our subsequent analysis. Specifically, the *differential action-value function* is defined as a solution to the following Bellman equation:

$$q_{sa}^{\boldsymbol{\pi}, \boldsymbol{p}} = \sum_{s'} p_{sas'} \left( c_{sas'} - J(\boldsymbol{\pi}, \boldsymbol{p}) + \sum_{a'} \pi_{s'a'} q_{s'a'}^{\boldsymbol{\pi}, \boldsymbol{p}} \right),$$

and the *differential state-value function* (also referred to as the *bias function* in (Puterman, 2014)) is defined as:

$$v_s^{\boldsymbol{\pi}, \boldsymbol{p}} = \sum_a \pi_{sa} \sum_{s'} p_{sas'} \left( c_{sas'} - J(\boldsymbol{\pi}, \boldsymbol{p}) + v_{s'}^{\boldsymbol{\pi}, \boldsymbol{p}} \right),$$

where it is known that $v_s^{\boldsymbol{\pi}, \boldsymbol{p}} = \sum_{a \in \mathcal{A}} \pi_{sa} q_{sa}^{\boldsymbol{\pi}, \boldsymbol{p}}$ (Sutton & Barto, 2018). Note that $\boldsymbol{v}^{\boldsymbol{\pi}, \boldsymbol{p}}$ and $\boldsymbol{q}^{\boldsymbol{\pi}, \boldsymbol{p}}$ are unique only up to an additive constant, *i.e.*, the above equations are satisfied by $\boldsymbol{q}^{\boldsymbol{\pi}, \boldsymbol{p}} + c_1 \boldsymbol{e}$ and $\boldsymbol{v}^{\boldsymbol{\pi}, \boldsymbol{p}} + c_2 \boldsymbol{e}$ for any arbitrary constants $c_1$ and $c_2$. To uniquely determine these functions, we impose the additional constraint $\sum_s d_s^{\boldsymbol{\pi}, \boldsymbol{p}} v_s^{\boldsymbol{\pi}, \boldsymbol{p}} = 0$ throughout the paper (Puterman, 2014; Wei et al., 2020; Bai et al., 2024; Cheng et al., 2024). Under this constraint, the differential state-value function be uniquely written as,

$$v_s^{\boldsymbol{\pi}, \boldsymbol{p}} := \mathbb{E}_{\boldsymbol{\pi}, \boldsymbol{p}, s_0 = s} \left[ \sum_{t=0}^{\infty} \left( c_{s_t a_t s_{t+1}} - J(\boldsymbol{\pi}, \boldsymbol{p}) \right) \right],$$

and the differential action-value function is

$$q_{sa}^{\boldsymbol{\pi}, \boldsymbol{p}} := \mathbb{E}_{\boldsymbol{\pi}, \boldsymbol{p}, s_0 = s, a_0 = a} \left[ \sum_{t=0}^{\infty} \left( c_{s_t a_t s_{t+1}} - J(\boldsymbol{\pi}, \boldsymbol{p}) \right) \right].$$

## 2.3. Robust Average-Reward Markov Decision Processes

In most applications, the exact transition kernel is not known precisely and must be estimated from data. These estimation errors often lead to policies that perform poorly when deployed. To address this challenge and ensure reliable policies under model uncertainty, RAMDPs, specified by $\langle \mathcal{S}, \mathcal{A}, \mathcal{P}, \boldsymbol{c}, \boldsymbol{\rho} \rangle$, aim to optimize the worst-case performance over a set of plausible errors (Goyal & Grand-Clement, 2023; Wang et al., 2023c; 2024b),

$$\min_{\boldsymbol{\pi} \in \Pi} \max_{\boldsymbol{p} \in \mathcal{P}} J(\boldsymbol{\pi}, \boldsymbol{p}), \quad (4)$$

where $\mathcal{P}$ is referred to as the *ambiguity set*. By appropriately calibrating $\mathcal{P}$, the optimal policy derived from (4) can guarantee reliable performance in the face of model errors (Grand-Clement et al., 2023; Wang et al., 2024b).

The concept of rectangular ambiguity set has been widely adopted in the context of robust MDPs due to their favorable computational properties (Iyengar, 2005; Nilim & El Ghaoui, 2005; Wiesemann et al., 2013; Ho et al., 2021). Two broad classes of rectangular ambiguity sets are mainly considered in this paper:

**Definition 2.2** $((s, a)$- and $s$-Rectangular Ambiguity Sets)**.** An ambiguity set $\mathcal{P} \subseteq (\Delta^S)^{S \times A}$ of transition kernel is called

1. $(s, a)$-*rectangular* (Iyengar, 2005; Nilim & El Ghaoui, 2005) if it is a Cartesian product of sets $\mathcal{P}_{s,a} \subseteq \Delta^S$ for each state $s \in \mathcal{S}$, *i.e.*, $\mathcal{P} = \Pi_{(s,a) \in \mathcal{S} \times \mathcal{A}} \mathcal{P}_{s,a}$;

2. $s$-*rectangular* (Wiesemann et al., 2013) if it is a Cartesian product of sets $\mathcal{P}_s \subseteq (\Delta^S)^A$ for each state $s \in \mathcal{S}$ and action $a \in \mathcal{A}$, *i.e.*, $\mathcal{P} = \Pi_{s \in \mathcal{S}} \mathcal{P}_s$.

Otherwise, we refer to an ambiguity set $\mathcal{P}$ as a *general ambiguity set* in this paper if it is neither $(s, a)$-rectangular nor $s$-rectangular, allowing for various dependencies across states and actions, including $k$-rectangular and $r$-rectangular ambiguity sets. While general ambiguity sets tend to be less conservative, they introduce significant analytical challenges, even in the discounted setting (Nilim & El Ghaoui, 2005; Wiesemann et al., 2013). We refer interested readers to Appendix E.6 for a detailed numerical illustration.

Note that, in contrast to most prior work that assumes rectangularity in RAMDPs (Goyal & Grand-Clement, 2023; Wang et al., 2023c;d; 2024b; Sun et al., 2024), our analysis of the proposed robust policy gradient method does not rely on this assumption. Instead, we only require that $\mathcal{P}$ be compact and convex. However, rectangularity assumptions can be helpful when developing algorithms for the inner maximization problem.

## 3. Robust Policy Gradient for RAMDPs

In this section, we introduce a policy gradient approach for solving RAMDPs. The key contribution of this section is to demonstrate that our algorithm computes a globally optimal solution of problem (4) with guarantees despite the non-convexity of the objective $J(\boldsymbol{\pi}, \boldsymbol{p})$. This result builds upon recent advancements in policy gradient methods for both ordinary MDPs (Agarwal et al., 2021; Bhandari & Russo, 2024) and AMDPs (Kumar et al., 2024b).

The rest of the section is organized as follows. In Section 3.1, we describe the motivation and details of our new policy gradient scheme. Then, in Section 3.2, we provide a standard convergence analysis, showing that our algorithm is guaranteed to converge to the global solution. To the best of our knowledge, this is the first generic robust policy gradient algorithm for a general ambiguity set that comes with global convergence guarantees.

---

**Algorithm 1** Robust Projected Policy Gradient (RP2G)

---

**Input:** initial policy $\boldsymbol{\pi}_0$, iteration number $T$, step sizes $\{\alpha_t\}_{t \geq 0}$, tolerances $\{\delta_t\}_{t \geq 0}$ with $\delta_{t+1} \leq \tau \delta_t$ for some $\tau \in (0, 1)$
**for** $t = 0, 1, \ldots, T-1$ **do**
  // Worst-Case Transition Evaluation
  Compute $\boldsymbol{p}_t$ such that $J(\boldsymbol{\pi}_t, \boldsymbol{p}_t) \geq \max_{\boldsymbol{p} \in \mathcal{P}} J(\boldsymbol{\pi}_t, \boldsymbol{p}) - \delta_t$;
  // Policy Improvement
  Update $\boldsymbol{\pi}_{t+1} \leftarrow \text{Proj}_\Pi(\boldsymbol{\pi}_t - \alpha_t \nabla_{\boldsymbol{\pi}} J(\boldsymbol{\pi}_t, \boldsymbol{p}_t))$;
**end for**
**Output:** $\boldsymbol{\pi}_{t^\star} \in \{\boldsymbol{\pi}_0, \ldots, \boldsymbol{\pi}_{T-1}\}$ such that $J(\boldsymbol{\pi}_{t^\star}, \boldsymbol{p}_{t^\star}) = \min_{t' \in \{0, \ldots, T-1\}} J(\boldsymbol{\pi}_{t'}, \boldsymbol{p}_{t'})$

---

### 3.1. Robust Projected Policy Gradient (RP2G)

From an optimization perspective, the optimal policy $\boldsymbol{\pi}^\star$ for the RAMDP is the solution $(\boldsymbol{\pi}^\star, \boldsymbol{p}^\star)$ of the minimax problem (4), where $\boldsymbol{\pi}^\star$ minimizes the function $\max_{\boldsymbol{p} \in \mathcal{P}} J(\boldsymbol{\pi}, \boldsymbol{p})$, and $\boldsymbol{p}^\star$ represents the worst-case transition kernel that maximizes $J(\boldsymbol{\pi}^\star, \boldsymbol{p})$ (Jin et al., 2020; Luo et al., 2020; Razaviyayn et al., 2020; Zhang et al., 2020). Thus, solving the RAMDP can be equivalently formulated as

$$\min_{\boldsymbol{\pi} \in \Pi} \left\{ \Psi(\boldsymbol{\pi}) := \max_{\boldsymbol{p} \in \mathcal{P}} J(\boldsymbol{\pi}, \boldsymbol{p}) \right\}. \quad (5)$$

It may seem natural to attempt solving (5) by performing gradient descent on the function $\Psi$. However, this approach is not applicable since $\Psi$ is not differentiable due to the inherent "max" operation (Razaviyayn et al., 2020). Furthermore, as $\Psi$ is neither convex nor concave, its subgradient does not exist either (Nouiehed et al., 2019; Lin et al., 2020). To overcome these challenges, we propose a specialized robust policy gradient algorithm summarized in Algorithm 1, termed *Robust Projected Policy Gradien* (RP2G).

RP2G adopts the well-known gradient-descent-ascent (GDA) scheme, drawing inspiration from the two-timescale rule to form a nested-loop structure with a max-oracle. In this section, we assume the existence of an oracle capable of solving the inner maximization problem. Further details regarding the evaluation of the inner worst-case transition kernel will be provided in Section 4.

Specifically, RP2G iteratively searches for an optimal policy in (5) by taking steps along the policy gradient. At each iteration $t$, Algorithm 1 first performs an inner update to approximate the worst-case transition kernel $\boldsymbol{p}_t$ for some given precision $\delta_t$. Once $\boldsymbol{p}_t$ is obtained, RP2G performs the projected gradient descent on $\boldsymbol{\pi}$ with fixed $\boldsymbol{p}_t$:

$$\boldsymbol{\pi}_{t+1} = \text{Proj}_\Pi (\boldsymbol{\pi}_t - \alpha_t \nabla_{\boldsymbol{\pi}} J(\boldsymbol{\pi}_t, \boldsymbol{p}_t)),$$

where $\text{Proj}_\Pi$ is the projection operator onto $\Pi$ and $\alpha_t > 0$ is the step size.

When chosen appropriately, the sequence $\{\delta_t\}_{t \geq 0}$ effectively reduces the computational burden while maintaining global convergence. This adaptive tolerance sequence, inspired by previous work on robust discounted MDP algorithms (Ho et al., 2021; Wang et al., 2024a), accelerates policy updates during the initial stages. As a result, it leads to significantly improved performance, as demonstrated by our experimental results in Section 5.2.

It is worth emphasizing that RP2G relies only on first-order information $\nabla_{\boldsymbol{\pi}} J(\boldsymbol{\pi}, \boldsymbol{p})$ to solve (5). Since $\boldsymbol{p}_t$ is fixed, this gradient is identical to the one used in ordinary AMDPs (Sutton & Barto, 2018); that is,

$$\frac{\partial J(\boldsymbol{\pi}, \boldsymbol{p})}{\partial \pi_{sa}} = d_s^{\boldsymbol{\pi}, \boldsymbol{p}} \cdot q_{sa}^{\boldsymbol{\pi}, \boldsymbol{p}}. \tag{6}$$

As a result, the non-differentiability of $\Psi(\boldsymbol{\pi})$ does not hinder the implementation of RP2G.

### 3.2. Global Convergence Analysis

In this subsection, we provide a convergence analysis of RP2G. In particular, we first leverage the sensitive analysis technique from (Cheng et al., 2024) to establish the weak convexity of non-convex, non-differentiable objective function $\Psi$. We then derive a tailored gradient dominance property for $\Psi$ in Theorem 3.4, which quantifies the gap between the function value and its optimum. Finally, we present the global convergence result in Theorem 3.5.

The following lemma establishes analytical bounds on the differential sensitivity of differential value functions, which are essential for proving continuity and convexity properties.

**Lemma 3.1.** *(Policy Sensitivity Bounds for Average-Reward MDPs) For any policies $\boldsymbol{\pi}, \boldsymbol{\pi}' \in \Pi$, transition kernel $\boldsymbol{p} \in (\Delta^S)^{S \times A}$, and state $s \in \mathcal{S}$, the following bounds hold:*

$$
\begin{aligned}
|d_s^{\boldsymbol{\pi}, \boldsymbol{p}} - d_s^{\boldsymbol{\pi}', \boldsymbol{p}}| &\leq C_d^{\boldsymbol{\pi}} \|\boldsymbol{\pi} - \boldsymbol{\pi}'\|_{1, \infty}, \\
|J(\boldsymbol{\pi}, \boldsymbol{p}) - J(\boldsymbol{\pi}', \boldsymbol{p})| &\leq C_J^{\boldsymbol{\pi}} \|\boldsymbol{\pi} - \boldsymbol{\pi}'\|_{1, \infty}, \\
\|\boldsymbol{v}^{\boldsymbol{\pi}, \boldsymbol{p}} - \boldsymbol{v}^{\boldsymbol{\pi}', \boldsymbol{p}}\|_\infty &\leq C_v^{\boldsymbol{\pi}} \|\boldsymbol{\pi} - \boldsymbol{\pi}'\|_{1, \infty}, \\
\|\boldsymbol{q}_s^{\boldsymbol{\pi}, \boldsymbol{p}} - \boldsymbol{q}_s^{\boldsymbol{\pi}', \boldsymbol{p}}\|_\infty &\leq C_q^{\boldsymbol{\pi}} \|\boldsymbol{\pi} - \boldsymbol{\pi}'\|_{1, \infty}.
\end{aligned}
$$

Due to page limit, the proof of this lemma, along with all remaining results, is provided in the appendix. Appendix A also includes a table that define all parameters. Using these sensitivity bounds, the weak convexity of the objective function $\Psi(\boldsymbol{\pi})$ can be established.

**Lemma 3.2.** *The objective function $J(\boldsymbol{\pi}, \boldsymbol{p})$ in (2) is $L_{\boldsymbol{\pi}}$-Lipschitz and $\ell_{\boldsymbol{\pi}}$-smooth in $\boldsymbol{\pi}$, implying that the robust objective $\Psi(\boldsymbol{\pi})$ is $\ell_{\boldsymbol{\pi}}$-weakly convex and $L_{\boldsymbol{\pi}}$-Lipschitz.*

*Remark* 3.3. Similar continuity results for AMDPs with respect to $\boldsymbol{\pi}$ were recently established in (Kumar et al., 2024b). Our analysis improves upon these results by providing

tighter Lipschitz constants $L_{\boldsymbol{\pi}} = \mathcal{O}(\sqrt{A})$ and $\ell_{\boldsymbol{\pi}} = \mathcal{O}(S)$, compared to $L_{\boldsymbol{\pi}} = \mathcal{O}(\sqrt{A}S^2)$ and $\ell_{\boldsymbol{\pi}} = \mathcal{O}(AS^3)$ in the prior work. This significantly reduces the dependence on the sizes of state and action spaces.

Lemma 3.2 establishes the continuity properties of $\Psi(\boldsymbol{\pi})$, which provides a crucial foundation for proving the global convergence of RP2G. However, weak convexity alone can not provide guarantees for convergence to a global optimum. Following classic results from stochastic approximation and optimization (Beck, 2017; Ostrovskii et al., 2021), Algorithm 1 is expected to converge to stationary points only.

Recent work (Agarwal et al., 2021; Bhandari & Russo, 2024) shows that policy gradient methods achieve global convergence in discounted MDPs under the *gradient dominance condition*, which ensures the gradient does not vanish prematurely. Informally, a function $h(\boldsymbol{x})$ satisfies this condition if $h(\boldsymbol{x}) - h(\boldsymbol{x}^\star) = \mathcal{O}(G(\boldsymbol{x}))$ where $G(\cdot)$ is a measure of the gradient of $h$ and $\boldsymbol{x}^\star$ is the global optimum of $h$.

Although $\Psi$ is non-smooth, weakly convex problems naturally admit an inherent smooth approximation through the Moreau envelope (Davis & Drusvyatskiy, 2019; Mai & Johansson, 2020). Extending the concept of gradient dominance, we introduce the gradient of the Moreau envelope and establish a tailored gradient dominance condition satisfied by $\Psi$, as presented in the following theorem.

**Theorem 3.4.** *Let $\boldsymbol{\pi}^\star$ be the globally optimal policy for RAMDPs. For any policy $\boldsymbol{\pi}$, the following holds:*

$$\Psi(\boldsymbol{\pi}) - \Psi(\boldsymbol{\pi}^\star) \leq \left(M\sqrt{SA} + \frac{L_{\boldsymbol{\pi}}}{2\ell_{\boldsymbol{\pi}}}\right) \cdot \|\nabla \Psi_{1/2\ell_{\boldsymbol{\pi}}}(\boldsymbol{\pi})\|,$$

*where $\Psi_\lambda(\boldsymbol{\pi})$ is the Moreau envelope of $\Psi(\boldsymbol{\pi})$.*

To derive this result, we introduce the *distribution mismatch coefficient* between two stationary distributions $\|d^{\boldsymbol{\pi}, \boldsymbol{p}}/d^{\boldsymbol{\pi}', \boldsymbol{p}'}\|_\infty$, which is often assumed to be bounded in prior works on average reward problems (Wang et al., 2023c; Kumar et al., 2024b; Sun et al., 2024), denoting as $M := \sup_{\boldsymbol{\pi}, \boldsymbol{\pi}', \boldsymbol{p}, \boldsymbol{p}'} \|d^{\boldsymbol{\pi}, \boldsymbol{p}}/d^{\boldsymbol{\pi}', \boldsymbol{p}'}\|_\infty < \infty$.

Theorem 3.4 shows that any first-order stationary point of the Moreau envelope corresponds to an approximately globally optimal policy. Building on this foundation, we now present a theorem that guarantees global convergence.

**Theorem 3.5.** *Let $\boldsymbol{\pi}_{t^\star}$ be the policy produced by Algorithm 1. With a constant step size $\alpha := 1/\sqrt{T}$ and an initial tolerance $\delta_0 \leq \sqrt{T}$, we have*

$$\Psi(\boldsymbol{\pi}_{t^\star}) - \min_{\boldsymbol{\pi} \in \Pi} \Psi(\boldsymbol{\pi}) \leq \epsilon,$$

*where $T$ is chosen such that*

$$T \geq \frac{\left(M\sqrt{SA} + \frac{L_{\boldsymbol{\pi}}}{2\ell_{\boldsymbol{\pi}}}\right)^4 \left(4\ell_{\boldsymbol{\pi}}S + 2\ell_{\boldsymbol{\pi}}L_{\boldsymbol{\pi}}^2 + \frac{4\ell_{\boldsymbol{\pi}}}{1-\tau}\right)^2}{\epsilon^4} = \mathcal{O}(\epsilon^{-4}).$$

At a high level, our proof of Theorem 3.5 first invokes a standard analysis of nonconvex stochastic subgradient descent (Davis & Drusvyatskiy, 2019) to analyze the number of iterations that is needed for computing a solution with sufficiently small Moreau envelope gradient. Building on this, the gradient dominance property established in Theorem 3.4 allows us to complete the proof. Note that the guarantee we provide is for the $\epsilon$-global optimum found within $\mathcal{O}(\epsilon^{-4})$ iterations, consistent with other GDA convergence results that apply the two-timescale rule in non-convex minimax optimization (Daskalakis et al., 2020; Jin et al., 2020).

The global convergence of RP2G hinges on an inner loop that identifies one worst-case transition kernel for a given policy $\boldsymbol{\pi}$. However, this computation is not trivial, as methods for evaluating the worst-case transition remain of RAMDPs largely unexplored. To address this challenge, we propose several tailored gradient-based algorithms for the inner maximization under different ambiguity assumptions.

## 4. Worst-Case Transition Evaluation

As yet, we have outlined RP2G and established its global convergence, assuming the worst-case transition kernel is computable. In this section, we focus on solving the inner maximization problem,

$$\Psi(\boldsymbol{\pi}) = \max_{\boldsymbol{p} \in \mathcal{P}} J(\boldsymbol{\pi}, \boldsymbol{p}), \tag{7}$$

referred to as the *worst-case transition evaluation problem*, by developing two gradient-based solution methods. Notably, the convergence results in Section 3 are independent of the inner evaluation method. We begin by deriving key properties of the inner evaluation problem in Section 4.1. Subsequently, Section 4.2 and Section 4.3 introduce and analyze tailored gradient-based algorithms designed for rectangular and general ambiguity sets, respectively.

### 4.1. General Properties

In general, the worst-case transition evaluation can be interpreted as an adversarial nature maximizing decision maker's average cost by selecting a proper transition kernel from the ambiguity set $\mathcal{P}$ (Lim et al., 2013; Goyal & Grand-Clement, 2023). To apply the gradient-based update on the transition kernel, we introduce the following lemma to derive the gradient of the evaluation problem.

**Lemma 4.1.** *(Adversary's Policy Gradient) For any policy $\boldsymbol{\pi} \in \Pi$ and transition kernel $\boldsymbol{p} \in (\Delta^S)^{S \times A}$, the gradient of $J(\boldsymbol{\pi}, \boldsymbol{p})$ over $\boldsymbol{p}$ has the analytical form as follows:*

$$\frac{\partial J(\boldsymbol{\pi}, \boldsymbol{p})}{\partial p_{sas'}} = d_s^{\boldsymbol{\pi},\boldsymbol{p}} \cdot \pi_{sa} \cdot \left(c_{sas'} - J(\boldsymbol{\pi}, \boldsymbol{p}) + v_{s'}^{\boldsymbol{\pi},\boldsymbol{p}}\right),$$

*where $g_{sas'}^{\boldsymbol{\pi},\boldsymbol{p}} := c_{sas'} - J(\boldsymbol{\pi}, \boldsymbol{p}) + v_{s'}^{\boldsymbol{\pi},\boldsymbol{p}}$ is referred to as the differential action-next-state value function (Li et al., 2023; Wang et al., 2024b).*

---

**Algorithm 2** Projected gradient ascent for solving the worst-case transition kernel

**Input:** current policy $\boldsymbol{\pi}$, initial kernel $\boldsymbol{p}_0$, iteration number $K$, step size sequences $\{\beta_k\}_{k \geq 0}$
**for** $k = 0, 1, \ldots, K-1$ **do**
    Update $\boldsymbol{p}_{k+1} \leftarrow \text{Proj}_{\mathcal{P}}(\boldsymbol{p}_k + \beta_k \nabla_{\boldsymbol{p}} J(\boldsymbol{\pi}, \boldsymbol{p}_k))$;
**end for**
**Output:** $\boldsymbol{p}_{k^\star} \in \{\boldsymbol{p}_0, \ldots, \boldsymbol{p}_{K-1}\}$ such that $J(\boldsymbol{\pi}, \boldsymbol{p}_{k^\star}) = \max_{k' \in \{0, \ldots, K-1\}} J(\boldsymbol{\pi}, \boldsymbol{p}_{k'})$

---

Note that with the policy $\boldsymbol{\pi}$ is being fixed, the transition kernel evaluation could be regarded as a constrained nonconcave maximization problem. From standard optimization analysis, a smooth function ensures that small gradient ascent updates improve the objective value (see Appendix B.2). To establish the required smoothness conditions, we first derive relevant sensitivity bounds for the transition kernel, as stated in the following lemma.

**Lemma 4.2.** *(Adversary Sensitivity Bounds for Average-Reward MDPs) For any transition kernels $\boldsymbol{p}_1, \boldsymbol{p}_2 \in (\Delta^S)^{S \times A}$, policy $\boldsymbol{\pi} \in \Pi$, and state-action pair $(s, a) \in \mathcal{S} \times \mathcal{A}$, the following sensitivity bounds are established:*

$$|d_s^{\boldsymbol{\pi},\boldsymbol{p}_1} - d_s^{\boldsymbol{\pi},\boldsymbol{p}_2}| \leq C_d^{\boldsymbol{p}} \|\boldsymbol{p}_1 - \boldsymbol{p}_2\|_{1,\infty},$$
$$|J(\boldsymbol{\pi}, \boldsymbol{p}_1) - J(\boldsymbol{\pi}, \boldsymbol{p}_2)| \leq C_J^{\boldsymbol{p}} \|\boldsymbol{p}_1 - \boldsymbol{p}_2\|_{1,\infty},$$
$$\|\boldsymbol{v}^{\boldsymbol{\pi},\boldsymbol{p}_1} - \boldsymbol{v}^{\boldsymbol{\pi},\boldsymbol{p}_2}\|_{\infty} \leq C_v^{\boldsymbol{p}} \|\boldsymbol{p}_1 - \boldsymbol{p}_2\|_{1,\infty},$$
$$\|\boldsymbol{g}_{sa}^{\boldsymbol{\pi},\boldsymbol{p}_1} - \boldsymbol{g}_{sa}^{\boldsymbol{\pi},\boldsymbol{p}_2}\|_{\infty} \leq C_g^{\boldsymbol{p}} \|\boldsymbol{p}_1 - \boldsymbol{p}_2\|_{1,\infty}.$$

Using the result of sensitivity bounds, we can obtain the continuity of the transition evaluation problem, showing the Lipschitz continuity and smoothness.

**Lemma 4.3.** *The objective function $J(\boldsymbol{\pi}, \boldsymbol{p})$ in (2) is $L_{\boldsymbol{p}}$-Lipschitz continuous and $\ell_{\boldsymbol{p}}$-smooth with respect to $\boldsymbol{p}$.*

### 4.2. Rectangular Ambiguity Sets

Under the common rectangularity assumption on the ambiguity set (Iyengar, 2005; Nilim & El Ghaoui, 2005; Wiesemann et al., 2013), Algorithm 2 is proposed as a first gradient-based method to solve the worst-case transition evaluation problem with a guarantee of global convergence. To maximize $J(\boldsymbol{\pi}, \boldsymbol{p})$ over $\boldsymbol{p}$, Algorithm 2 iteratively performs the *projected gradient update* on $\boldsymbol{p}$:

$$\boldsymbol{p}_{k+1} = \text{Proj}_{\mathcal{P}}(\boldsymbol{p}_k + \beta_k \nabla_{\boldsymbol{p}} J(\boldsymbol{\pi}, \boldsymbol{p}_k)),$$

which depends on the explicit form of $\mathcal{P}$. Given the specific structure of rectangularity, this projected gradient update can be further decoupled to multiple projection updates across $(s, a)$- or $s$-tuple: for $(s, a)$-rectangular RAMDPs, we have for any $s \in \mathcal{S}$,

$$\boldsymbol{p}_{k+1,sa} = \text{Proj}_{\mathcal{P}_{s,a}}(\boldsymbol{p}_{k,sa} + \beta_k \nabla_{\boldsymbol{p}_{sa}} J(\boldsymbol{\pi}, \boldsymbol{p}_k)),$$

whereas for $s$-rectangular RAMDPs,

$$\boldsymbol{p}_{k+1,s} = \text{Proj}_{\mathcal{P}_s}(\boldsymbol{p}_{k,s} + \beta_k \nabla_{\boldsymbol{p}_s} J(\boldsymbol{\pi}, \boldsymbol{p}_k)).$$

Since $s$-rectangularity is more general compared to $(s, a)$-rectangularity, our analysis is primarily based on the $s$-rectangular ambiguity set. However, our results readily extend to the $(s, a)$-rectangular case.

Due to the non-convex nature of $J$, the smoothness property established in Lemma 4.3 alone is insufficient to ensure global convergence. To address this, we derive the following specialized gradient dominance condition for the evaluation problem, which provides the foundation of our global convergence guarantee.

**Theorem 4.4.** *(Adversary's Gradient Dominance) When the ambiguity set $\mathcal{P}$ is $s$-rectangular, for any $\boldsymbol{\pi} \in \Pi$, we have,*

$$J(\boldsymbol{\pi}, \boldsymbol{p}^\star) - J(\boldsymbol{\pi}, \boldsymbol{p}) \leq M \cdot \max_{\bar{\boldsymbol{p}} \in \mathcal{P}} \langle \bar{\boldsymbol{p}} - \boldsymbol{p}, \nabla_{\boldsymbol{p}} J(\boldsymbol{\pi}, \boldsymbol{p}) \rangle,$$

*where $\boldsymbol{p}^\star$ be one of worst-case transition kernel over $\boldsymbol{\pi}$, i.e., $\boldsymbol{p}^\star \in \arg\max_{\boldsymbol{p} \in \mathcal{P}} J(\boldsymbol{\pi}, \boldsymbol{p})$.*

The above lemma ensures that any stationary point of $J(\boldsymbol{\pi}, \boldsymbol{p})$ is globally optimal. By leveraging the above result, we estabilish the convergence rate of Algorithm 2.

**Theorem 4.5.** *Let $\boldsymbol{p}_{k^\star}$ be the output of Algorithm 2 and $\delta_{\boldsymbol{\pi}} > 0$ be the precision. Then, for $s$-rectangular RAMDPs, Algorithm 2 with constant step size $\beta = 1/\ell_{\boldsymbol{p}}$ satisfies*

$$\max_{\boldsymbol{p} \in \mathcal{P}} J_{\boldsymbol{\rho}}(\boldsymbol{\pi}, \boldsymbol{p}) - J_{\boldsymbol{\rho}}(\boldsymbol{\pi}, \boldsymbol{p}_{k^\star}) \leq \delta_{\boldsymbol{\pi}},$$

*whenever*

$$K \geq \frac{32\ell_{\boldsymbol{p}} M^2 SA}{\delta_{\boldsymbol{\pi}}^2} = \mathcal{O}(\delta_{\boldsymbol{\pi}}^{-2}).$$

### 4.3. General Ambiguity Sets

While $(s, a)$- and $s$- rectangularity assumptions simplify the inner maximization problem due to the independence among state-action pairs (and states), many practical scenarios involve general ambiguity where such independence no longer holds (Wiesemann et al., 2013; Li et al., 2023), resulting in a more challenging optimization landscape.

To tackle this challenge, we draw inspirtion from (Lamperski, 2021; Li et al., 2023) and extend our discussion to propose a new tailored Markov Chain Monte Carlo algorithm designed for the general evaluation problem with probabilistic global optimality guarantees. Specifically, for the worst-case evaluation problem, we consider $J(\boldsymbol{\pi}, \boldsymbol{p}) \colon \mathcal{P} \to \mathbb{R}$ and the following relevant Gibbs distribution:

$$\nu_\lambda(\mathcal{B}) = \frac{\int_{\mathcal{B}} \exp(\lambda J(\boldsymbol{\pi}, \boldsymbol{p})) \mathrm{d}\boldsymbol{p}}{\int_{\mathcal{P}} \exp(\lambda J(\boldsymbol{\pi}, \bar{\boldsymbol{p}})) \mathrm{d}\bar{\boldsymbol{p}}},$$

---

**Algorithm 3** Projected Langevin dynamics for solving the worst-case transition kernel

**Input:** current policy $\boldsymbol{\pi}$, initial kernel $\boldsymbol{p}_0$, Gibbs parameter $\lambda > 1$, step size $\eta > 0$, iteration number $K$
**for** $k = 0, 1, \dots, K-1$ **do**

Sample $w_{k+1} \sim \mathcal{N}(0, \boldsymbol{I}_{(AS^2) \times (AS^2)})$;

Set $\hat{\boldsymbol{p}}_k = \boldsymbol{p}_k + \eta \nabla_{\boldsymbol{p}} J(\boldsymbol{\pi}, \boldsymbol{p})|_{\boldsymbol{p}=\boldsymbol{p}_k} + \sqrt{\frac{2\eta}{\lambda}} w_{k+1}$;

Update $\boldsymbol{p}_{k+1} = \arg\min_{\boldsymbol{p} \in \mathcal{P}} \|\boldsymbol{p} - \hat{\boldsymbol{p}}_k\|$;

**end for**

---

where $\lambda > 1$ is the temperature parameter. Sampling from $\nu_\lambda$ is of interest because it converges weakly to the uniform distribution over the global maxima of $J(\boldsymbol{\pi}, \boldsymbol{p})$ as $\lambda \to \infty$ (Hwang, 1980). Notably, the compactness of $\mathcal{P}$ and the continuity of $J(\boldsymbol{\pi}, \boldsymbol{p})$ ensure the denominator remains finite.

Building on the insights for the above Gibbs distribution $\nu_\lambda$, we employ the discrete-time Langevin diffusion to generate samples from the Gibbs distribution, as outlined in Algorithm 3. At each iteration $k$, Algorithm 3 iteratively applies the projected gradient ascent step perturbed by Gaussian noise to update the transition kernel:

$$\boldsymbol{p}_{k+1} = \text{Proj}_{\mathcal{P}}\left(\boldsymbol{p}_k + \eta \nabla_{\boldsymbol{p}} J(\boldsymbol{\pi}, \boldsymbol{p})|_{\boldsymbol{p}=\boldsymbol{p}_k} + \sqrt{\frac{2\eta}{\lambda}} w_{k+1}\right).$$

After $K$ iterations, the output $\boldsymbol{p}_K$ follows a distribution $\nu_K$ that approaches $\nu_\lambda$ within the 1-Wasserstein distance (Lamperski, 2021).

**Theorem 4.6.** *Assume $\eta < 1/2$, $\delta_{\boldsymbol{\pi}} > 0$ and $\kappa \in (0, 1)$. Then, there exist positive constants $a > 4$, $b > 1$, and $c_1, c_2, c_3 > 0$ such that $\lambda \geq c_1^{-1}(2AS^2/(c_1(1 - \kappa)\delta_{\boldsymbol{\pi}}e))^{1/\kappa}$ and $K \geq \max\{4, c_2 \exp\{c_3 A^b S^{2b}\}/\delta_{\boldsymbol{\pi}}^a\}$, the distribution $\nu_K$ of the output $\boldsymbol{p}_K$ of Algorithm 3 satisfies $\mathbb{E}_{\boldsymbol{p} \sim \nu_K}[J(\boldsymbol{\pi}, \boldsymbol{p})] \geq \max_{\boldsymbol{p} \in \mathcal{P}} J(\boldsymbol{\pi}, \boldsymbol{p}) - \delta_{\boldsymbol{\pi}}$.*

Theorem 4.6 establishes that the number of iterations required to achieve a $\delta_{\boldsymbol{\pi}}$-optimal inner solution grows exponentially with the dimension $AS^2$ of the transition kernel and with the number of desired accuracy digits, $\log(1/\delta_{\boldsymbol{\pi}})$. While the complexity may appear high, this algorithm is the first general approach capable of addressing the worst-case transition evaluation problem for general RAMDPs.

## 5. Numerical Experiments

We now demonstrate the convergence and robustness of RP2G, along with the two proposed inner solution methods, on the standard benchmark, GARNET MDPs (Archibald et al., 1995). All results were generated on an Apple M2 Max with 32 GB LPDDR5 memory. The algorithms are implemented in Python 3.11.5, and we use Gurobi 11.0.3

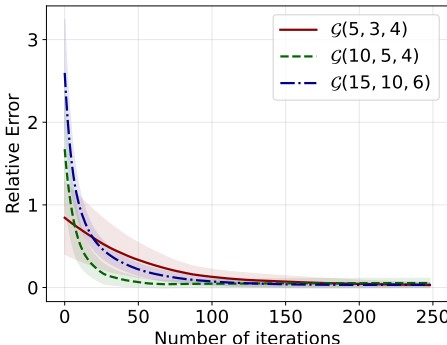

*Figure 1.* The relative difference of objective values computed by RP2G and RVI for Garnet problems with different sizes.

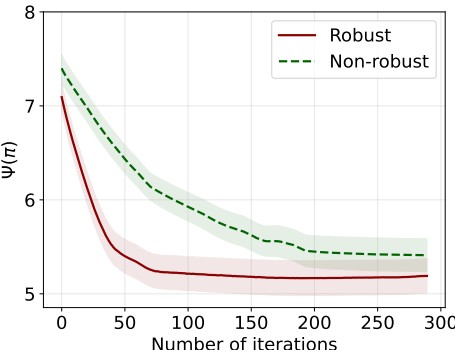

*Figure 2.* Performance comparison of RP2G and non-robust PG on Garnet problems with ellipsoid general ambiguity sets.

to solve any linear optimization problems involved. To support reproducibility, the full source code used to generate the results is available at https://github.com/Charliez7/robust-AMDP. Additional information on the benchmark and experimental setup is given in Appendix E.

### 5.1. Rectangular RAMDPs

We validate the convergence of RP2G on random GARNET MDPs across varying problem sizes with $(s, a)$-rectangular ambiguity. For each size, we generate 50 instances and compare the objective values of RP2G at different iterations with the optimal values $J^\star$ computed using the robust value iteration method from (Wang et al., 2023c). Figure 1 illustrates how the relative error (*i.e.*, $|J(\boldsymbol{\pi}_t, \boldsymbol{p}_t) - J^\star|/J^\star$) decreases consistently as the number of iterations increases, demonstrating the convergence and optimality of our algorithm. The upper and lower envelopes of the curves correspond to the 95 and 5 percentiles of the 50 samples, respectively.

### 5.2. Runtime Comparison

We now conduct experiment on GARNET MDPs with $(s, a)$-rectangular ambiguity to assess the impact of the decreasing tolerance sequence $\{\delta_t\}_{t\geq 0}$ on the computational efficiency. Specifically, we compare RP2G with the only existing gradient-based method, robust policy mirror descent (RPMD) (Sun et al., 2024), which assumes exact inner

*Table 1.* Average runtimes and standard deviations (in seconds) comparison of algorithms.

| Problem Size | RPMD | RP2G |
|---|---|---|
| $\mathcal{G}(5, 3, 3)$ | 13.99(13.05) | 0.63(0.35) |
| $\mathcal{G}(10, 5, 5)$ | 532.64(356.84) | 2.48(0.90) |
| $\mathcal{G}(15, 10, 6)$ | 2711.79(849.00) | 12.54(3.85) |

solutions. For this comparison, we set the tolerance of the worst-case transition evaluation problem in RPMD to a fixed value $\delta = 10^{-5}$, whereas RP2G uses a decreasing sequence initialized at $\delta_0 = 1$ with a decay rate of $\tau = 0.95$. For each problem size, we run 30 instances and report the average runtimes and standard deviations in Table 1, with termination based on minimal changes in the objective (*i.e.*, $\|J(\boldsymbol{\pi}_{t+1}, \boldsymbol{p}_{t+1}) - J(\boldsymbol{\pi}_t, \boldsymbol{p}_t)\| \leq 10^{-4}$). The results indicate that RP2G, leveraging the decreasing tolerance sequence, significantly outperforms RPMD in runtime efficiency.

### 5.3. General RAMDPs

In this experiment, we implement RP2G using the general inner solution method (Algorithm 3). We consider an ellipsoidal ambiguity set $\mathcal{P}$ (Li et al., 2023), which is not neither $(s, a)$-rectangular nor $s$-rectangular:

$$\mathcal{P} = \left\{ \boldsymbol{p} : (\boldsymbol{p} - \bar{\boldsymbol{p}})^\top \boldsymbol{\Sigma} (\boldsymbol{p} - \bar{\boldsymbol{p}}) \leq r \right\},$$

with size parameter $r > 0$, Hessian matrix $\boldsymbol{\Sigma}$, and nominal transition kernel $\bar{\boldsymbol{p}}$. To evaluate RP2G's robustness, we compare it against the non-robust policy gradient (PG) method, which optimizes under the nominal model. We apply both methods to 20 sample problems, recording $\Psi(\boldsymbol{\pi}_t) = \max_{\boldsymbol{p} \in \mathcal{P}} J(\boldsymbol{\pi}_t, \boldsymbol{p})$ for policies generated by RP2G and PG, respectively, at each iteration $t$. As shown in Figure 2, RP2G achieves robust performance and converges under general ambiguity. The shaded regions indicate the range between the 5 and 95 percentiles over the 20 samples.

## 6. Conclusion

In this paper, we proposed RP2G, a novel policy optimization algorithm for solving RAMDPs with general ambiguity sets. RP2G ensures global convergence under mild conditions by incorporating a suitable step size and an adaptive

tolerance sequence. Additionally, we conducted the first study on the inner worst-case transition evaluation problem, developing gradient-based solution methods in both rectangular and more general settings. Experiments validate the global convergence of RP2G, its efficiency, and robustness compared to non-robust approaches. Future work could explore extensions to scalable, model-free algorithms.

# Acknowledgements

We thank our friend Min Cheng for her helpful insights on deriving the sensitivity bounds. This work was supported, in part, by CityUHK Start-Up Grant (Project No. 9610481) and the Research Grants Council of Hong Kong (General Research Fund, Project No. 11508623). This work was also supported, in part, by NSF grants 2144601 and 2218063.

# Impact Statement

This paper focuses on theoretical aspects of Machine Learning. There are no new societal impacts that we can identify.

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

# A. Table of constants needed in analysis

We restate the table of constants and their description in this appendix for the sake of convenience.

| | Definition | Remark |
|---|---|---|
| $t_{\mathrm{mix}}$ | $\max_{\boldsymbol{\pi}\in\Pi,\boldsymbol{p}\in\mathcal{P}} t_{\mathrm{mix}}^{\boldsymbol{\pi},\boldsymbol{p}}$ | Uniform bound on the mix time (See Definition B.2) |
| $C$ | $\max_{\boldsymbol{\pi}\in\Pi,\boldsymbol{p}\in\mathcal{P}} \|(\boldsymbol{I}-\boldsymbol{P}^{\boldsymbol{\pi}}+\boldsymbol{P}^{\boldsymbol{\pi},\infty})^{-1}\|_{\infty}$ | See Definition A.1 |
| $C_d^{\boldsymbol{\pi}}$ | $7t_{\mathrm{mix}}$ | Sensitive bound on $\boldsymbol{d}^{\boldsymbol{\pi},\boldsymbol{p}}$ w.r.t. $\boldsymbol{\pi}$ |
| $C_J^{\boldsymbol{\pi}}$ | $7t_{\mathrm{mix}}$ | Sensitive bound on $J(\boldsymbol{\pi},\boldsymbol{p})$ w.r.t. $\boldsymbol{\pi}$ |
| $C_v^{\boldsymbol{\pi}}$ | $2C+C_d^{\boldsymbol{\pi}}SC+C^2S+C_d^{\boldsymbol{\pi}}C^2S$ | Sensitive bound on $\boldsymbol{v}^{\boldsymbol{\pi},\boldsymbol{p}}$ w.r.t. $\boldsymbol{\pi}$ |
| $C_q^{\boldsymbol{\pi}}$ | $C_J^{\boldsymbol{\pi}}+C_v^{\boldsymbol{\pi}}$ | Sensitive bound on $\boldsymbol{q}^{\boldsymbol{\pi},\boldsymbol{p}}$ w.r.t. $\boldsymbol{\pi}$ |
| $L_{\boldsymbol{\pi}}$ | $7t_{\mathrm{mix}}\sqrt{A}$ | Restricted Lipschitz constant w.r.t. $\boldsymbol{\pi}$ |
| $\ell_{\boldsymbol{\pi}}$ | $4C_q^{\boldsymbol{\pi}}+28t_{\mathrm{mix}}C_d^{\boldsymbol{\pi}}S$ | Restricted gradient Lipschitz constant w.r.t. $\boldsymbol{\pi}$ (Smoothness) |
| $C_d^{\boldsymbol{p}}$ | $2+5t_{\mathrm{mix}}$ | Sensitive bound on $\boldsymbol{d}^{\boldsymbol{\pi},\boldsymbol{p}}$ w.r.t. $\boldsymbol{p}$ |
| $C_J^{\boldsymbol{p}}$ | $2+5t_{\mathrm{mix}}$ | Sensitive bound on $J(\boldsymbol{\pi},\boldsymbol{p})$ w.r.t. $\boldsymbol{p}$ |
| $C_v^{\boldsymbol{p}}$ | $2C+CSC_d^{\boldsymbol{p}}+C^2+C^2SC_d^{\boldsymbol{p}}$ | Sensitive bound on $\boldsymbol{v}^{\boldsymbol{\pi},\boldsymbol{p}}$ w.r.t. $\boldsymbol{p}$ |
| $C_q^{\boldsymbol{p}}$ | $C_J^{\boldsymbol{p}}+C_v^{\boldsymbol{p}}$ | Sensitive bound on $\boldsymbol{q}^{\boldsymbol{\pi},\boldsymbol{p}}$ w.r.t. $\boldsymbol{p}$ |
| $L_{\boldsymbol{\pi}}$ | $(2+5t_{\mathrm{mix}})\sqrt{S}$ | Restricted Lipschitz constant w.r.t. $\boldsymbol{p}$ |
| $\ell_{\boldsymbol{\pi}}$ | $4C_g^{\boldsymbol{p}}+4(2+5t_{\mathrm{mix}})C_d^{\boldsymbol{p}}S$ | Restricted gradient Lipschitz constant w.r.t. $\boldsymbol{p}$ (Smoothness) |

Table 2. List of Constants

**Definition A.1.** [(Wang et al., 2023c; Cheng et al., 2024)] For any policy $\boldsymbol{\pi}\in\Pi$ and transition kernel $\boldsymbol{p}\in\mathcal{P}$, the matrix $(\boldsymbol{I}-\boldsymbol{P}^{\boldsymbol{\pi}}+\boldsymbol{P}^{\boldsymbol{\pi},\infty})$ is invertible (Puterman, 2014). We define

$$C := \max_{\boldsymbol{\pi}\in\Pi,\boldsymbol{p}\in\mathcal{P}} \|(\boldsymbol{I}-\boldsymbol{P}^{\boldsymbol{\pi}}+\boldsymbol{P}^{\boldsymbol{\pi},\infty})^{-1}\|_{\infty}.$$

# B. Auxiliary Lemmas

## B.1. Definitions and Properties of Ergodic Average-Reward Markov Decision Process

At the beginning, we consider the differential state-value function and provide some useful results. As we add the constraint $\sum_s d_s^{\boldsymbol{\pi},\boldsymbol{p}} v_s^{\boldsymbol{\pi},\boldsymbol{p}} = 0$, the differential value function takes the following form:

$$v_s^{\boldsymbol{\pi},\boldsymbol{p}} := \mathbb{E}_{\boldsymbol{\pi},\boldsymbol{p},s_0=s}\left[\sum_{t=0}^{\infty}\left(c_{s_t a_t s_{t+1}} - J(\boldsymbol{\pi},\boldsymbol{p})\right)\right]$$

$$= \sum_{t=0}^{\infty}\sum_{s'}\left(P_{ss'}^{\boldsymbol{\pi},(t)} - d_{s'}^{\boldsymbol{\pi},\boldsymbol{p}}\right)c_{s'}^{\boldsymbol{\pi},\boldsymbol{p}},$$

where $c_s^{\boldsymbol{\pi},\boldsymbol{p}} = \mathbb{E}_{a\sim\boldsymbol{\pi}_s,s'\sim\boldsymbol{p}_{sa}}[c_{sas'}]$ is defined as the state cost, and $\boldsymbol{P}^{\boldsymbol{\pi},(t)} \subseteq (\Delta^S)^S$ is denoted as the $t$-step transition matrix induced by $\boldsymbol{\pi}$ and $\boldsymbol{p}$. where for $t\geq 1$,

$$P_{ss''}^{\boldsymbol{\pi},(t)} = \sum_{s'} P_{ss'}^{\boldsymbol{\pi},(t-1)} P_{s's''}^{\boldsymbol{\pi}}, \ P_{ss'}^{\boldsymbol{\pi}} = \sum_a \pi_{sa} p_{sas'}.$$

Then, we can obtain the analytical form of the differential state-value function.

**Lemma B.1** (Analytical Form of Value Function (Puterman, 2014)). *Let* $\boldsymbol{P}^{\boldsymbol{\pi},\infty} = \lim_{T\to\infty} \frac{1}{T} \sum_{t=0}^{T-1} \boldsymbol{P}^{\boldsymbol{\pi},(t)}$ *be the limit matrix, where each row corresponds to the stationary distribution* $\boldsymbol{d}^{\boldsymbol{\pi},\boldsymbol{p}}$. *Then, we obtain a closed-form expression for the differential value function:*

$$\boldsymbol{v}^{\boldsymbol{\pi},\boldsymbol{p}} \;=\; (\boldsymbol{I} - \boldsymbol{P}^{\boldsymbol{\pi}} + \boldsymbol{P}^{\boldsymbol{\pi},\infty})^{-1}(\boldsymbol{I} - \boldsymbol{P}^{\boldsymbol{\pi},\infty})\boldsymbol{c}^{\boldsymbol{\pi},\boldsymbol{p}}.$$

Then, we introduce a crucial definition that benefits our further analysis. Under the assumption of ergodicity, a finite mixing time is guaranteed and is defined as follows (Levin & Peres, 2017; Wei et al., 2020):

**Definition B.2** (Mixing time). The mixing time of an ergodic MDP with respect to a policy $\boldsymbol{\pi}$ and transition kernel $\boldsymbol{p}$ is defined as

$$t_{\mathrm{mix}}^{\boldsymbol{\pi},\boldsymbol{p}} := \min\left\{ t \geq 1 \;\Big|\; \left\| \boldsymbol{P}_s^{\boldsymbol{\pi},(t)} - \boldsymbol{d}^{\boldsymbol{\pi},\boldsymbol{p}} \right\|_1 \leq \frac{1}{4}, \; \forall s \in \mathcal{S} \right\}, \tag{8}$$

where $\boldsymbol{P}_s^{\boldsymbol{\pi},(t)}$ is the $s$-th row of the $t$-step transition matrix.

For analytical convenience, we define the upper bound on the overall mixing time as $t_{\mathrm{mix}} := \max_{\boldsymbol{\pi}\in\Pi, \boldsymbol{p}\in\mathcal{P}} t_{\mathrm{mix}}^{\boldsymbol{\pi},\boldsymbol{p}}$. This represents the maximum time, across all policies and transition kernels, required for the state distribution to be within $1/4$ of the stationary distribution.

**Lemma B.3.** *(Policy performance difference lemma) For any* $\boldsymbol{\pi}, \boldsymbol{\pi}' \in \Pi$ *and* $\boldsymbol{p} \in (\Delta^S)^{S\times A}$, *we have*

$$J(\boldsymbol{\pi}, \boldsymbol{p}) - J(\boldsymbol{\pi}', \boldsymbol{p}) \;=\; \sum_{s\in\mathcal{S}} d_s^{\boldsymbol{\pi},\boldsymbol{p}} \sum_{a\in\mathcal{A}} (\pi_{sa} - \pi_{sa}') \cdot q_{sa}^{\boldsymbol{\pi}',\boldsymbol{p}}. \tag{9}$$

*Proof of Lemma B.3.* Using Bellman equation on the differential action value function, we have

$$\sum_s d_s^{\boldsymbol{\pi},\boldsymbol{p}} \sum_a \pi_{sa} q_{sa}^{\boldsymbol{\pi}',\boldsymbol{p}} \;=\; \sum_s d_s^{\boldsymbol{\pi},\boldsymbol{p}} \sum_a \pi_{sa} \sum_{s'} p_{sas'} \left( c_{sas'} - J(\boldsymbol{\pi}',\boldsymbol{p}) + v_{s'}^{\boldsymbol{\pi}',\boldsymbol{p}} \right)$$

$$= \; J(\boldsymbol{\pi},\boldsymbol{p}) - J(\boldsymbol{\pi}',\boldsymbol{p}) + \sum_{s'} \underbrace{\left( \sum_s d_s^{\boldsymbol{\pi},\boldsymbol{p}} \sum_a \pi_{sa} p_{sas'} \right)}_{d_{s'}^{\boldsymbol{\pi},\boldsymbol{p}}} v_{s'}^{\boldsymbol{\pi}',\boldsymbol{p}}$$

$$= \; J(\boldsymbol{\pi},\boldsymbol{p}) - J(\boldsymbol{\pi}',\boldsymbol{p}) + \sum_{s'} d_{s'}^{\boldsymbol{\pi},\boldsymbol{p}} v_{s'}^{\boldsymbol{\pi}',\boldsymbol{p}}$$

$$= \; J(\boldsymbol{\pi},\boldsymbol{p}) - J(\boldsymbol{\pi}',\boldsymbol{p}) + \sum_s d_s^{\boldsymbol{\pi},\boldsymbol{p}} \sum_a \pi_{sa}' q_{sa}^{\boldsymbol{\pi}',\boldsymbol{p}},$$

where the second equality is obtained by using the fact that $J(\boldsymbol{\pi},\boldsymbol{p}) = \sum_s d_s^{\boldsymbol{\pi},\boldsymbol{p}} \sum_a \pi_{sa} \sum_{s'} p_{sas'} c_{sas'}$ and $d_{s'}^{\boldsymbol{\pi},\boldsymbol{p}} = \sum_s d_s^{\boldsymbol{\pi},\boldsymbol{p}} \sum_a \pi_{sa} p_{sas'}$. $\square$

**Lemma B.4.** *(Bounds for differential value functions) For an ergodic MDP satisfying Assumption 2.1 with any* $\boldsymbol{\pi} \in \Pi$ *and* $\boldsymbol{p} \in (\Delta^S)^{S\times A}$, *we have for any* $s \in \mathcal{S}$, $a \in \mathcal{A}$,

$$|v_s^{\boldsymbol{\pi},\boldsymbol{p}}| \;\leq\; 5t_{mix}, \quad and \quad |q_{sa}^{\boldsymbol{\pi},\boldsymbol{p}}| \;\leq\; 7t_{mix}.$$

*Proof of Lemma B.4.* By the identity $v_s^{\boldsymbol{\pi},\boldsymbol{P}}$ satisfies, we have

$$
\begin{aligned}
|v_s^{\boldsymbol{\pi},\boldsymbol{P}}| &= \left| \sum_{t=0}^{\infty} \sum_{s'} \left( P_{ss'}^{\boldsymbol{\pi},t} - d_{s'}^{\boldsymbol{\pi},\boldsymbol{P}} \right) c_{s'}^{\boldsymbol{\pi},\boldsymbol{P}} \right| \\
&\leq \sum_{t=0}^{\infty} \| \boldsymbol{P}_s^{\boldsymbol{\pi},t} - \boldsymbol{d}^{\boldsymbol{\pi},\boldsymbol{P}} \|_1 \| \boldsymbol{c}^{\boldsymbol{\pi},\boldsymbol{P}} \|_{\infty} \\
&\leq \sum_{t=0}^{2t_{\text{mix}}-1} \| \boldsymbol{P}_s^{\boldsymbol{\pi},t} - \boldsymbol{d}^{\boldsymbol{\pi},\boldsymbol{P}} \|_1 + \sum_{i=2}^{\infty} \sum_{t=it_{\text{mix}}}^{(i+1)t_{\text{mix}}-1} \| \boldsymbol{P}_s^{\boldsymbol{\pi},t} - \boldsymbol{d}^{\boldsymbol{\pi},\boldsymbol{P}} \|_1 \\
&\leq 4t_{\text{mix}} + \sum_{i=2}^{\infty} 2 \cdot 2^{-i} \cdot t_{\text{mix}} \\
&\leq 5t_{\text{mix}},
\end{aligned}
$$

where the penultimate equality is obtained by applying Corollary 13.1 of (Wei et al., 2020). Therefore, we also obtain

$$
\begin{aligned}
|q_{sa}^{\boldsymbol{\pi},\boldsymbol{P}}| &= \left| \sum_{s'} p_{sas'} \left( c_{sas'} - J(\boldsymbol{\pi},\boldsymbol{p}) + v_{s'}^{\boldsymbol{\pi},\boldsymbol{P}} \right) \right| \\
&\leq \left| \sum_{s'} p_{sas'} c_{sas'} \right| + \left| \sum_{s'} p_{sas'} J(\boldsymbol{\pi},\boldsymbol{p}) \right| + \left| \sum_{s'} p_{sas'} v_{s'}^{\boldsymbol{\pi},\boldsymbol{P}} \right| \\
&\leq 1 + 1 + 5t_{\text{mix}} \\
&\leq 7t_{\text{mix}},
\end{aligned}
$$

where we bound the average reward objective as

$$
|J(\boldsymbol{\pi},\boldsymbol{p})| = \left| \mathbb{E}_{s \sim \boldsymbol{d}^{\boldsymbol{\pi},\boldsymbol{p}}, a \sim \boldsymbol{\pi}_s, s' \sim \boldsymbol{p}_{sa}} [c_{sas'}] \right| \leq 1.
$$

$\square$

It is worth noting that, the original upper bound result of the differential action value function in (Wei et al., 2020) missed the bound of the objective. It is worth noting that, the original upper bound result of the differential action value function in (Wei et al., 2020) missed the bound of the objective. Here, we revise the original result to the current new one.

Here we include the result showing the form of gradient over $\boldsymbol{\pi}$ for the sake of completeness.

**Lemma B.5.** *For any policy* $\boldsymbol{\pi} \in \Pi$ *and transition kernel* $\boldsymbol{p} \in (\Delta^S)^{S \times A}$, *the gradient of* $J(\boldsymbol{\pi},\boldsymbol{p})$ *over* $\boldsymbol{\pi}$ *has the analytical form as follows:*

$$
\frac{\partial J(\boldsymbol{\pi},\boldsymbol{p})}{\partial \pi_{sa}} = d_s^{\boldsymbol{\pi},\boldsymbol{p}} \cdot q_{sa}^{\boldsymbol{\pi},\boldsymbol{p}}.
$$

*Proof of Lemma B.5.* We now derive the form of partial derivative for $\pi_{sa}$ to obtain (6). Note that for any $s \in \mathcal{S}$, the gradient of the differential value function can be written as

$$
\begin{aligned}
\frac{\partial v_s^{\boldsymbol{\pi},\boldsymbol{P}}}{\partial \pi_{\hat{s}\hat{a}}} &= \frac{\partial}{\partial \pi_{\hat{s}\hat{a}}} \left[ \sum_a \pi_{sa} q_{sa}^{\boldsymbol{\pi},\boldsymbol{P}} \right] \\
&= \sum_a \left[ \frac{\partial \pi_{sa}}{\partial \pi_{\hat{s}\hat{a}}} q_{sa}^{\boldsymbol{\pi},\boldsymbol{P}} + \pi_{sa} \frac{\partial q_{sa}^{\boldsymbol{\pi},\boldsymbol{P}}}{\partial \pi_{\hat{s}\hat{a}}} \right] \\
&= \sum_a \left[ \frac{\partial \pi_{sa}}{\partial \pi_{\hat{s}\hat{a}}} q_{sa}^{\boldsymbol{\pi},\boldsymbol{P}} + \pi_{sa} \frac{\partial}{\partial \pi_{\hat{s}\hat{a}}} \left( \sum_{s'} p_{sas'} \left( c_{sas'} - J(\boldsymbol{\pi},\boldsymbol{p}) + v_{s'}^{\boldsymbol{\pi},\boldsymbol{P}} \right) \right) \right] \\
&= \sum_a \left[ \frac{\partial \pi_{sa}}{\partial \pi_{\hat{s}\hat{a}}} q_{sa}^{\boldsymbol{\pi},\boldsymbol{P}} + \pi_{sa} \left( -\frac{\partial J(\boldsymbol{\pi},\boldsymbol{p})}{\partial \pi_{\hat{s}\hat{a}}} + \sum_{s'} p_{sas'} \frac{\partial v_{s'}^{\boldsymbol{\pi},\boldsymbol{P}}}{\partial \pi_{\hat{s}\hat{a}}} \right) \right].
\end{aligned}
$$

Multiplying each side with $d_s^{\boldsymbol{\pi},\boldsymbol{p}}$, taking the summation over $s \in \mathcal{S}$, and rearranging terms, we can obtain

$$
\begin{aligned}
\frac{\partial J(\boldsymbol{\pi},\boldsymbol{p})}{\partial \pi_{\hat{s}\hat{a}}} &= \sum_s d_s^{\boldsymbol{\pi},\boldsymbol{p}} \left( \sum_a \left[ \frac{\partial \pi_{sa}}{\partial \pi_{\hat{s}\hat{a}}} q_{sa}^{\boldsymbol{\pi},\boldsymbol{p}} + \pi_{sa} \sum_{s'} p_{sas'} \frac{\partial v_{s'}^{\boldsymbol{\pi},\boldsymbol{p}}}{\partial \pi_{\hat{s}\hat{a}}} \right] - \frac{\partial v_s^{\boldsymbol{\pi},\boldsymbol{p}}}{\partial \pi_{\hat{s}\hat{a}}} \right) \\
&= \sum_s d_s^{\boldsymbol{\pi},\boldsymbol{p}} \sum_a \frac{\partial \pi_{sa}}{\partial \pi_{\hat{s}\hat{a}}} q_{sa}^{\boldsymbol{\pi},\boldsymbol{p}} + \sum_s d_s^{\boldsymbol{\pi},\boldsymbol{p}} \sum_a \pi_{sa} \sum_{s'} p_{sas'} \frac{\partial v_{s'}^{\boldsymbol{\pi},\boldsymbol{p}}}{\partial \pi_{\hat{s}\hat{a}}} - \sum_s d_s^{\boldsymbol{\pi},\boldsymbol{p}} \frac{\partial v_s^{\boldsymbol{\pi},\boldsymbol{p}}}{\partial \pi_{\hat{s}\hat{a}}} \\
&= \sum_s d_s^{\boldsymbol{\pi},\boldsymbol{p}} \sum_a \frac{\partial \pi_{sa}}{\partial \pi_{\hat{s}\hat{a}}} q_{sa}^{\boldsymbol{\pi},\boldsymbol{p}} = d_{\hat{s}}^{\boldsymbol{\pi},\boldsymbol{p}} q_{\hat{s}\hat{a}}^{\boldsymbol{\pi},\boldsymbol{p}}.
\end{aligned}
$$

$\square$

By introducing the *distribution mismatch coefficient* between two stationary distributions $\left\| \frac{\boldsymbol{d}^{\boldsymbol{\pi},\boldsymbol{p}}}{\boldsymbol{d}^{\boldsymbol{\pi}',\boldsymbol{p}'}} \right\|_\infty$ and $M :=$ $\sup_{\boldsymbol{\pi}_1,\boldsymbol{\pi}_2 \in \Pi, \boldsymbol{p}_1,\boldsymbol{p}_2 \in \mathcal{P}} \left\| \frac{\boldsymbol{d}^{\boldsymbol{\pi}_1,\boldsymbol{p}_1}}{\boldsymbol{d}^{\boldsymbol{\pi}_2,\boldsymbol{p}_2}} \right\|_\infty < \infty$, we can reach the gradient dominance condition that AMDPs satisfy.

**Lemma B.6.** *(Policy Gradient Dominance ([Kumar et al., 2024b](#))) For any $\boldsymbol{p} \in (\Delta^S)^{S \times A}$, we let $\boldsymbol{\pi}^\star$ be one of optimal policies over $\boldsymbol{p}$, i.e., $\boldsymbol{\pi}^\star \in \arg\min_{\boldsymbol{\pi} \in \Pi} J(\boldsymbol{\pi},\boldsymbol{p})$, then we have,*

$$
J(\boldsymbol{\pi},\boldsymbol{p}) - J(\boldsymbol{\pi}^\star,\boldsymbol{p}) \le M \cdot \max_{\bar{\boldsymbol{\pi}} \in \Pi} \langle (\boldsymbol{\pi} - \bar{\boldsymbol{\pi}}, \nabla_{\boldsymbol{\pi}} J(\boldsymbol{\pi},\boldsymbol{p}) \rangle. \tag{10}
$$

*Proof of Lemma B.6.* By the policy difference performance lemma (Lemma B.3), we have for any $\boldsymbol{\pi} \in \Pi$ and $\boldsymbol{p} \in \mathcal{P}$, we have

$$
J(\boldsymbol{\pi}^\star,\boldsymbol{p}) - J(\boldsymbol{\pi},\boldsymbol{p}) = \sum_{s \in \mathcal{S}} d_s^{\boldsymbol{\pi}^\star,\boldsymbol{p}} \sum_{a \in \mathcal{A}} (\pi_{sa}^\star - \pi_{sa}) \cdot q_{sa}^{\boldsymbol{\pi},\boldsymbol{p}}.
$$

Then, we obtain that

$$
\begin{aligned}
0 \le J(\boldsymbol{\pi},\boldsymbol{p}) - J(\boldsymbol{\pi}^\star,\boldsymbol{p}) &= \sum_{s \in \mathcal{S}} d_s^{\boldsymbol{\pi}^\star,\boldsymbol{p}} \sum_{a \in \mathcal{A}} (\pi_{sa} - \pi_{sa}^\star) \cdot q_{sa}^{\boldsymbol{\pi},\boldsymbol{p}} \\
&= \sum_{s \in \mathcal{S}} \frac{d_s^{\boldsymbol{\pi}^\star,\boldsymbol{p}}}{d_s^{\boldsymbol{\pi},\boldsymbol{p}}} d_s^{\boldsymbol{\pi},\boldsymbol{p}} \sum_{a \in \mathcal{A}} (\pi_{sa} - \pi_{sa}^\star) \cdot q_{sa}^{\boldsymbol{\pi},\boldsymbol{p}} \\
&\le \sum_{s \in \mathcal{S}} \frac{d_s^{\boldsymbol{\pi}^\star,\boldsymbol{p}}}{d_s^{\boldsymbol{\pi},\boldsymbol{p}}} d_s^{\boldsymbol{\pi},\boldsymbol{p}} \max_{\bar{\boldsymbol{\pi}}_s \in \Pi_s} \left\{ \sum_{a \in \mathcal{A}} (\bar{\pi}_{sa} - \pi_{sa}^\star) \cdot q_{sa}^{\boldsymbol{\pi},\boldsymbol{p}} \right\} \\
&\overset{(a)}{\le} \left\| \frac{\boldsymbol{d}^{\boldsymbol{\pi}^\star,\boldsymbol{p}}}{\boldsymbol{d}^{\boldsymbol{\pi},\boldsymbol{p}}} \right\|_\infty \sum_{s \in \mathcal{S}} d_s^{\boldsymbol{\pi},\boldsymbol{p}} \max_{\bar{\boldsymbol{\pi}}_s \in \Pi_s} \left\{ \sum_{a \in \mathcal{A}} (\bar{\pi}_{sa} - \pi_{sa}^\star) \cdot q_{sa}^{\boldsymbol{\pi},\boldsymbol{p}} \right\} \\
&= M \cdot \max_{\bar{\boldsymbol{\pi}} \in \Pi} \langle \boldsymbol{\pi} - \bar{\boldsymbol{\pi}}, \nabla_{\boldsymbol{\pi}} J(\boldsymbol{\pi},\boldsymbol{p}) \rangle,
\end{aligned}
$$

where the inequality $(a)$ is obtained due to

$$
J(\boldsymbol{\pi},\boldsymbol{p}) - J(\boldsymbol{\pi}^\star,\boldsymbol{p}) \ge 0, \quad \text{and} \quad \sum_{a \in \mathcal{A}} (\bar{\pi}_{sa} - \pi_{sa}^\star) \cdot q_{sa}^{\boldsymbol{\pi},\boldsymbol{p}} \ge 0, \ \forall s \in \mathcal{S}.
$$

$\square$

## B.2. Standard definitions and results in optimization

In this subsection, we present some standard optimization definitions ([Ghadimi & Lan](#), [2016](#); [Beck](#), [2017](#)), which are used in our work. Consider the following optimization problem

$$
\min_{\boldsymbol{x} \in \mathcal{X}} h(\boldsymbol{x}) \tag{11}
$$

with $\mathcal{X}$ being a nonempty closed and convex set and $h: \mathbb{R}^d \to \mathbb{R}$ being proper, closed and $\ell$-smooth. We first introduce the crucial definitions of smoothness and Lipschitz continuity.

**Definition B.7.** A function $h: \mathcal{X} \to \mathbb{R}$ is *L-Lipschitz* if for any $\boldsymbol{x}_1, \boldsymbol{x}_2 \in \mathcal{X}$, we have that $\|h(\boldsymbol{x}_1) - h(\boldsymbol{x}_2)\| \leq L\|\boldsymbol{x}_1 - \boldsymbol{x}_2\|$, and *$\ell$-smooth* if for any $\boldsymbol{x}_1, \boldsymbol{x}_2 \in \mathcal{X}$, we have $\|\nabla h(\boldsymbol{x}_1) - \nabla h(\boldsymbol{x}_2)\| \leq \ell\|\boldsymbol{x}_1 - \boldsymbol{x}_2\|$.

Another common definition we need to clarify is the indicator function.

**Definition B.8** (Indicator functions). For any subset $\mathcal{X} \subseteq \mathbb{R}^d$, the indicator function of $\mathcal{X}$ is defined to be the extended real-valued function given by

$$\mathbb{I}_{\mathcal{X}}(\boldsymbol{x}) = \begin{cases} 0, & \boldsymbol{x} \in \mathcal{X}, \\ \infty, & \boldsymbol{x} \notin \mathcal{X}. \end{cases}$$

**Definition B.9.** The Fréchet sub-differential of a function $h : \mathcal{X} \to \mathbb{R}$ at point $\boldsymbol{x} \in \mathcal{X}$ is defined as the set $\partial h(\boldsymbol{x}) = \{u| \liminf_{\boldsymbol{x}' \to \boldsymbol{x}} h(\boldsymbol{x}') - h(\boldsymbol{x}) - \langle \boldsymbol{u}, \boldsymbol{x}' - \boldsymbol{x}\rangle/\|\boldsymbol{x}' - \boldsymbol{x}\| \geq 0\}$.

Then, a common lemma is provided to illustrate a basic property that a smooth function satisfies.

**Lemma B.10.** *Let $h: \mathcal{X} \to \mathbb{R}$ be $\ell$-smooth, then it is a $\ell$-weakly convex function.*

*Proof of Lemma B.10.* Let $r(t) := h(x + t(x' - x))$, for any $x, x' \in \mathcal{X}$. The following holds true

$$h(x) = r(0), \quad \text{and} \quad h(x') = r(1).$$

Then, we observe that

$$h(x') - h(x) = r(1) - r(0) = \int_0^1 \nabla r(t) dt,$$

where

$$\nabla r(t) = \nabla h(x + t(x' - x))^\top (x' - x).$$

We complete the proof as

$$\begin{aligned}
\|h(x') - h(x) - \nabla h(x)^\top (x' - x)\| &\leq \left\|\int_0^1 \nabla r(t) dt - \nabla h(x)^\top (x' - x)\right\| \\
&\leq \int_0^1 \left\|\nabla r(t) - \nabla h(x)^\top (x' - x)\right\| dt \\
&= \int_0^1 \left\|\nabla h(x + t(x' - x))^\top (x' - x) - \nabla h(x)^\top (x' - x)\right\| dt \\
&\leq \int_0^1 \|\nabla h(x + t(x' - x)) - \nabla h(x)\| \cdot \|(x' - x)\| dt \\
&\leq \int_0^1 t\ell\|x' - x\|^2 dt = \frac{\ell}{2}\|x' - x\|^2.
\end{aligned}$$

$\square$

We present the standard optimization results from (Ghadimi & Lan, 2016; Beck, 2017) used in our proofs. We then denote the optimal $h$ value by $h(\boldsymbol{x}^\star)$.

**Definition B.11** (Gradient Mapping). The gradient mapping $G^\beta(\boldsymbol{x})$ is defined as

$$G^\beta(\boldsymbol{x}) := \frac{1}{\beta}\left(\boldsymbol{x} - \text{Proj}_{\mathcal{X}}(\boldsymbol{x} - \beta \nabla h(\boldsymbol{x}))\right), \tag{12}$$

where the operator $\text{Proj}_{\mathcal{X}}$ is the projection onto $\mathcal{X}$.

**Theorem B.12.** *(Beck, 2017, Theorem 10.15) Let $\{\boldsymbol{x}_k\}_{k\geq 0}$ be the sequence generated by the projected gradient descent algorithm for solving the problem (11) either with a constant step size defined by $\beta = 1/\ell$. Then*

1. *The sequence $\{h(\boldsymbol{x}_k)\}_{k\geq 0}$ is non-increasing.*

2. *$G^\beta(\boldsymbol{x}_k) \to 0$ as $t \to 0$.*

3. $\min_{t \in \{0, \cdots, T-1\}} \|G^\beta(\boldsymbol{x}_t)\| \leq \sqrt{\frac{2\ell(h(\boldsymbol{x}_0) - h(\boldsymbol{x}^\star))}{T}}$.

**Lemma B.13.** *(Ghadimi & Lan, 2016, Lemma 3) Let $\boldsymbol{x}^+ = \boldsymbol{x} - \beta G^\beta(\boldsymbol{x})$. If $\|G^\beta(\boldsymbol{x})\| \leq \epsilon$, then*

$$-\nabla h(\boldsymbol{x}^+) \in \mathcal{N}_\mathcal{X}(\boldsymbol{x}^+) + 2\epsilon\mathcal{B}(1), \tag{13}$$

*where $\mathcal{N}_\mathcal{X}$ is the norm cone of the set $\mathcal{X}$ and $\mathcal{B}(r) := \{\boldsymbol{x} \in \mathbb{R}^d : \|\boldsymbol{x}\| \leq r\}$.*

### B.3. Moreau Envelope and its property

For smooth function $h(x)$, a point $x \in \mathcal{X}$ is defined as the first-order stationary point (FOSP) when $0 \in \partial h(x)$. However, this notion of stationarity can be very restrictive when optimizing nonsmooth functions (Lin et al., 2020). In respond to this issue, an alternative measure of the first-order stationarity is proposed based on the construction of the Moreau envelope (Thekumparampil et al., 2019).

**Definition B.14.** For function $h : \mathcal{X} \to \mathbb{R}$ and $\lambda > 0$, the Moreau envelope function of $h$ is given by

$$h_\lambda(x) := \min_{x' \in \mathcal{X}} \left\{ h(x') + \frac{1}{2\lambda} \|x - x'\|^2 \right\}. \tag{14}$$

**Definition B.15.** Given an $\ell$-weakly convex function $h$, we say that $x^\star$ is an $\epsilon$-first order stationary point ($\epsilon$-FOSP) if, $\|\nabla h_{1/2\ell}(x^\star)\| \leq \epsilon$, where $h_{\frac{1}{2\ell}}(x)$ is the Moreau envelope function of $h$ with parameter $\lambda = 1/2\ell$.

The following lemma connects $\ell$-weakly convex function and its Moreau envelope function and will be useful in our proofs.

**Lemma B.16.** *Suppose the function $h \colon \mathcal{X} \subseteq \mathbb{R}^n \to \mathbb{R}$ is $\ell$-weakly convex and may be not differentiable at any point. Then for each $\lambda < \ell$:*

1. *The Moreau envelope function $h_\lambda$ is $C^1$-smooth with the gradient given by,*

$$\nabla h_\lambda(\boldsymbol{x}) = \lambda^{-1} \left( \boldsymbol{x} - \arg\min_{\boldsymbol{w} \in \Pi} \left( h(\boldsymbol{w}) + \frac{1}{2\lambda} \|\boldsymbol{x} - \boldsymbol{w}\|^2 \right) \right)$$

   *Meanwhile, by introducing $\hat{\boldsymbol{x}}_\lambda(\boldsymbol{x}) := \arg\min_{\boldsymbol{w} \in \mathcal{X}} h(\boldsymbol{w}) + 1/2\lambda \|\boldsymbol{x} - \boldsymbol{w}\|^2$, we have $\|\hat{\boldsymbol{x}}_\lambda(\boldsymbol{x}) - \boldsymbol{x}\| = \lambda\|\nabla h_\lambda(\boldsymbol{x})\|$.*

2. *The inequality $\|\nabla h_\lambda(\boldsymbol{x})\| \leq \epsilon$ implies $\|\hat{\boldsymbol{x}}_\lambda(\boldsymbol{x}) - \boldsymbol{x}\| \leq \lambda\epsilon$ and $\exists \boldsymbol{\xi} \in \partial h(\hat{\boldsymbol{x}}_\lambda(\boldsymbol{x}))$ such that*

$$-\boldsymbol{\xi} \in \mathcal{N}_\mathcal{X}(\hat{\boldsymbol{x}}_\lambda(\boldsymbol{x})) + \frac{1}{\lambda} (\hat{\boldsymbol{x}}_\lambda(\boldsymbol{x}) - \boldsymbol{x}) \subseteq \mathcal{N}_\mathcal{X}(\hat{\boldsymbol{x}}_\lambda(\boldsymbol{x})) + \frac{1}{\lambda} \|\hat{\boldsymbol{x}}_\lambda(\boldsymbol{x}) - \boldsymbol{x}\| \mathcal{B}(1),$$

   *where $\mathcal{N}_\mathcal{X}(\hat{\boldsymbol{x}}_\lambda(\boldsymbol{x}))$ is defined as the normal cone of $\mathcal{X}$ at $\hat{\boldsymbol{x}}_\lambda(\boldsymbol{x})$ and $\mathcal{B}(r) := \{\boldsymbol{x} \in \mathbb{R}^n : \|\boldsymbol{x}\| \leq r\}$. In particular, when $\mathcal{X} = \mathbb{R}^n$, we have that*

$$\min_{\boldsymbol{\xi} \in \partial h(\hat{\boldsymbol{x}}_\lambda(\boldsymbol{x}))} \|\boldsymbol{\xi}\| \leq \frac{1}{\lambda} \|\hat{\boldsymbol{x}}_\lambda(\boldsymbol{x}) - \boldsymbol{x}\| = \|\nabla h_\lambda(\boldsymbol{x})\|.$$

*Proof of Lemma B.16.* First, the analytical form of the Moreau envelope function's gradient is well-established by Proposition 13.37 in (Rockafellar & Wets, 2009). Then, let us consider the optimality appearing in the definition of the Moreau envelope function. Define $\phi_{\boldsymbol{x}}(\boldsymbol{y}) = h(\boldsymbol{y}) + \mathbb{I}_\mathcal{X}(\boldsymbol{y}) + \frac{1}{2\lambda} \|\boldsymbol{x} - \boldsymbol{y}\|^2$, and then we notice that for any $\boldsymbol{x} \in \mathbb{R}^n$, $\hat{\boldsymbol{x}}_\lambda(\boldsymbol{x})$ is the optimal solution of $\phi_{\boldsymbol{x}}(\boldsymbol{y})$, which leads to

$$\phi_{\boldsymbol{x}}(\hat{\boldsymbol{x}}_\lambda(\boldsymbol{x})) = \min_{\boldsymbol{y} \in \mathbb{R}^n} \phi_{\boldsymbol{x}}(\boldsymbol{y}) \iff 0 \in \partial \left( h(\boldsymbol{y}) + \mathbb{I}_\mathcal{X}(\boldsymbol{y}) + \frac{1}{2\lambda} \|\boldsymbol{x} - \boldsymbol{y}\|^2 \right) \Big|_{\boldsymbol{y}=\hat{\boldsymbol{x}}_\lambda(\boldsymbol{x})},$$

$$\iff 0 \in \boldsymbol{\xi} + \mathcal{N}_\mathcal{X}(\hat{\boldsymbol{x}}_\lambda(\boldsymbol{x})) + \frac{1}{\lambda} (\hat{\boldsymbol{x}}_\lambda(\boldsymbol{x}) - \boldsymbol{x})$$

$$\iff -\boldsymbol{\xi} \in \mathcal{N}_\mathcal{X}(\hat{\boldsymbol{x}}_\lambda(\boldsymbol{x})) + \frac{1}{\lambda} (\hat{\boldsymbol{x}}_\lambda(\boldsymbol{x}) - \boldsymbol{x}). \tag{15}$$

The last inequality of 15 implies that, for any $\boldsymbol{z} \in \mathbb{R}^n$,

$$
\langle \boldsymbol{\xi} + \frac{1}{\lambda} \left( \hat{\boldsymbol{x}}_\lambda(\boldsymbol{x}) - \boldsymbol{x} \right), \boldsymbol{z} - \hat{\boldsymbol{x}}_\lambda(\boldsymbol{x}) \rangle \geq 0
$$

$$
\iff \langle -\boldsymbol{\xi}, \boldsymbol{z} - \hat{\boldsymbol{x}}_\lambda(\boldsymbol{x}) \rangle \leq \langle \frac{1}{\lambda} \left( \hat{\boldsymbol{x}}_\lambda(\boldsymbol{x}) - \boldsymbol{x} \right), \boldsymbol{z} - \hat{\boldsymbol{x}}_\lambda(\boldsymbol{x}) \rangle, \ \forall \boldsymbol{z} \in \mathbb{R}^n
$$

$$
\iff \langle -\boldsymbol{\xi}, \boldsymbol{z} - \hat{\boldsymbol{x}}_\lambda(\boldsymbol{x}) \rangle \leq \frac{1}{\lambda} \| \hat{\boldsymbol{x}}_\lambda(\boldsymbol{x}) - \boldsymbol{x} \| \cdot \| \boldsymbol{z} - \hat{\boldsymbol{x}}_\lambda(\boldsymbol{x}) \|, \ \forall \boldsymbol{z} \in \mathbb{R}^n
$$

$$
\iff \langle -\boldsymbol{\xi}, \boldsymbol{z} - \hat{\boldsymbol{x}}_\lambda(\boldsymbol{x}) \rangle \leq \frac{1}{\lambda} \| \hat{\boldsymbol{x}}_\lambda(\boldsymbol{x}) - \boldsymbol{x} \|, \ \forall \boldsymbol{z} \in \mathbb{R}^n, \ \| \boldsymbol{z} - \hat{\boldsymbol{x}}_\lambda(\boldsymbol{x}) \| = 1,
$$

which is our desired result. Specifically, while we consider the case $\mathcal{X} = \mathbb{R}^n$, we have

$$
(15) \iff \frac{1}{\lambda} \left( \boldsymbol{x} - \hat{\boldsymbol{x}}_\lambda(\boldsymbol{x}) \right) \in \partial h(\hat{\boldsymbol{x}}),
$$

which implies that

$$
\min_{\boldsymbol{\xi} \in \partial h(\hat{\boldsymbol{x}}_\lambda(\boldsymbol{x}))} \| \boldsymbol{\xi} \| \leq \frac{1}{\lambda} \| \hat{\boldsymbol{x}}_\lambda(\boldsymbol{x}) - \boldsymbol{x} \| = \| \nabla h_\lambda(\boldsymbol{x}) \|.
$$

$\square$

Based on the above properties of the Moreau envelope of a weakly convex function, a small gradient $\| \nabla h_\lambda(\boldsymbol{x}) \|$ implies that $\boldsymbol{x}$ is near some point $\hat{\boldsymbol{x}}_\lambda(\boldsymbol{x})$ that is nearly stationary for $h$. In the broader non-smooth setting, the norm of the gradient, $\| \nabla h_\lambda(\boldsymbol{x}) \|$ has an intuitive interpretation in terms of near-stationarity for the target problem $\Phi(\boldsymbol{x})$ (Beck, 2017; Davis & Drusvyatskiy, 2019; Drusvyatskiy & Paquette, 2019).

### B.4. Danskin's Theorem

We also need to introduce the following Danskin's Theorem, which helps prove our global convergence theorem.

**Proposition B.17.** *(Bertsekas, 2016, Proposition B.25) Let $\mathcal{Z} \subseteq \mathbb{R}^m$ be a compact set, and let $h : \mathbb{R}^n \times \mathcal{Z} \to \mathbb{R}$ be continuous function and such that $h(\cdot, \boldsymbol{z}) : \mathbb{R}^n \to \mathbb{R}$ is convex for each $\boldsymbol{z} \in \mathcal{Z}$. If $h(\cdot, \boldsymbol{z})$ is differentiable for all $\boldsymbol{z} \in \mathcal{Z}$ and $\nabla h(\boldsymbol{x}, \cdot)$ is continuous on $\mathcal{Z}$ for each $\boldsymbol{x}$, then for $f(\boldsymbol{x}) := \max_{\boldsymbol{z} \in \mathcal{Z}} h(\boldsymbol{x}, \boldsymbol{z})$ and any $\boldsymbol{x} \in \mathbb{R}^n$,*

$$
\partial f(\boldsymbol{x}) = \mathrm{conv} \left\{ \nabla_x h(\boldsymbol{x}, \boldsymbol{z}) \,\middle|\, \boldsymbol{z} \in \arg\max_{\boldsymbol{z} \in \mathcal{Z}} h(\boldsymbol{x}, \boldsymbol{z}) \right\}.
$$

## C. Omitted Proofs in Section 3

The first key step in the analysis of RP2G is to determine the continuity property of this non-convex, non-differentiable (i.e., non-smooth) objective function $\Psi(\boldsymbol{\pi})$. To do so, we derive the following sensitivity bounds for differential value functions, which play an important role in establishing the continuity conditions.

*Proof of Lemma 3.1.* First, we provide the sensitivity analysis on the policy-induced cost and the state transition probability as follows:

$$
|c_s^{\boldsymbol{\pi}, \boldsymbol{p}} - c_s^{\boldsymbol{\pi}', \boldsymbol{p}}| = \left| \sum_{a'} (\pi_{sa} - \pi'_{sa}) \sum_{s'} p_{sas'} c_{sas'} \right| \leq \| \boldsymbol{\pi}_s - \boldsymbol{\pi}'_s \|_1, \ \forall s \in \mathcal{S},
$$

$$
|P_{ss'}^{\boldsymbol{\pi}} - P_{ss'}^{\boldsymbol{\pi}'}| = \left| \sum_{a} (\pi_{sa} - \pi'_{sa}) p_{sas'} \right| \leq \| \boldsymbol{\pi}_s - \boldsymbol{\pi}'_s \|_1, \ \forall s, s' \in \mathcal{S}.
$$

Next, we turn to derive our desired results. Notice that, the stationary distribution $d_s^{\boldsymbol{\pi}, \boldsymbol{p}}$ could be viewed as a particular average-reward objective with taking $\mathbf{1} \{ \cdot = s \}$ as the cost function, which is also bounded in $[0, 1]$. Therefore, by applying

the policy performance difference lemma (Lemma B.3),

$$
\begin{aligned}
|J(\boldsymbol{\pi}, \boldsymbol{p}) - J(\boldsymbol{\pi}', \boldsymbol{p})| &= \left| \sum_s d_s^{\boldsymbol{\pi}, \boldsymbol{p}} \sum_a (\pi_{sa} - \pi'_{sa}) \, q_{sa}^{\boldsymbol{\pi}', \boldsymbol{p}} \right| \\
&\leq \sum_s |d_s^{\boldsymbol{\pi}, \boldsymbol{p}}| \cdot \sum_a |(\pi_{sa} - \pi'_{sa})| \cdot |q_{sa}^{\boldsymbol{\pi}', \boldsymbol{p}}| \\
&\leq 7 t_{\text{mix}} \sum_s |d_s^{\boldsymbol{\pi}, \boldsymbol{p}}| \cdot \sum_a |(\pi_{sa} - \pi'_{sa})| && \text{(Due to } |q_{sa}^{\boldsymbol{\pi}', \boldsymbol{p}}| \leq 7 t_{\text{mix}}) \\
&\leq 7 t_{\text{mix}} \cdot \left( \max_s \sum_a |(\pi_{sa} - \pi'_{sa})| \right) \cdot \sum_{s \in \mathcal{S}} |d_s^{\boldsymbol{\pi}, \boldsymbol{p}}| \\
&\leq 7 t_{\text{mix}} \cdot \|\boldsymbol{\pi} - \boldsymbol{\pi}'\|_{1, \infty},
\end{aligned}
$$

which also leads to

$$
|d_s^{\boldsymbol{\pi}, \boldsymbol{p}} - d_s^{\boldsymbol{\pi}', \boldsymbol{p}}| \leq 7 t_{\text{mix}} \cdot \|\boldsymbol{\pi} - \boldsymbol{\pi}'\|_{1, \infty}.
$$

Recall the analytical form of the differential value function as $\boldsymbol{v}^{\boldsymbol{\pi}, \boldsymbol{p}} = (\boldsymbol{I} - \boldsymbol{P}^{\boldsymbol{\pi}} + \boldsymbol{P}^{\boldsymbol{\pi}, \infty})^{-1}(\boldsymbol{I} - \boldsymbol{P}^{\boldsymbol{\pi}, \infty}) \boldsymbol{c}^{\boldsymbol{\pi}, \boldsymbol{p}}$ (See Lemma B.1), and for simplicity of our analysis, we introduce $\boldsymbol{H}^{\boldsymbol{\pi}, \boldsymbol{p}} := (\boldsymbol{I} - \boldsymbol{P}^{\boldsymbol{\pi}} + \boldsymbol{P}^{\boldsymbol{\pi}, \infty})^{-1}(\boldsymbol{I} - \boldsymbol{P}^{\boldsymbol{\pi}, \infty})$. We note that

$$
\begin{aligned}
\|\boldsymbol{v}^{\boldsymbol{\pi}, \boldsymbol{p}} - \boldsymbol{v}^{\boldsymbol{\pi}', \boldsymbol{p}}\|_\infty &= \|\boldsymbol{H}^{\boldsymbol{\pi}, \boldsymbol{p}} \boldsymbol{c}^{\boldsymbol{\pi}, \boldsymbol{p}} - \boldsymbol{H}^{\boldsymbol{\pi}', \boldsymbol{p}} \boldsymbol{c}^{\boldsymbol{\pi}', \boldsymbol{p}}\|_\infty \\
&= \|\boldsymbol{H}^{\boldsymbol{\pi}, \boldsymbol{p}} \left( \boldsymbol{c}^{\boldsymbol{\pi}, \boldsymbol{p}} - \boldsymbol{c}^{\boldsymbol{\pi}', \boldsymbol{p}} \right) + \left( \boldsymbol{H}^{\boldsymbol{\pi}, \boldsymbol{p}} - \boldsymbol{H}^{\boldsymbol{\pi}', \boldsymbol{p}} \right) \boldsymbol{c}^{\boldsymbol{\pi}', \boldsymbol{p}}\|_\infty,
\end{aligned}
$$

where

$$
\begin{aligned}
\boldsymbol{H}^{\boldsymbol{\pi}, \boldsymbol{p}} - \boldsymbol{H}^{\boldsymbol{\pi}', \boldsymbol{p}} &= (\boldsymbol{I} - \boldsymbol{P}^{\boldsymbol{\pi}} + \boldsymbol{P}^{\boldsymbol{\pi}, \infty})^{-1}(\boldsymbol{I} - \boldsymbol{P}^{\boldsymbol{\pi}, \infty}) - (\boldsymbol{I} - \boldsymbol{P}^{\boldsymbol{\pi}'} + \boldsymbol{P}^{\boldsymbol{\pi}', \infty})^{-1}(\boldsymbol{I} - \boldsymbol{P}^{\boldsymbol{\pi}', \infty}) \\
&= \left( (\boldsymbol{I} - \boldsymbol{P}^{\boldsymbol{\pi}} + \boldsymbol{P}^{\boldsymbol{\pi}, \infty})^{-1} - (\boldsymbol{I} - \boldsymbol{P}^{\boldsymbol{\pi}'} + \boldsymbol{P}^{\boldsymbol{\pi}', \infty})^{-1} \right) (\boldsymbol{I} - \boldsymbol{P}^{\boldsymbol{\pi}, \infty}) \\
&\quad + (\boldsymbol{I} - \boldsymbol{P}^{\boldsymbol{\pi}'} + \boldsymbol{P}^{\boldsymbol{\pi}', \infty})^{-1} \left( \boldsymbol{P}^{\boldsymbol{\pi}', \infty} - \boldsymbol{P}^{\boldsymbol{\pi}, \infty} \right) \\
&= \left( (\boldsymbol{I} - \boldsymbol{P}^{\boldsymbol{\pi}} + \boldsymbol{P}^{\boldsymbol{\pi}, \infty})^{-1} \left( \boldsymbol{P}^{\boldsymbol{\pi}} - \boldsymbol{P}^{\boldsymbol{\pi}, \infty} - \boldsymbol{P}^{\boldsymbol{\pi}'} + \boldsymbol{P}^{\boldsymbol{\pi}', \infty} \right) (\boldsymbol{I} - \boldsymbol{P}^{\boldsymbol{\pi}'} + \boldsymbol{P}^{\boldsymbol{\pi}', \infty})^{-1} \right) (\boldsymbol{I} - \boldsymbol{P}^{\boldsymbol{\pi}, \infty}) \\
&\quad + (\boldsymbol{I} - \boldsymbol{P}^{\boldsymbol{\pi}'} + \boldsymbol{P}^{\boldsymbol{\pi}', \infty})^{-1} \left( \boldsymbol{P}^{\boldsymbol{\pi}', \infty} - \boldsymbol{P}^{\boldsymbol{\pi}, \infty} \right).
\end{aligned}
$$

Then, we obtain that

$$
\begin{aligned}
\|\boldsymbol{v}^{\boldsymbol{\pi}, \boldsymbol{p}} - \boldsymbol{v}^{\boldsymbol{\pi}', \boldsymbol{p}}\|_\infty &\leq \|\boldsymbol{H}^{\boldsymbol{\pi}, \boldsymbol{p}} \left( \boldsymbol{c}^{\boldsymbol{\pi}, \boldsymbol{p}} - \boldsymbol{c}^{\boldsymbol{\pi}', \boldsymbol{p}} \right)\|_\infty + \|(\boldsymbol{I} - \boldsymbol{P}^{\boldsymbol{\pi}'} + \boldsymbol{P}^{\boldsymbol{\pi}', \infty})^{-1} \left( \boldsymbol{P}^{\boldsymbol{\pi}', \infty} - \boldsymbol{P}^{\boldsymbol{\pi}, \infty} \right) \boldsymbol{c}^{\boldsymbol{\pi}', \boldsymbol{p}}\|_\infty \\
&\quad + \left\| \left( \boldsymbol{I} - \boldsymbol{P}^{\boldsymbol{\pi}} + \boldsymbol{P}^{\boldsymbol{\pi}, \infty} \right)^{-1} \left( \boldsymbol{P}^{\boldsymbol{\pi}} - \boldsymbol{P}^{\boldsymbol{\pi}, \infty} - \boldsymbol{P}^{\boldsymbol{\pi}'} + \boldsymbol{P}^{\boldsymbol{\pi}', \infty} \right) (\boldsymbol{I} - \boldsymbol{P}^{\boldsymbol{\pi}'} + \boldsymbol{P}^{\boldsymbol{\pi}', \infty})^{-1} \right) (\boldsymbol{I} - \boldsymbol{P}^{\boldsymbol{\pi}, \infty}) \boldsymbol{c}^{\boldsymbol{\pi}', \boldsymbol{p}} \right\|_\infty \\
&\overset{(a)}{\leq} \|\boldsymbol{H}^{\boldsymbol{\pi}, \boldsymbol{p}}\|_\infty \cdot \|\boldsymbol{c}^{\boldsymbol{\pi}, \boldsymbol{p}} - \boldsymbol{c}^{\boldsymbol{\pi}', \boldsymbol{p}}\|_\infty + \|(\boldsymbol{I} - \boldsymbol{P}^{\boldsymbol{\pi}'} + \boldsymbol{P}^{\boldsymbol{\pi}', \infty})^{-1}\|_\infty \cdot \|\boldsymbol{P}^{\boldsymbol{\pi}', \infty} - \boldsymbol{P}^{\boldsymbol{\pi}, \infty}\|_\infty \\
&\quad + \|(\boldsymbol{I} - \boldsymbol{P}^{\boldsymbol{\pi}} + \boldsymbol{P}^{\boldsymbol{\pi}, \infty})^{-1}\|_\infty \cdot \|\boldsymbol{P}^{\boldsymbol{\pi}} - \boldsymbol{P}^{\boldsymbol{\pi}, \infty} - \boldsymbol{P}^{\boldsymbol{\pi}'} + \boldsymbol{P}^{\boldsymbol{\pi}', \infty}\|_\infty \cdot \|(\boldsymbol{I} - \boldsymbol{P}^{\boldsymbol{\pi}'} + \boldsymbol{P}^{\boldsymbol{\pi}', \infty})^{-1}\|_\infty \\
&\overset{(b)}{\leq} 2C \|\boldsymbol{c}^{\boldsymbol{\pi}, \boldsymbol{p}} - \boldsymbol{c}^{\boldsymbol{\pi}', \boldsymbol{p}}\|_\infty + C \|\boldsymbol{d}^{\boldsymbol{\pi}, \boldsymbol{p}} - \boldsymbol{d}^{\boldsymbol{\pi}', \boldsymbol{p}}\|_1 + C^2 \|\boldsymbol{P}^{\boldsymbol{\pi}} - \boldsymbol{P}^{\boldsymbol{\pi}'}\|_\infty + C^2 \|\boldsymbol{d}^{\boldsymbol{\pi}, \boldsymbol{p}} - \boldsymbol{d}^{\boldsymbol{\pi}', \boldsymbol{p}}\|_1 \\
&\leq C(2 + C_d^{\boldsymbol{\pi}} S + CS + C_d^{\boldsymbol{\pi}} CS) \|\boldsymbol{\pi} - \boldsymbol{\pi}'\|_{1, \infty},
\end{aligned}
$$

where the inequality (a) is attained from the fact that $\|(\boldsymbol{I} - \boldsymbol{P}^{\boldsymbol{\pi}, \infty}) \boldsymbol{c}^{\boldsymbol{\pi}', \boldsymbol{p}}\|_\infty \leq 1$, and the inequality (b) is obtained due to

$$
\begin{aligned}
\|\boldsymbol{P}^{\boldsymbol{\pi}, \infty} - \boldsymbol{P}^{\boldsymbol{\pi}', \infty}\|_\infty &= \sum_s |d_s^{\boldsymbol{\pi}, \boldsymbol{p}} - d_s^{\boldsymbol{\pi}', \boldsymbol{p}}| = \|\boldsymbol{d}^{\boldsymbol{\pi}, \boldsymbol{p}} - \boldsymbol{d}^{\boldsymbol{\pi}', \boldsymbol{p}}\|_1, \\
\|\boldsymbol{P}^{\boldsymbol{\pi}} - \boldsymbol{P}^{\boldsymbol{\pi}'}\|_\infty &= \max_s \sum_{s'} |P_{ss'}^{\boldsymbol{\pi}} - P_{ss'}^{\boldsymbol{\pi}'}|.
\end{aligned}
$$

By applying the Bellman equation that $q_{sa}^{\boldsymbol{\pi}, \boldsymbol{p}}$ satisfies, we have for any $(s, a) \in \mathcal{S} \times \mathcal{A}$

$$
|q_{sa}^{\boldsymbol{\pi}, \boldsymbol{p}} - q_{sa}^{\boldsymbol{\pi}', \boldsymbol{p}}| = |J(\boldsymbol{\pi}, \boldsymbol{p}) - J(\boldsymbol{\pi}', \boldsymbol{p})| + \left| \sum_{s'} p_{sas'} (v_{s'}^{\boldsymbol{\pi}, \boldsymbol{p}} - v_{s'}^{\boldsymbol{\pi}', \boldsymbol{p}}) \right|,
$$

which leads to the following result:

$$\|\boldsymbol{q}_s^{\boldsymbol{\pi},\boldsymbol{p}} - \boldsymbol{q}_s^{\boldsymbol{\pi}',\boldsymbol{p}}\|_\infty \ \leq \ |J(\boldsymbol{\pi},\boldsymbol{p}) - J(\boldsymbol{\pi}',\boldsymbol{p})| + \|\boldsymbol{p}_{sa}\|_1 \cdot \|\boldsymbol{v}^{\boldsymbol{\pi},\boldsymbol{p}} - \boldsymbol{v}^{\boldsymbol{\pi}',\boldsymbol{p}}\|_\infty \ \leq \ (C_J^{\boldsymbol{\pi}} + C_v^{\boldsymbol{\pi}})\|\boldsymbol{\pi} - \boldsymbol{\pi}'\|_{1,\infty}$$

$\square$

*Proof of Lemma 3.2.* While the form of partial derivative over $\boldsymbol{\pi}$ has already been derived (Lemma B.5), we demonstrate that $J(\boldsymbol{\pi},\boldsymbol{p})$ is $L_{\boldsymbol{\pi}}$-Lipschitz in $\boldsymbol{\pi}$ by showing the boundedness of $\nabla_{\boldsymbol{\pi}}J(\boldsymbol{\pi},\boldsymbol{p})$, which has been shown as below

$$\|\nabla_{\boldsymbol{\pi}}J(\boldsymbol{\pi},\boldsymbol{p})\| \ = \ \sqrt{\sum_{s,a}\left(\frac{\partial J(\boldsymbol{\pi},\boldsymbol{p})}{\partial \pi_{sa}}\right)^2} \ = \ \sqrt{\sum_a \sum_s (d_s^{\boldsymbol{\pi},\boldsymbol{p}} q_{sa}^{\boldsymbol{\pi},\boldsymbol{p}})^2} \ \leq \ 7t_{\text{mix}}\sqrt{A},$$

where the last inequality is obtained from the facts $|q_{sa}^{\boldsymbol{\pi},\boldsymbol{p}}| \leq 7t_{\text{mix}}$ and $\sum_s (d^{\boldsymbol{\pi},\boldsymbol{p}}(s))^2 \leq 1$. We then turn to prove the smoothness condition of $J(\boldsymbol{\pi},\boldsymbol{p})$ with the help of perturbation theory of stochastic matrices. Let $\boldsymbol{\pi},\boldsymbol{\pi}' \in \Pi$ be any policies within the policy class. We introduce $\boldsymbol{\pi}(\alpha)$ as a convex combination of policies $\boldsymbol{\pi}$ and $\boldsymbol{\pi}'$, that is, $\boldsymbol{\pi}(\alpha) = (1-\alpha)\boldsymbol{\pi}+\alpha\boldsymbol{\pi}' := \boldsymbol{\pi} + \alpha\boldsymbol{u}$ where $\boldsymbol{u} = \boldsymbol{\pi}' - \boldsymbol{\pi}$. Notice that the partial derivative of $J(\boldsymbol{\pi}(\alpha),\boldsymbol{p})$ over $\alpha$ is

$$\frac{\partial J(\boldsymbol{\pi}(\alpha),\boldsymbol{p})}{\partial \alpha} \ = \ \sum_{s,a} d_s^{\boldsymbol{\pi}(\alpha),\boldsymbol{p}} u_{sa} q_{sa}^{\boldsymbol{\pi}(\alpha),\boldsymbol{p}},$$

then, we are going to show that $|\frac{\partial J(\boldsymbol{\pi}(\alpha),\boldsymbol{p})}{\partial \alpha} - \frac{\partial J(\boldsymbol{\pi},\boldsymbol{p})}{\partial \alpha}| \leq \alpha\ell_{\boldsymbol{\pi}}$, which is obtained as follows:

$$
\begin{aligned}
\left|\frac{\partial J(\boldsymbol{\pi}(\alpha),\boldsymbol{p})}{\partial \alpha} - \frac{\partial J(\boldsymbol{\pi},\boldsymbol{p})}{\partial \alpha}\right| &= \left|\sum_{s,a} d_s^{\boldsymbol{\pi}(\alpha),\boldsymbol{p}} u_{sa} q_{sa}^{\boldsymbol{\pi}(\alpha),\boldsymbol{p}} - \sum_{s,a} d_s^{\boldsymbol{\pi},\boldsymbol{p}} u_{sa} q_{sa}^{\boldsymbol{\pi},\boldsymbol{p}}\right| \\
&\leq \left|\sum_{s,a} d_s^{\boldsymbol{\pi}(\alpha),\boldsymbol{p}} u_{sa}(q_{sa}^{\boldsymbol{\pi}(\alpha),\boldsymbol{p}} - q_{sa}^{\boldsymbol{\pi},\boldsymbol{p}})\right| + \left|\sum_{s,a}(d_s^{\boldsymbol{\pi}(\alpha),\boldsymbol{p}} - d_s^{\boldsymbol{\pi},\boldsymbol{p}}) u_{sa} q_{sa}^{\boldsymbol{\pi},\boldsymbol{p}}\right| \\
&\leq \sum_s d_s^{\boldsymbol{\pi}(\alpha),\boldsymbol{p}}\left|\langle \boldsymbol{u}_s, \boldsymbol{q}_s^{\boldsymbol{\pi}(\alpha),\boldsymbol{p}} - \boldsymbol{q}_s^{\boldsymbol{\pi},\boldsymbol{p}}\rangle\right| + \sum_s \left|d_s^{\boldsymbol{\pi}(\alpha),\boldsymbol{p}} - d_s^{\boldsymbol{\pi},\boldsymbol{p}}\right| \cdot |\langle \boldsymbol{u}_s, \boldsymbol{q}_s^{\boldsymbol{\pi},\boldsymbol{p}}\rangle| \\
&\leq \sum_s d_s^{\boldsymbol{\pi}(\alpha),\boldsymbol{p}}\|\boldsymbol{q}_s^{\boldsymbol{\pi}(\alpha),\boldsymbol{p}} - \boldsymbol{q}_s^{\boldsymbol{\pi},\boldsymbol{p}}\|_\infty\|\boldsymbol{u}_s\|_1 + \sum_s \left|d_s^{\boldsymbol{\pi}(\alpha),\boldsymbol{p}} - d_s^{\boldsymbol{\pi},\boldsymbol{p}}\right|\|\boldsymbol{u}_s\|_1\|\boldsymbol{q}_s^{\boldsymbol{\pi},\boldsymbol{p}}\|_\infty \\
&\overset{(a)}{\leq} 2C_q^{\boldsymbol{\pi}}\|\alpha\boldsymbol{u}\|_{1,\infty} + 2\cdot 7t_{\text{mix}}C_d^{\boldsymbol{\pi}}S\|\alpha\boldsymbol{u}\|_{1,\infty} \\
&\leq (4C_q^{\boldsymbol{\pi}} + 28t_{\text{mix}}C_d^{\boldsymbol{\pi}}S)\alpha,
\end{aligned}
$$

where the inequality $(a)$ is obtained from the sensitivity of $\boldsymbol{q}_s^{\boldsymbol{\pi},\boldsymbol{p}}$ and $d_s^{\boldsymbol{\pi},\boldsymbol{p}}$ (Lemma 3.1), as well as the facts $|q_{sa}^{\boldsymbol{\pi},\boldsymbol{p}}| \leq 7t_{\text{mix}}$ and $\|\boldsymbol{u}_s\|_1 \leq 2$. Therefore, the smoothness is proved.

We next show the continuity of $\Psi(\boldsymbol{\pi})$. We first show $\Psi(\boldsymbol{\pi})$ is $L_{\boldsymbol{\pi}}$-Lipschitz if $J(\boldsymbol{\pi},\boldsymbol{p})$ is $L_{\boldsymbol{\pi}}$-Lipschitz in $\boldsymbol{\pi}$. Without loss of generality, we assume that for any $\boldsymbol{\pi}_1,\boldsymbol{\pi}_2 \in \Pi$, $\Psi(\boldsymbol{\pi}_1) \leq \Psi(\boldsymbol{\pi}_2)$ and $\boldsymbol{p}_1 := \arg\max_{\boldsymbol{p}\in\mathcal{P}} J(\boldsymbol{\pi}_1,\boldsymbol{p})$ and $\boldsymbol{p}_2 := \arg\max_{\boldsymbol{p}\in\mathcal{P}} J(\boldsymbol{\pi}_2,\boldsymbol{p})$, then we have

$$0 \ \leq \ \Psi(\boldsymbol{\pi}_1) - \Psi(\boldsymbol{\pi}_2) \ = \ J_{\boldsymbol{\rho}}(\boldsymbol{\pi}_1,\boldsymbol{p}_1) - J_{\boldsymbol{\rho}}(\boldsymbol{\pi}_2,\boldsymbol{p}_2) \ \leq \ J_{\boldsymbol{\rho}}(\boldsymbol{\pi}_1,\boldsymbol{p}_1) - J_{\boldsymbol{\rho}}(\boldsymbol{\pi}_2,\boldsymbol{p}_1) \ \leq \ L_{\boldsymbol{\pi}}\|\boldsymbol{\pi}_1 - \boldsymbol{\pi}_2\|.$$

Then, we notice that Lemma 3 in (Thekumparampil et al., 2019) verifies that $\Phi(\boldsymbol{\pi})$ is $\ell_{\boldsymbol{\pi}}$-weakly convex if $J(\boldsymbol{\pi},\boldsymbol{p})$ is $\ell_{\boldsymbol{\pi}}$-smooth, which can intuitively determine the weakly convexity of $\Psi(\boldsymbol{\pi})$. $\square$

Note that, similar smoothness conditions are derived in (Cheng et al., 2024; Kumar et al., 2024b), however, there are mistakes in the smoothness conditions derivation in (Cheng et al., 2024). Compared to the results in (Kumar et al., 2024b) we mentioned, we efficiently reduce the dependency on the state and action numbers.

Now, we turn to derive our main theorems. First of all, we prove the gradient dominance condition that our robust objective $\Psi(\boldsymbol{\pi})$ satisfies.

*Proof of Theorem 3.4.* We denote $\boldsymbol{\pi}^\star$ is the optimal policy for the robust AMDPs. We note that while $J(\boldsymbol{\pi}, \boldsymbol{p})$ is non-concave with respect to $\boldsymbol{p}$ and the ambiguity set $\mathcal{P}$ is assumed to be a compact set, it is possible to have multiple $N$ inner maximum points. For simplicity of analysis, we consider the case $N = 1$, and refer interested reader to Theorem 3.2 in (Wang et al., 2023c) for more detailed discussion about the similar general case. Specifically, we denote $\boldsymbol{p}^{\boldsymbol{\pi}} := \arg\max_{\boldsymbol{p} \in \mathcal{P}} J(\boldsymbol{\pi}, \boldsymbol{p})$ as the worst-case transition kernel for fixed policy $\boldsymbol{\pi} \in \Pi$. By utilizing the gradient domination condition established for nonrobust AMDPs (Lemma B.6), we can derive the following inequality:

$$\Psi(\boldsymbol{\pi}) - \Psi(\boldsymbol{\pi}^\star) \ \leq \ J(\boldsymbol{\pi}, \boldsymbol{p}^{\boldsymbol{\pi}}) - \min_{\boldsymbol{\pi} \in \Pi} J(\boldsymbol{\pi}, \boldsymbol{p}^{\boldsymbol{\pi}}) \ \leq \ M \cdot \max_{\bar{\boldsymbol{\pi}} \in \Pi} \langle \boldsymbol{\pi} - \bar{\boldsymbol{\pi}}, \nabla_{\boldsymbol{\pi}} J(\boldsymbol{\pi}, \boldsymbol{p}^{\boldsymbol{\pi}}) \rangle. \tag{16}$$

Notice that, by applying Lemma B.10, $J_\rho(\boldsymbol{\pi}, \boldsymbol{p})$ is $\ell_{\boldsymbol{\pi}}$-weakly convex in $\boldsymbol{\pi}$, which leads to the fact that $\tilde{J}(\boldsymbol{\pi}, \boldsymbol{p}) := J(\boldsymbol{\pi}, \boldsymbol{p}) + \frac{\ell_{\boldsymbol{\pi}}}{2}\|\boldsymbol{\pi}\|^2$ is convex in $\boldsymbol{\pi}$ (Kruger, 2003). Let $\tilde{\Psi}(\boldsymbol{\pi}) := \max_{\boldsymbol{p} \in \mathcal{P}} \tilde{J}(\boldsymbol{\pi}, \boldsymbol{p})$. By leveraging the convexity of $\tilde{J}_\rho(\boldsymbol{\pi}, \boldsymbol{p})$ and the compactness of $\mathcal{P}$, we can apply Danskin's Theorem (Proposition B.17) to attain

$$
\begin{aligned}
\partial \tilde{\Psi}(\boldsymbol{\pi}) &= \nabla_{\boldsymbol{\pi}} \tilde{J}(\boldsymbol{\pi}, \boldsymbol{p}^{\boldsymbol{\pi}}) \\
\implies \quad \partial \Psi(\boldsymbol{\pi}) + \ell_{\boldsymbol{\pi}} \boldsymbol{\pi} &= \nabla_{\boldsymbol{\pi}} J(\boldsymbol{\pi}, \boldsymbol{p}^{\boldsymbol{\pi}}) + \ell_{\boldsymbol{\pi}} \boldsymbol{\pi} \\
\implies \quad \partial \Psi(\boldsymbol{\pi}) &= \nabla_{\boldsymbol{\pi}} J(\boldsymbol{\pi}, \boldsymbol{p}^{\boldsymbol{\pi}}),
\end{aligned}
$$

which also implies that $\boldsymbol{\xi} = \partial \Psi(\boldsymbol{\pi}) = \nabla_{\boldsymbol{\pi}} J(\boldsymbol{\pi}, \boldsymbol{p}^{\boldsymbol{\pi}})$. By introducing $\tilde{\boldsymbol{\pi}} = \arg\min_{\boldsymbol{y} \in \Pi} \Psi(\boldsymbol{y}) + \ell_{\boldsymbol{\pi}}\|\boldsymbol{\pi} - \boldsymbol{y}\|^2$, Lemma B.16 implies that there exists $\tilde{\boldsymbol{\xi}} = \partial \Psi(\tilde{\boldsymbol{\pi}})$ such that $-\tilde{\boldsymbol{\xi}} \subseteq \mathcal{N}_\mathcal{X}(\tilde{\boldsymbol{\pi}}) + 2\ell_{\boldsymbol{\pi}}\|\tilde{\boldsymbol{\pi}} - \boldsymbol{\pi}\| \cdot \mathcal{B}(1)$. Then, we have

$$\Psi(\tilde{\boldsymbol{\pi}}) - \Psi(\boldsymbol{\pi}^\star) \ \leq \ M \cdot \max_{\bar{\boldsymbol{\pi}} \in \Pi} \langle \tilde{\boldsymbol{\pi}} - \bar{\boldsymbol{\pi}}, \nabla_{\boldsymbol{\pi}} J(\tilde{\boldsymbol{\pi}}, \tilde{\boldsymbol{p}}^{\tilde{\boldsymbol{\pi}}}) \rangle \ \leq \ M \cdot \max_{\bar{\boldsymbol{\pi}} \in \Pi} \langle \bar{\boldsymbol{\pi}} - \tilde{\boldsymbol{\pi}}, -\nabla_{\boldsymbol{\pi}} J(\tilde{\boldsymbol{\pi}}, \tilde{\boldsymbol{p}}^{\tilde{\boldsymbol{\pi}}}) \rangle. \tag{17}$$

Notice that for any $\boldsymbol{\pi}_1, \boldsymbol{\pi}_2 \in \Pi$, we have $-\boldsymbol{e} \leq \boldsymbol{\pi}_1 - \boldsymbol{\pi}_2 \leq \boldsymbol{e}$ where $\boldsymbol{e}$ is all-one vector. Then, we introduce a adaptive all-one vector $\hat{\boldsymbol{e}}$, whose $i$-th element $\hat{e}_i = 1$ while the corresponding element of $-\nabla_{\boldsymbol{\pi}} J(\tilde{\boldsymbol{\pi}}, \tilde{\boldsymbol{p}}^{\boldsymbol{\pi}})$ is 1 and $\hat{e}_i = -1$ while the corresponding element of $-\nabla_{\boldsymbol{\pi}} J(\tilde{\boldsymbol{\pi}}, \tilde{\boldsymbol{p}}^{\tilde{\boldsymbol{\pi}}})$ is $-1$. Therefore, we have

$$(17) \ \leq \ M \cdot \langle \hat{\boldsymbol{e}}, -\nabla_{\boldsymbol{\pi}} J(\tilde{\boldsymbol{\pi}}, \tilde{\boldsymbol{p}}^{\tilde{\boldsymbol{\pi}}}) \rangle \ = \ M \cdot \langle \hat{\boldsymbol{e}}, -\tilde{\boldsymbol{\xi}} \rangle \ \leq \ M\sqrt{SA} \cdot \|\nabla \Psi_{1/2\ell_{\boldsymbol{\pi}}}(\boldsymbol{\pi})\|. \tag{18}$$

The final inequality can be derived from the result in Lemma B.16. It is worth noting that Lemma 3.2 implies the $L_{\boldsymbol{\pi}}$-Lipschitz continuity of $\Psi(\boldsymbol{\pi})$. By leveraging this Lipschitz property in conjunction with the aforementioned equation (18), we derive the desired result

$$
\begin{aligned}
\Psi(\boldsymbol{\pi}) - \Psi(\boldsymbol{\pi}^\star) &= \Psi(\boldsymbol{\pi}) - \Psi(\tilde{\boldsymbol{\pi}}) + \Psi(\tilde{\boldsymbol{\pi}}) - \Psi(\boldsymbol{\pi}^\star) \\
&\leq M\sqrt{SA}\|\nabla \Psi_{1/2\ell_{\boldsymbol{\pi}}}(\boldsymbol{\pi})\| + \Psi(\boldsymbol{\pi}) - \Psi(\tilde{\boldsymbol{\pi}}) \\
&\leq M\sqrt{SA}\|\nabla \Psi_{1/2\ell_{\boldsymbol{\pi}}}(\boldsymbol{\pi})\| + L_{\boldsymbol{\pi}}\|\boldsymbol{\pi} - \tilde{\boldsymbol{\pi}}\| \\
&= \left(M\sqrt{SA} + \frac{L_{\boldsymbol{\pi}}}{2\ell_{\boldsymbol{\pi}}}\right) \cdot \|\nabla \Psi_{1/2\ell_{\boldsymbol{\pi}}}(\boldsymbol{\pi})\|,
\end{aligned}
\tag{19}
$$

where (19) holds by using arguments of Lemma B.16 and $\Psi(\boldsymbol{\pi}) \geq \Psi(\tilde{\boldsymbol{\pi}})$. $\qquad\square$

*Proof of Theorem 3.5.* We begin by defining a policy $\tilde{\boldsymbol{\pi}}_t = \arg\min_{\tilde{\boldsymbol{\pi}} \in \Pi} \Psi(\tilde{\boldsymbol{\pi}}) + \ell_{\boldsymbol{\pi}}\|\boldsymbol{\pi}_t - \tilde{\boldsymbol{\pi}}\|^2$, then, we have

$$\Psi_{1/2\ell_{\boldsymbol{\pi}}}(\boldsymbol{\pi}_{t+1}) \ = \ \min_{\boldsymbol{\pi}} \left(\Psi(\boldsymbol{\pi}) + \ell_{\boldsymbol{\pi}}\|\boldsymbol{\pi}_{t+1} - \boldsymbol{\pi}\|^2\right) \ \leq \ \Psi(\tilde{\boldsymbol{\pi}}_t) + \ell_{\boldsymbol{\pi}}\|\boldsymbol{\pi}_{t+1} - \tilde{\boldsymbol{\pi}}_t\|^2.$$

The proposed RP2G updates the policy by using the projected gradient descent step:

$$\boldsymbol{\pi}_{t+1} \ = \ \text{Proj}_\Pi \left(\boldsymbol{\pi}_t - \alpha_t \nabla_{\boldsymbol{\pi}} J(\boldsymbol{\pi}_t, \boldsymbol{p}_t)\right).$$

Therefore, the Moreau envelope function $\Psi_{1/2\ell_{\boldsymbol{\pi}}}(\boldsymbol{\pi}_{t+1})$ satisfies

$$
\begin{aligned}
\Psi_{1/2\ell_{\boldsymbol{\pi}}}(\boldsymbol{\pi}_{t+1}) &\leq \Psi(\tilde{\boldsymbol{\pi}}_t) + \ell_{\boldsymbol{\pi}}\|\text{Proj}_\Pi(\boldsymbol{\pi}_t - \alpha \nabla_{\boldsymbol{\pi}} J(\boldsymbol{\pi}_t, \boldsymbol{p}_t)) - \text{Proj}_\Pi(\tilde{\boldsymbol{\pi}}_t)\|^2 \\
&\overset{(a)}{\leq} \Psi(\tilde{\boldsymbol{\pi}}_t) + \ell_{\boldsymbol{\pi}}\|\boldsymbol{\pi}_t - \alpha \nabla_{\boldsymbol{\pi}} J(\boldsymbol{\pi}_t, \boldsymbol{p}_t) - \tilde{\boldsymbol{\pi}}_t\|^2 \\
&= \Psi(\tilde{\boldsymbol{\pi}}_t) + \ell_{\boldsymbol{\pi}}\|\boldsymbol{\pi}_t - \tilde{\boldsymbol{\pi}}_t\|^2 - 2\ell_{\boldsymbol{\pi}}\alpha \langle \nabla_{\boldsymbol{\pi}} J(\boldsymbol{\pi}_t, \boldsymbol{p}_t), \boldsymbol{\pi}_t - \tilde{\boldsymbol{\pi}}_t \rangle + \alpha^2 \ell_{\boldsymbol{\pi}}\|\nabla_{\boldsymbol{\pi}} J(\boldsymbol{\pi}_t, \boldsymbol{p}_t)\|^2 \\
&\leq \Psi_{1/2\ell_{\boldsymbol{\pi}}}(\boldsymbol{\pi}_t) + 2\ell_{\boldsymbol{\pi}}\alpha \left(\Psi(\tilde{\boldsymbol{\pi}}_t) - \Psi(\boldsymbol{\pi}_t) + \delta_t + \frac{\ell_{\boldsymbol{\pi}}}{2}\|\boldsymbol{\pi}_t - \tilde{\boldsymbol{\pi}}_t\|^2\right) + \alpha^2 \ell_{\boldsymbol{\pi}} L_{\boldsymbol{\pi}}^2.
\end{aligned}
\tag{20}
$$

Here, $(\boldsymbol{\pi}_t, \boldsymbol{p}_t)$ is produced by the RP2G scheme at iteration step $t$. The inequality $(a)$ follows the basic projection property (Rockafellar, 1976), *i.e.*, for any $\boldsymbol{x}_1, \boldsymbol{x}_2 \in \mathbb{R}^n$,

$$\|\text{Proj}_{\mathcal{X}}(\boldsymbol{x}_1) - \text{Proj}_{\mathcal{X}}(\boldsymbol{x}_2)\| \leq \|\boldsymbol{x}_1 - \boldsymbol{x}_2\|,$$

and the last inequality holds due to the fact that $J(\boldsymbol{\pi}, \boldsymbol{p})$ is $\ell_{\boldsymbol{\pi}}$-smooth in $\boldsymbol{\pi}$, in the sense that, for $\tilde{\boldsymbol{\pi}}_t$,

$$\Psi(\tilde{\boldsymbol{\pi}}_t) \geq J(\tilde{\boldsymbol{\pi}}_t, \boldsymbol{p}_t) \geq J(\boldsymbol{\pi}_t, \boldsymbol{p}_t) + \langle \nabla_{\boldsymbol{\pi}} J(\boldsymbol{\pi}_t, \boldsymbol{p}_t), \tilde{\boldsymbol{\pi}}_t - \boldsymbol{\pi}_t \rangle - \frac{\ell_{\boldsymbol{\pi}}}{2} \|\tilde{\boldsymbol{\pi}}_t - \boldsymbol{\pi}_t\|^2$$

$$\geq \underbrace{\max_{\boldsymbol{p} \in \mathcal{P}} J(\boldsymbol{\pi}_t, \boldsymbol{p})}_{\Psi(\boldsymbol{\pi}_t)} - \delta_t + \langle \nabla_{\boldsymbol{\pi}} J(\boldsymbol{\pi}_t, \boldsymbol{p}_t), \tilde{\boldsymbol{\pi}}_t - \boldsymbol{\pi}_t \rangle - \frac{\ell_{\boldsymbol{\pi}}}{2} \|\tilde{\boldsymbol{\pi}}_t - \boldsymbol{\pi}_t\|^2.$$

By summing (20) up over $t$, we deduce that,

$$\Psi_{1/2\ell_{\boldsymbol{\pi}}}(\boldsymbol{\pi}_{T-1}) \leq \Psi_{1/2\ell_{\boldsymbol{\pi}}}(\boldsymbol{\pi}_0) + 2\ell_{\boldsymbol{\pi}}\alpha \sum_{t=0}^{T-1} \left( \Psi(\tilde{\boldsymbol{\pi}}_t) - \Psi(\boldsymbol{\pi}_t) + \delta_t + \frac{\ell_{\boldsymbol{\pi}}}{2} \|\tilde{\boldsymbol{\pi}}_t - \boldsymbol{\pi}_t\|^2 \right) + T\alpha^2 \ell_{\boldsymbol{\pi}} L_{\boldsymbol{\pi}}^2.$$

Rearranging the above inequality yields

$$\sum_{t=0}^{T-1} \left( \Psi(\boldsymbol{\pi}_t) - \Psi(\tilde{\boldsymbol{\pi}}_t) - \frac{\ell_{\boldsymbol{\pi}}}{2} \|\tilde{\boldsymbol{\pi}}_t - \boldsymbol{\pi}_t\|^2 \right) \leq \frac{\Psi_{1/2\ell_{\boldsymbol{\pi}}}(\boldsymbol{\pi}_0) - \Psi_{1/2\ell_{\boldsymbol{\pi}}}(\boldsymbol{\pi}_{T-1})}{2\ell_{\boldsymbol{\pi}}\alpha} + \frac{T\alpha L_{\boldsymbol{\pi}}^2}{2} + \sum_{t=0}^{T-1} \delta_t. \tag{21}$$

It is worth noting that,

$$\Psi(\boldsymbol{\pi}_t) - \Psi(\tilde{\boldsymbol{\pi}}_t) - \frac{\ell_{\boldsymbol{\pi}}}{2} \|\tilde{\boldsymbol{\pi}}_t - \boldsymbol{\pi}_t\|^2 = \Psi(\boldsymbol{\pi}_t) + \ell_{\boldsymbol{\pi}} \|\boldsymbol{\pi}_t - \boldsymbol{\pi}_t\|^2 - \Psi(\tilde{\boldsymbol{\pi}}_t) - \ell_{\boldsymbol{\pi}} \|\tilde{\boldsymbol{\pi}}_t - \boldsymbol{\pi}_t\|^2 + \frac{\ell_{\boldsymbol{\pi}}}{2} \|\tilde{\boldsymbol{\pi}}_t - \boldsymbol{\pi}_t\|^2$$

$$= \Psi(\boldsymbol{\pi}_t) + \ell_{\boldsymbol{\pi}} \|\boldsymbol{\pi}_t - \boldsymbol{\pi}_t\|^2 - \min_{\boldsymbol{\pi} \in \Pi} \left( \Psi(\boldsymbol{\pi}) + \ell_{\boldsymbol{\pi}} \|\boldsymbol{\pi}_t - \boldsymbol{\pi}\|^2 \right) + \frac{\ell_{\boldsymbol{\pi}}}{2} \|\tilde{\boldsymbol{\pi}}_t - \boldsymbol{\pi}_t\|^2$$

$$\overset{(a)}{\geq} \ell_{\boldsymbol{\pi}} \|\boldsymbol{\pi}_t - \tilde{\boldsymbol{\pi}}_t\|^2 = \frac{1}{4\ell_{\boldsymbol{\pi}}} \left\| \nabla \Psi_{1/2\ell_{\boldsymbol{\pi}}}(\boldsymbol{\pi}_t) \right\|^2, \tag{22}$$

where the inequality $(a)$ in (22) is obtained due to the strong convexity of $\Psi(\boldsymbol{\pi}) + \ell_{\boldsymbol{\pi}} \|\boldsymbol{\pi}_t - \boldsymbol{\pi}\|^2$, for example see Lemma E.3 in (Wang et al., 2023c). The last equality in (22) is obtained by directly utilizing the gradient of Moreau envelope function proposed in Lemma B.16, *i.e.*,

$$\nabla \Psi_{1/2\ell_{\boldsymbol{\pi}}}(\boldsymbol{\pi}_t) = 2\ell_{\boldsymbol{\pi}} \left( \boldsymbol{\pi}_t - \arg\max_{\boldsymbol{\pi} \in \Pi} \left( \Psi(\boldsymbol{\pi}) + \ell_{\boldsymbol{\pi}} \|\boldsymbol{\pi}_t - \boldsymbol{\pi}\|^2 \right) \right) = 2\ell_{\boldsymbol{\pi}} \left( \boldsymbol{\pi}_t - \tilde{\boldsymbol{\pi}}_t \right).$$

Let us introduce $\bar{\boldsymbol{\pi}}_1 := \arg\min_{\bar{\boldsymbol{\pi}} \in \Pi} \Psi(\bar{\boldsymbol{\pi}}) + \ell_{\boldsymbol{\pi}} \|\boldsymbol{\pi}_1 - \bar{\boldsymbol{\pi}}\|^2$ and $\bar{\boldsymbol{\pi}}_2 := \arg\min_{\bar{\boldsymbol{\pi}} \in \Pi} \Psi(\bar{\boldsymbol{\pi}}) + \ell_{\boldsymbol{\pi}} \|\boldsymbol{\pi}_2 - \bar{\boldsymbol{\pi}}\|^2$ for any $\boldsymbol{\pi}_1, \boldsymbol{\pi}_2 \in \Pi$, and then we have

$$\Psi_{1/2\ell_{\boldsymbol{\pi}}}(\boldsymbol{\pi}_1) - \Psi_{1/2\ell_{\boldsymbol{\pi}}}(\boldsymbol{\pi}_2) = \min_{\bar{\boldsymbol{\pi}} \in \Pi} \left( \Psi(\bar{\boldsymbol{\pi}}) + \ell_{\boldsymbol{\pi}} \|\boldsymbol{\pi}_1 - \bar{\boldsymbol{\pi}}\|^2 \right) - \min_{\bar{\boldsymbol{\pi}} \in \Pi} \left( \Psi(\bar{\boldsymbol{\pi}}) + \ell_{\boldsymbol{\pi}} \|\boldsymbol{\pi}_2 - \bar{\boldsymbol{\pi}}\|^2 \right)$$

$$= \Psi(\bar{\boldsymbol{\pi}}_1) + \ell_{\boldsymbol{\pi}} \|\boldsymbol{\pi}_1 - \bar{\boldsymbol{\pi}}_1\|^2 - \Psi(\bar{\boldsymbol{\pi}}_2) - \ell_{\boldsymbol{\pi}} \|\boldsymbol{\pi}_2 - \bar{\boldsymbol{\pi}}_2\|^2$$

$$\leq \Psi(\bar{\boldsymbol{\pi}}_2) + \ell_{\boldsymbol{\pi}} \|\boldsymbol{\pi}_1 - \bar{\boldsymbol{\pi}}_2\|^2 - \Psi(\bar{\boldsymbol{\pi}}_2) - \ell_{\boldsymbol{\pi}} \|\boldsymbol{\pi}_2 - \bar{\boldsymbol{\pi}}_2\|^2$$

$$= \ell_{\boldsymbol{\pi}} \left( \|\boldsymbol{\pi}_1 - \bar{\boldsymbol{\pi}}_2\|^2 - \|\boldsymbol{\pi}_2 - \bar{\boldsymbol{\pi}}_2\|^2 \right)$$

$$\leq 2\ell_{\boldsymbol{\pi}} S. \tag{23}$$

Therefore, we obtain an upper bound such that

$$\Psi_{1/2\ell_{\boldsymbol{\pi}}}(\boldsymbol{\pi}_0) - \Psi_{1/2\ell_{\boldsymbol{\pi}}}(\boldsymbol{\pi}_{T-1}) \leq 2\ell_{\boldsymbol{\pi}} S.$$

Plug (23) and (22) into (21) and then we obtain that

$$\sum_{t=0}^{T-1} \left\| \nabla \Psi_{1/2\ell_{\boldsymbol{\pi}}}(\boldsymbol{\pi}_t) \right\|^2 \leq \frac{4\ell_{\boldsymbol{\pi}} S}{\alpha} + 2T\alpha \ell_{\boldsymbol{\pi}} L_{\boldsymbol{\pi}}^2 + 4\ell_{\boldsymbol{\pi}} \sum_{t=0}^{T-1} \delta_t.$$

We next show that for some tolerance $\epsilon > 0$, there exists some $t$ such that

$$\Psi(\boldsymbol{\pi}_t) - \min_{\boldsymbol{\pi} \in \Pi} \Psi(\boldsymbol{\pi}) \leq \epsilon.$$

We define the globally optimal policy for the RMDP as $\boldsymbol{\pi}^\star$. By applying the result stated in Theorem 3.4, we have

$$\Psi(\boldsymbol{\pi}_t) - \Psi(\boldsymbol{\pi}^\star) \leq \left( M\sqrt{SA} + \frac{L_{\boldsymbol{\pi}}}{2\ell_{\boldsymbol{\pi}}} \right) \cdot \left\| \nabla \Psi_{1/2\ell_{\boldsymbol{\pi}}}(\boldsymbol{\pi}_t) \right\|, \tag{24}$$

By summing up (24) over $t$ and lower-bounding it, we can observe that

$$\min_{t \in \{0, \cdots, T-1\}} \{\Psi(\boldsymbol{\pi}_t) - \Psi(\boldsymbol{\pi}^\star)\} \leq \frac{1}{T} \sum_{t=0}^{T-1} (\Psi(\boldsymbol{\pi}_t) - \Psi(\boldsymbol{\pi}^\star)) \leq \frac{1}{T} \left( M\sqrt{SA} + \frac{L_{\boldsymbol{\pi}}}{2\ell_{\boldsymbol{\pi}}} \right) \sum_{t=0}^{T-1} \left\| \nabla \Psi_{1/2\ell_{\boldsymbol{\pi}}}(\boldsymbol{\pi}_t) \right\|.$$

By Cauchy–Schwarz inequality, we can obtain

$$\frac{1}{\sqrt{T}} \sum_{t=0}^{T-1} \left\| \nabla \Psi_{1/2\ell_{\boldsymbol{\pi}}}(\boldsymbol{\pi}_t) \right\| \leq \sqrt{\sum_{t=0}^{T-1} \left\| \nabla \Psi_{1/2\ell_{\boldsymbol{\pi}}}(\boldsymbol{\pi}_t) \right\|^2}.$$

Set $\alpha := \frac{1}{\sqrt{T}}$, $\delta_0 \leq \sqrt{T}$, $\delta_{t+1} \leq \tau \delta_t$ and $\boldsymbol{\pi}_{t^\star}$ as the output of Algorithm 1, and then we obtain

$$
\begin{aligned}
\Psi(\boldsymbol{\pi}_{t^\star}) - \Psi(\boldsymbol{\pi}^\star) &\leq \frac{1}{\sqrt{T}} \left( M\sqrt{SA} + \frac{L_{\boldsymbol{\pi}}}{2\ell_{\boldsymbol{\pi}}} \right) \sqrt{\sum_{t=0}^{T-1} \left\| \nabla \Psi_{1/2\ell_{\boldsymbol{\pi}}}(\boldsymbol{\pi}_t) \right\|^2} \\
&= \frac{1}{\sqrt{T}} \left( M\sqrt{SA} + \frac{L_{\boldsymbol{\pi}}}{2\ell_{\boldsymbol{\pi}}} \right) \sqrt{\left( \frac{4\ell_{\boldsymbol{\pi}}S}{\alpha} + 2T\alpha\ell_{\boldsymbol{\pi}}L_{\boldsymbol{\pi}}^2 + 4\ell_{\boldsymbol{\pi}} \sum_{t=0}^{T-1} \delta_t \right)} \\
&\overset{(a)}{\leq} \frac{1}{\sqrt{T}} \left( M\sqrt{SA} + \frac{L_{\boldsymbol{\pi}}}{2\ell_{\boldsymbol{\pi}}} \right) \sqrt{\left( 4\ell_{\boldsymbol{\pi}}S\sqrt{T} + 2\sqrt{T}\ell_{\boldsymbol{\pi}}L_{\boldsymbol{\pi}}^2 + \frac{4\ell_{\boldsymbol{\pi}}\delta_0}{1-\tau} \right)} \\
&\leq \frac{1}{\sqrt{T}} \left( M\sqrt{SA} + \frac{L_{\boldsymbol{\pi}}}{2\ell_{\boldsymbol{\pi}}} \right) \sqrt{\left( 4\ell_{\boldsymbol{\pi}}S\sqrt{T} + 2\sqrt{T}\ell_{\boldsymbol{\pi}}L_{\boldsymbol{\pi}}^2 + \frac{4\ell_{\boldsymbol{\pi}}\sqrt{T}}{1-\tau} \right)},
\end{aligned}
$$

where the inequality $(a)$ holds due to the adaptive tolerance sequence, in the sense that,

$$\sum_{t=0}^{T-1} \delta_t \leq \sum_{t=0}^{\infty} \delta_t \leq \delta_0 \cdot \left( 1 + \tau + \tau^2 + \cdots \right) \leq \frac{\delta_0}{1-\tau}.$$

Now we can attain our final result that, when $T$ satisfies the following condition,

$$T \geq \frac{\left( M\sqrt{SA} + \frac{L_{\boldsymbol{\pi}}}{2\ell_{\boldsymbol{\pi}}} \right)^4 \left( 4\ell_{\boldsymbol{\pi}}S + 2\ell_{\boldsymbol{\pi}}L_{\boldsymbol{\pi}}^2 + \frac{4\ell_{\boldsymbol{\pi}}}{1-\tau} \right)^2}{\epsilon^4} = \mathcal{O}(\epsilon^{-4}),$$

then, we have

$$\Psi(\boldsymbol{\pi}_{t^\star}) - \min_{\boldsymbol{\pi} \in \Pi} \Psi(\boldsymbol{\pi}) \leq \epsilon,$$

$\square$

## D. Omitted Proofs in Section 4 and Relative Supporting Results

*Proof of Lemma 4.1.* For any $s \in \mathcal{S}$, we can formulate the gradient of the differential value function as

$$
\begin{aligned}
\frac{\partial v_s^{\boldsymbol{\pi},\boldsymbol{p}}}{\partial p_{s_1 a_1 s_2}} &= \sum_a \pi_{sa} \frac{\partial q_{sa}^{\boldsymbol{\pi},\boldsymbol{p}}}{\partial p_{s_1 a_1 s_2}} \\
&= \sum_a \pi_{sa} \frac{\partial}{\partial p_{s_1 a_1 s_2}} \left( \sum_{s'} p_{sas'} \left( c_{sas'} - J(\boldsymbol{\pi},\boldsymbol{p}) + v_{s'}^{\boldsymbol{\pi},\boldsymbol{p}} \right) \right) \\
&= \sum_a \pi_{sa} \sum_{s'} \frac{\partial p_{sas'}}{\partial p_{s_1 a_1 s_2}} \left( c_{sas'} - J(\boldsymbol{\pi},\boldsymbol{p}) + v_{s'}^{\boldsymbol{\pi},\boldsymbol{p}} \right) + \sum_a \pi_{sa} \sum_{s'} p_{sas'} \left( -\frac{\partial J(\boldsymbol{\pi},\boldsymbol{p})}{\partial p_{s_1 a_1 s_2}} + \frac{\partial v_{s'}^{\boldsymbol{\pi},\boldsymbol{p}}}{\partial p_{s_1 a_1 s_2}} \right).
\end{aligned}
$$

Multiplying each side with $d_s^{\boldsymbol{\pi},\boldsymbol{p}^{\boldsymbol{\xi}}}$, taking the summation over $s \in \mathcal{S}$, and rearranging terms, we then obtain

$$
\begin{aligned}
\frac{\partial J(\boldsymbol{\pi},\boldsymbol{p})}{\partial p_{s_1 a_1 s_2}} &= \sum_s d_s^{\boldsymbol{\pi},\boldsymbol{p}} \left( \sum_a \pi_{sa} \sum_{s'} \frac{\partial p_{sas'}}{\partial p_{s_1 a_1 s_2}} \left( c_{sas'} - J(\boldsymbol{\pi},\boldsymbol{p}) + v_{s'}^{\boldsymbol{\pi},\boldsymbol{p}} \right) + \sum_a \pi_{sa} \sum_{s'} p_{sas'} \frac{\partial v_{s'}^{\boldsymbol{\pi},\boldsymbol{p}}}{\partial p_{s_1 a_1 s_2}} - \frac{\partial v_s^{\boldsymbol{\pi},\boldsymbol{p}}}{\partial p_{s_1 a_1 s_2}} \right) \\
&= \sum_s d_s^{\boldsymbol{\pi},\boldsymbol{p}} \sum_a \pi_{sa} \sum_{s'} \frac{\partial p_{sas'}}{\partial p_{s_1 a_1 s_2}} \left( c_{sas'} - J(\boldsymbol{\pi},\boldsymbol{p}) + v_{s'}^{\boldsymbol{\pi},\boldsymbol{p}} \right) \\
&= d_{s_1}^{\boldsymbol{\pi},\boldsymbol{p}} \pi_{s_1 a_1} \left( c_{s_1 a_1 s_2} - J(\boldsymbol{\pi},\boldsymbol{p}) + v_{s_2}^{\boldsymbol{\pi},\boldsymbol{p}} \right)
\end{aligned}
$$

$\square$

Then, we show the corresponding performance difference lemma that the adversary satisfies.

**Lemma D.1.** *(Adversary's Performance Difference Lemma) For any policy $\boldsymbol{\pi} \in \Pi$ and $\boldsymbol{p}, \boldsymbol{p}' \in (\Delta^S)^{S \times A}$, we have*

$$
J(\boldsymbol{\pi},\boldsymbol{p}) - J(\boldsymbol{\pi},\boldsymbol{p}') = \sum_{s \in \mathcal{S}} d_s^{\boldsymbol{\pi},\boldsymbol{p}} \sum_{a \in \mathcal{A}} \pi_{sa} \sum_{s'} (p_{sas'} - p'_{sas'}) \cdot g_{sas'}^{\boldsymbol{\pi},\boldsymbol{p}'}.
$$

*Proof of Lemma D.1.* By the definition of the differential action-next-state value function, we have

$$
\begin{aligned}
\sum_s d_s^{\boldsymbol{\pi},\boldsymbol{p}} \sum_a \pi_{sa} \sum_{s'} p_{sas'} g_{sas'}^{\boldsymbol{\pi},\boldsymbol{p}'} &= \sum_s d_s^{\boldsymbol{\pi},\boldsymbol{p}} \sum_a \pi_{sa} \sum_{s'} p_{sas'} \left( c_{sas'} - J(\boldsymbol{\pi},\boldsymbol{p}') + v_{s'}^{\boldsymbol{\pi},\boldsymbol{p}'} \right) \\
&= J(\boldsymbol{\pi},\boldsymbol{p}) - J(\boldsymbol{\pi},\boldsymbol{p}') + \sum_s d_s^{\boldsymbol{\pi},\boldsymbol{p}} \sum_a \pi_{sa} \sum_{s'} p_{sas'} v_{s'}^{\boldsymbol{\pi},\boldsymbol{p}'} \\
&= J(\boldsymbol{\pi},\boldsymbol{p}) - J(\boldsymbol{\pi},\boldsymbol{p}') + \sum_{s'} d_{s'}^{\boldsymbol{\pi},\boldsymbol{p}} v_{s'}^{\boldsymbol{\pi},\boldsymbol{p}'} \\
&= J(\boldsymbol{\pi},\boldsymbol{p}) - J(\boldsymbol{\pi},\boldsymbol{p}') + \sum_s d_s^{\boldsymbol{\pi},\boldsymbol{p}} \sum_a \pi_{sa} \sum_{s'} p'_{sas'} g_{sas'}^{\boldsymbol{\pi},\boldsymbol{p}'},
\end{aligned}
$$

which leads to the desired result. $\square$

*Proof of Lemma 4.2.* We first follow the similar strategy of Lemma 3.1 to propose the sensitivity analysis on the transition-induced cost and the state transition probability as follows, that is, for any $s \in \mathcal{S}$

$$
|c_s^{\boldsymbol{\pi},\boldsymbol{p}_1} - c_s^{\boldsymbol{\pi},\boldsymbol{p}_2}| = \left| \sum_a \pi_{sa} \sum_{s'} (p_{1,sas'} - p_{2,sas'}) c_{sas'} \right| \leq \sum_a \pi_{sa} \max_{s,a} \sum_{s'} |p_{1,sas'} - p_{2,sas'}| = \|\boldsymbol{p}_1 - \boldsymbol{p}_2\|_{1,\infty},
$$

$$
|\boldsymbol{P}_{1,s}^{\boldsymbol{\pi}} - \boldsymbol{P}_{2,s}^{\boldsymbol{\pi}}| = \sum_{s'} |P_{1,ss'}^{\boldsymbol{\pi}} - P_{2,ss'}^{\boldsymbol{\pi}}| \leq \sum_a \pi_{sa} \sum_{s'} |p_{1,sas'} - p_{2,sas'}| \leq \|\boldsymbol{p}_1 - \boldsymbol{p}_2\|_{1,\infty}.
$$

Then, by utilizing the adversial policy performance difference lemma (Lemma D.1), we can reach our first two desire results. For the average-reward objective, we have

$$
\begin{aligned}
|J(\boldsymbol{\pi}, \boldsymbol{p}_1) - J(\boldsymbol{\pi}, \boldsymbol{p})_2| &= \left| \sum_{s \in \mathcal{S}} d_s^{\boldsymbol{\pi}, \boldsymbol{p}_1} \sum_{a \in \mathcal{A}} \pi_{sa} \sum_{s'} (p_{1,sas'} - p_{2,sas'}) g_{sas'}^{\boldsymbol{\pi}, \boldsymbol{p}_2} \right| \\
&\overset{(a)}{\leq} (2 + 5t_{\mathrm{mix}}) \sum_s d_s^{\boldsymbol{\pi}, \boldsymbol{p}_1} \sum_{a \in \mathcal{A}} \pi_{sa} \sum_{s'} |p_{1,sas'} - p_{2,sas'}| \\
&\leq (2 + 5t_{\mathrm{mix}}) \|\boldsymbol{p}_1 - \boldsymbol{p}_2\|_{1,\infty},
\end{aligned}
$$

where the inequality $(a)$ is obtained due to the fact that for any $\boldsymbol{\pi} \in \Pi$, $\boldsymbol{p} \in (\Delta^S)^{S \times A}$, and $(s, a, s') \in \mathcal{S} \times \mathcal{A} \times \mathcal{S}$,

$$
|g_{sas'}^{\boldsymbol{\pi}, \boldsymbol{p}}| \leq |c_{sas'} - J(\boldsymbol{\pi}, \boldsymbol{p}) + v_{s'}^{\boldsymbol{\pi}, \boldsymbol{p}}| \leq |c_{sas'}| + \|\boldsymbol{d}^{\boldsymbol{\pi}, \boldsymbol{p}}\|_1 \cdot \|\boldsymbol{c}^{\boldsymbol{\pi}, \boldsymbol{p}}\|_\infty + |v_{s'}^{\boldsymbol{\pi}, \boldsymbol{p}}| \leq 2 + 5t_{\mathrm{mix}}.
$$

As for the stationary distribution, we can straightforward obtain that

$$
|d_s^{\boldsymbol{\pi}, \boldsymbol{p}_1} - d_s^{\boldsymbol{\pi}, \boldsymbol{p}_2}| \leq (2 + 5t_{\mathrm{mix}}) \cdot \|\boldsymbol{p}_1 - \boldsymbol{p}_2\|_{1,\infty}, \ \forall s \in \mathcal{S}.
$$

Then, we consider the sensitive bound for the differential value function, that is

$$
\begin{aligned}
\|\boldsymbol{v}^{\boldsymbol{\pi}, \boldsymbol{p}_1} - \boldsymbol{v}^{\boldsymbol{\pi}, \boldsymbol{p}_2}\|_\infty &= \|\boldsymbol{H}^{\boldsymbol{\pi}, \boldsymbol{p}_1} \boldsymbol{c}^{\boldsymbol{\pi}, \boldsymbol{p}_1} - \boldsymbol{H}^{\boldsymbol{\pi}, \boldsymbol{p}_2} \boldsymbol{c}^{\boldsymbol{\pi}, \boldsymbol{p}_2}\|_\infty \\
&= \|\boldsymbol{H}^{\boldsymbol{\pi}, \boldsymbol{p}_1} (\boldsymbol{c}^{\boldsymbol{\pi}, \boldsymbol{p}_1} - \boldsymbol{c}^{\boldsymbol{\pi}, \boldsymbol{p}_2}) + (\boldsymbol{H}^{\boldsymbol{\pi}, \boldsymbol{p}_1} - \boldsymbol{H}^{\boldsymbol{\pi}, \boldsymbol{p}_2}) \boldsymbol{c}^{\boldsymbol{\pi}, \boldsymbol{p}_2}\|_\infty,
\end{aligned}
$$

where $\boldsymbol{H}^{\boldsymbol{\pi}, \boldsymbol{p}} := (\boldsymbol{I} - \boldsymbol{P}^{\boldsymbol{\pi}} + \boldsymbol{P}^{\boldsymbol{\pi}, \infty})^{-1} (\boldsymbol{I} - \boldsymbol{P}^{\boldsymbol{\pi}, \infty})$ is defined in the proof of Lemma 3.1. We notice that

$$
\begin{aligned}
\boldsymbol{H}^{\boldsymbol{\pi}, \boldsymbol{p}_1} - \boldsymbol{H}^{\boldsymbol{\pi}, \boldsymbol{p}_2} &= (\boldsymbol{I} - \boldsymbol{P}_1^{\boldsymbol{\pi}} + \boldsymbol{P}_1^{\boldsymbol{\pi}, \infty})^{-1} (\boldsymbol{I} - \boldsymbol{P}_1^{\boldsymbol{\pi}, \infty}) - (\boldsymbol{I} - \boldsymbol{P}_2^{\boldsymbol{\pi}} + \boldsymbol{P}_2^{\boldsymbol{\pi}, \infty})^{-1} (\boldsymbol{I} - \boldsymbol{P}_2^{\boldsymbol{\pi}', \infty}) \\
&= \left( (\boldsymbol{I} - \boldsymbol{P}_1^{\boldsymbol{\pi}} + \boldsymbol{P}_1^{\boldsymbol{\pi}, \infty})^{-1} - (\boldsymbol{I} - \boldsymbol{P}_1^{\boldsymbol{\pi}} + \boldsymbol{P}_1^{\boldsymbol{\pi}, \infty})^{-1} \right) (\boldsymbol{I} - \boldsymbol{P}_1^{\boldsymbol{\pi}, \infty}) \\
&\quad + (\boldsymbol{I} - \boldsymbol{P}_2^{\boldsymbol{\pi}} + \boldsymbol{P}_2^{\boldsymbol{\pi}, \infty})^{-1} (\boldsymbol{P}_2^{\boldsymbol{\pi}, \infty} - \boldsymbol{P}_1^{\boldsymbol{\pi}, \infty}) \\
&= \left( (\boldsymbol{I} - \boldsymbol{P}_1^{\boldsymbol{\pi}} + \boldsymbol{P}_1^{\boldsymbol{\pi}, \infty})^{-1} (\boldsymbol{P}_1^{\boldsymbol{\pi}} - \boldsymbol{P}_1^{\boldsymbol{\pi}, \infty} - \boldsymbol{P}_2^{\boldsymbol{\pi}} + \boldsymbol{P}_2^{\boldsymbol{\pi}, \infty}) (\boldsymbol{I} - \boldsymbol{P}_2^{\boldsymbol{\pi}} + \boldsymbol{P}_2^{\boldsymbol{\pi}, \infty})^{-1} \right) (\boldsymbol{I} - \boldsymbol{P}_1^{\boldsymbol{\pi}, \infty}) \\
&\quad + (\boldsymbol{I} - \boldsymbol{P}_2^{\boldsymbol{\pi}} + \boldsymbol{P}_2^{\boldsymbol{\pi}, \infty})^{-1} (\boldsymbol{P}_2^{\boldsymbol{\pi}, \infty} - \boldsymbol{P}_1^{\boldsymbol{\pi}, \infty}).
\end{aligned}
$$

Thus, we have

$$
\begin{aligned}
\|\boldsymbol{v}^{\boldsymbol{\pi}, \boldsymbol{p}_1} - \boldsymbol{v}^{\boldsymbol{\pi}, \boldsymbol{p}_2}\|_\infty &\leq \|\boldsymbol{H}^{\boldsymbol{\pi}, \boldsymbol{p}_1} (\boldsymbol{c}^{\boldsymbol{\pi}, \boldsymbol{p}_1} - \boldsymbol{c}^{\boldsymbol{\pi}, \boldsymbol{p}_2})\|_\infty + \|(\boldsymbol{I} - \boldsymbol{P}_2^{\boldsymbol{\pi}} + \boldsymbol{P}_2^{\boldsymbol{\pi}, \infty})^{-1} (\boldsymbol{P}_2^{\boldsymbol{\pi}, \infty} - \boldsymbol{P}_1^{\boldsymbol{\pi}, \infty}) \boldsymbol{c}^{\boldsymbol{\pi}, \boldsymbol{p}_2}\|_\infty \\
&\quad + \left\| \left( (\boldsymbol{I} - \boldsymbol{P}_1^{\boldsymbol{\pi}} + \boldsymbol{P}_1^{\boldsymbol{\pi}, \infty})^{-1} (\boldsymbol{P}_1^{\boldsymbol{\pi}} - \boldsymbol{P}_1^{\boldsymbol{\pi}, \infty} - \boldsymbol{P}_2^{\boldsymbol{\pi}} + \boldsymbol{P}_2^{\boldsymbol{\pi}, \infty}) (\boldsymbol{I} - \boldsymbol{P}_2^{\boldsymbol{\pi}} + \boldsymbol{P}_2^{\boldsymbol{\pi}, \infty})^{-1} \right) (\boldsymbol{I} - \boldsymbol{P}_1^{\boldsymbol{\pi}, \infty}) \boldsymbol{c}^{\boldsymbol{\pi}, \boldsymbol{p}_2} \right\|_\infty \\
&\leq \|\boldsymbol{H}^{\boldsymbol{\pi}, \boldsymbol{p}_1}\|_\infty \cdot \|\boldsymbol{c}^{\boldsymbol{\pi}, \boldsymbol{p}_1} - \boldsymbol{c}_2^{\boldsymbol{\pi}, \boldsymbol{p}}\|_\infty + \|(\boldsymbol{I} - \boldsymbol{P}_2^{\boldsymbol{\pi}} + \boldsymbol{P}_2^{\boldsymbol{\pi}, \infty})^{-1}\|_\infty \cdot \|\boldsymbol{P}_2^{\boldsymbol{\pi}, \infty} - \boldsymbol{P}_1^{\boldsymbol{\pi}, \infty}\|_\infty \\
&\quad + \|(\boldsymbol{I} - \boldsymbol{P}_1^{\boldsymbol{\pi}} + \boldsymbol{P}_1^{\boldsymbol{\pi}, \infty})^{-1}\|_\infty \cdot \|\boldsymbol{P}_1^{\boldsymbol{\pi}} - \boldsymbol{P}_1^{\boldsymbol{\pi}, \infty} - \boldsymbol{P}_2^{\boldsymbol{\pi}'} + \boldsymbol{P}_2^{\boldsymbol{\pi}', \infty}\|_\infty \cdot \|(\boldsymbol{I} - \boldsymbol{P}_2^{\boldsymbol{\pi}} + \boldsymbol{P}_2^{\boldsymbol{\pi}, \infty})^{-1}\|_\infty \\
&\overset{(a)}{\leq} 2C \|\boldsymbol{c}^{\boldsymbol{\pi}, \boldsymbol{p}_1} - \boldsymbol{c}^{\boldsymbol{\pi}, \boldsymbol{p}_1}\|_\infty + C \|\boldsymbol{d}^{\boldsymbol{\pi}, \boldsymbol{p}_1} - \boldsymbol{d}^{\boldsymbol{\pi}, \boldsymbol{p}_2}\|_1 + C^2 \|\boldsymbol{P}_1^{\boldsymbol{\pi}} - \boldsymbol{P}_2^{\boldsymbol{\pi}}\|_\infty + C^2 \|\boldsymbol{P}_1^{\boldsymbol{\pi}, \infty} - \boldsymbol{P}_2^{\boldsymbol{\pi}, \infty}\|_\infty \\
&\leq (2C + CSC_d^{\boldsymbol{p}} + C^2 + C^2 SC_d^{\boldsymbol{p}}) \|\boldsymbol{p}_1 - \boldsymbol{p}_2\|_{1,\infty},
\end{aligned}
$$

where the inequality $(a)$ is obtained due to $\|(\boldsymbol{I} - \boldsymbol{P}_1^{\boldsymbol{\pi}, \infty}) \boldsymbol{c}^{\boldsymbol{\pi}, \boldsymbol{p}_2}\|_\infty \leq 1$. By applying the definition of $g_{sas'}^{\boldsymbol{\pi}, \boldsymbol{p}}$, we have

$$
\begin{aligned}
\|\boldsymbol{g}_{sa}^{\boldsymbol{\pi}, \boldsymbol{p}_1} - \boldsymbol{g}_{sa}^{\boldsymbol{\pi}, \boldsymbol{p}_2}\| &= \max_{s'} |c_{sas'} - J(\boldsymbol{\pi}, \boldsymbol{p}_1) + v_{s'}^{\boldsymbol{\pi}, \boldsymbol{p}_1} - (c_{sas'} - J(\boldsymbol{\pi}, \boldsymbol{p}_2) + v_{s'}^{\boldsymbol{\pi}, \boldsymbol{p}_2})| \\
&\leq |J(\boldsymbol{\pi}, \boldsymbol{p}_1) - J(\boldsymbol{\pi}, \boldsymbol{p}_2)| + \|\boldsymbol{v}^{\boldsymbol{\pi}, \boldsymbol{p}_1} - \boldsymbol{v}^{\boldsymbol{\pi}, \boldsymbol{p}_2}\|_\infty \leq (C_J^{\boldsymbol{p}} + C_v^{\boldsymbol{p}}) \|\boldsymbol{p}_1 - \boldsymbol{p}_2\|_{1,\infty}.
\end{aligned}
$$

$\square$

*Proof of Lemma 4.3.* Lemma 4.1 provides the analytical form of the partial derivative, that is,

$$
\frac{\partial J(\boldsymbol{\pi}, \boldsymbol{p})}{\partial p_{sas'}} = d_s^{\boldsymbol{\pi}, \boldsymbol{p}} \pi_{sa} g_{sas'}^{\boldsymbol{\pi}, \boldsymbol{p}}.
$$

Then, we have

$$\|\nabla_{\boldsymbol{p}} J(\boldsymbol{\pi}, \boldsymbol{p})\| = \sqrt{\sum_{s,a,s'} (d_s^{\boldsymbol{\pi},\boldsymbol{p}} \pi_{sa} g_{sas'}^{\boldsymbol{\pi},\boldsymbol{p}})^2} \leq (2 + 5t_{\mathrm{mix}}) \sqrt{\sum_{s,a,s'} (d_s^{\boldsymbol{\pi},\boldsymbol{p}} \pi_{sa})^2} \leq (2 + 5t_{\mathrm{mix}}) \sqrt{S},$$

which verifies that $J(\boldsymbol{\pi}, \boldsymbol{p})$ is $L_{\boldsymbol{p}}$-Lipschitz in $\boldsymbol{p}$ by showing the boundedness of $\nabla_{\boldsymbol{p}} J(\boldsymbol{\pi}, \boldsymbol{p})$. We then turn to derive the smoothness condition of $J(\boldsymbol{\pi}, \boldsymbol{p})$ utilizing the similar perturbation theory of stochastic matrices applied in Lemma 3.1. Let $\boldsymbol{p}, \boldsymbol{p}' \in (\Delta^S)^{S \times A}$ be any transition kernels. We introduce $\boldsymbol{p}(\alpha)$ as a convex combination of transition kernels $\boldsymbol{p}$ and $\boldsymbol{p}'$, that is, $\boldsymbol{p}(\alpha) = (1 - \alpha)\boldsymbol{p} + \alpha\boldsymbol{p}' := \boldsymbol{p} + \alpha\boldsymbol{v}$ where $\boldsymbol{v} = \boldsymbol{p}' - \boldsymbol{p}$. Notice that the partial derivative of $J(\boldsymbol{\pi}, \boldsymbol{p}(\alpha))$ over $\alpha$ is

$$\frac{\partial J(\boldsymbol{\pi}, \boldsymbol{p}(\alpha))}{\partial \alpha} = \sum_{s,a,s'} d_s^{\boldsymbol{\pi},\boldsymbol{p}(\alpha)} \pi_{sa} v_{sas'} g_{sas'}^{\boldsymbol{\pi},\boldsymbol{p}(\alpha)},$$

then, we are going to derive the smoothness by showing $|\frac{\partial J(\boldsymbol{\pi}(\alpha),\boldsymbol{p})}{\partial \alpha} - \frac{\partial J(\boldsymbol{\pi},\boldsymbol{p})}{\partial \alpha}| \leq \alpha \ell_{\boldsymbol{\pi}}$:

$$\left| \frac{\partial J(\boldsymbol{\pi}, \boldsymbol{p}(\alpha))}{\partial \alpha} - \frac{\partial J(\boldsymbol{\pi}, \boldsymbol{p})}{\partial \alpha} \right| = \left| \sum_{s,a,s'} d_s^{\boldsymbol{\pi},\boldsymbol{p}(\alpha)} \pi_{sa} v_{sas'} g_{sas'}^{\boldsymbol{\pi},\boldsymbol{p}(\alpha)} - \sum_{s,a,s'} d_s^{\boldsymbol{\pi},\boldsymbol{p}} \pi_{sa} v_{sas'} g_{sas'}^{\boldsymbol{\pi},\boldsymbol{p}} \right|$$

$$\leq \left| \sum_{s,a,s'} d_s^{\boldsymbol{\pi},\boldsymbol{p}(\alpha)} \pi_{sa} v_{sas'} (g_{sas'}^{\boldsymbol{\pi},\boldsymbol{p}(\alpha)} - g_{sas'}^{\boldsymbol{\pi},\boldsymbol{p}}) \right| + \left| \sum_{s,a,s'} (d_s^{\boldsymbol{\pi},\boldsymbol{p}(\alpha)} - d_s^{\boldsymbol{\pi},\boldsymbol{p}}) \pi_{sa} v_{sas'} g_{sas'}^{\boldsymbol{\pi},\boldsymbol{p}} \right|$$

$$\leq \sum_{s,a} d_s^{\boldsymbol{\pi},\boldsymbol{p}(\alpha)} \pi_{sa} \left| \langle \boldsymbol{v}_{sa}, \boldsymbol{g}_{sa}^{\boldsymbol{\pi},\boldsymbol{p}(\alpha)} - \boldsymbol{g}_{sa}^{\boldsymbol{\pi},\boldsymbol{p}} \rangle \right| + \sum_s \left| d_s^{\boldsymbol{\pi},\boldsymbol{p}(\alpha)} - d_s^{\boldsymbol{\pi},\boldsymbol{p}} \right| \sum_a \pi_{sa} \left| \langle \boldsymbol{v}_{sa}, \boldsymbol{g}_{sa}^{\boldsymbol{\pi},\boldsymbol{p}} \rangle \right|$$

$$\leq \sum_{s,a} d_s^{\boldsymbol{\pi},\boldsymbol{p}(\alpha)} \pi_{sa} \|\boldsymbol{g}_{sa}^{\boldsymbol{\pi},\boldsymbol{p}(\alpha)} - \boldsymbol{g}_{sa}^{\boldsymbol{\pi},\boldsymbol{p}}\|_\infty \|\boldsymbol{v}_{sa}\|_1 + \sum_s \left| d_s^{\boldsymbol{\pi},\boldsymbol{p}(\alpha)} - d_s^{\boldsymbol{\pi},\boldsymbol{p}} \right| \sum_a \pi_{sa} \|\boldsymbol{v}_{sa}\|_1 \|\boldsymbol{g}_{sa}^{\boldsymbol{\pi},\boldsymbol{p}}\|_\infty$$

$$\overset{(a)}{\leq} 2 C_g^{\boldsymbol{p}} \|\alpha \boldsymbol{v}\|_{1,\infty} + 2(2 + 5t_{\mathrm{mix}}) C_d^{\boldsymbol{p}} S \|\alpha \boldsymbol{v}\|_{1,\infty}$$

$$\leq 2(2 C_g^{\boldsymbol{p}} + 2(2 + 5t_{\mathrm{mix}}) C_d^{\boldsymbol{p}} S) \alpha,$$

where the inequality $(a)$ is obtained from the sensitivity of $\boldsymbol{q}_s^{\boldsymbol{\pi},\boldsymbol{p}}$ and $d_s^{\boldsymbol{\pi},\boldsymbol{p}}$ (Lemma 4.2), as well as the facts $|g_{sas'}^{\boldsymbol{\pi},\boldsymbol{p}}| \leq 2 + 5t_{\mathrm{mix}}$ and $\|\boldsymbol{v}\|_{1,\infty} \leq 2$. Therefore, the smoothness is proved. $\qquad \square$

*Proof of Theorem 4.4.* By the adversary' difference performance lemma (Lemma D.1), we have for any $\boldsymbol{\pi} \in \Pi$ and $\boldsymbol{p} \in (\Delta^S)^{S \times A}$, we have

$$J(\boldsymbol{\pi}, \boldsymbol{p}^\star) - J(\boldsymbol{\pi}, \boldsymbol{p}) = \sum_{s \in \mathcal{S}} d_s^{\boldsymbol{\pi},\boldsymbol{p}^\star} \sum_{a \in \mathcal{A}} \pi_{sa} \sum_{s'} (p_{sas'}^\star - p_{sas'}) \cdot g_{sas'}^{\boldsymbol{\pi},\boldsymbol{p}}.$$

Then, we can obtain that

$$0 \leq J(\boldsymbol{\pi}, \boldsymbol{p}^\star) - J(\boldsymbol{\pi}, \boldsymbol{p}) = \sum_{s \in \mathcal{S}} d_s^{\boldsymbol{\pi},\boldsymbol{p}^\star} \sum_{a \in \mathcal{A}} \pi_{sa} \sum_{s'} (p_{sas'}^\star - p_{sas'}) \cdot g_{sas'}^{\boldsymbol{\pi},\boldsymbol{p}}$$

$$= \sum_{s \in \mathcal{S}} \left( \frac{d_s^{\boldsymbol{\pi},\boldsymbol{p}^\star}}{d_s^{\boldsymbol{\pi},\boldsymbol{p}}} \right) d_s^{\boldsymbol{\pi},\boldsymbol{p}^\star} \sum_{a \in \mathcal{A}} \pi_{sa} \sum_{s'} (p_{sas'}^\star - p_{sas'}) \cdot g_{sas'}^{\boldsymbol{\pi},\boldsymbol{p}}$$

$$\leq \max_{\bar{\boldsymbol{p}} \in \mathcal{P}} \sum_{s \in \mathcal{S}} \left( \frac{d_s^{\boldsymbol{\pi},\boldsymbol{p}^\star}}{d_s^{\boldsymbol{\pi},\boldsymbol{p}}} \right) d_s^{\boldsymbol{\pi},\boldsymbol{p}^\star} \sum_{a \in \mathcal{A}} \pi_{sa} \sum_{s'} (p_{sas'}^\star - \bar{p}_{sas'}) \cdot g_{sas'}^{\boldsymbol{\pi},\boldsymbol{p}}$$

$$\overset{(a)}{\leq} \sum_{s \in \mathcal{S}} \left( \frac{d_s^{\boldsymbol{\pi},\boldsymbol{p}^\star}}{d_s^{\boldsymbol{\pi},\boldsymbol{p}}} \right) d_s^{\boldsymbol{\pi},\boldsymbol{p}^\star} \underbrace{\max_{\bar{\boldsymbol{p}}_s \in \mathcal{P}_s} \sum_{a \in \mathcal{A}} \pi_{sa} \sum_{s'} (p_{sas'}^\star - \bar{p}_{sas'}) \cdot g_{sas'}^{\boldsymbol{\pi},\boldsymbol{p}}}_{\geq 0, \quad = 0 \text{ while } \bar{\boldsymbol{p}}_s = \boldsymbol{p}_s^\star}$$

$$\leq M \cdot \max_{\bar{\boldsymbol{p}} \in \mathcal{P}} \langle \bar{\boldsymbol{p}} - \boldsymbol{p}, \nabla_{\boldsymbol{p}} J(\boldsymbol{\pi}, \boldsymbol{p}) \rangle,$$

where the inequality $(a)$ holds only under the rectangularity condition. $\qquad \square$

Now, we proceed to show our main convergence result on the inner worst-case kernel evaluation. Here we define the gradient mapping

$$G^{\beta}(\boldsymbol{p}) := \frac{1}{\beta}\left(\text{Proj}_{\mathcal{P}}(\boldsymbol{p} + \beta \nabla_{\boldsymbol{p}} J(\boldsymbol{\pi}, \boldsymbol{p})) - \boldsymbol{p}\right). \tag{25}$$

Notice that $\mathcal{P}$ is convex and $J(\boldsymbol{\pi}, \boldsymbol{p})$ is $\ell_{\boldsymbol{p}}$-smooth in $\boldsymbol{p}$, then we turn to derive our main result.

*Proof of Theorem 4.5.* Lemma B.13 implies that if $\|G^{\beta}(\boldsymbol{p})\| \leq \epsilon$, then

$$\nabla_{\boldsymbol{p}} J(\boldsymbol{\pi}, \boldsymbol{p}^+) \in \mathcal{N}_{\mathcal{P}}(\boldsymbol{p}^+) + 2\epsilon \mathcal{B}(1), \tag{26}$$

where $\boldsymbol{p}^+ := \boldsymbol{p} + \beta G^{\beta}(\boldsymbol{p})$, $\mathcal{N}_{\mathcal{P}}$ is the norm cone of the set $\mathcal{P}$, and $\mathcal{B}(r) := \{\boldsymbol{x} \in \mathbb{R}^n : \|\boldsymbol{x}\| \leq r\}$. By the gradient dominance condition established in Lemma 4.4,

$$\min_{k \in \{0, \cdots, K-1\}} \{J(\boldsymbol{\pi}, \boldsymbol{p}^{\boldsymbol{\pi}}) - J(\boldsymbol{\pi}, \boldsymbol{p}_t)\} \leq M \cdot \min_{k \in \{0, \cdots, K-1\}} \max_{\bar{\boldsymbol{p}} \in \mathcal{P}} \langle \bar{\boldsymbol{p}} - \boldsymbol{p}_k, \nabla_{\boldsymbol{p}} J(\boldsymbol{\pi}, \boldsymbol{p}_k)\rangle$$
$$\leq M \cdot \max_{\bar{\boldsymbol{p}} \in \mathcal{P}} \langle \bar{\boldsymbol{p}} - \boldsymbol{p}_{\hat{k}}, \nabla_{\boldsymbol{p}} J(\boldsymbol{\pi}, \boldsymbol{p}_{\hat{k}})\rangle, \tag{27}$$

where $\hat{k} := 1 + \arg\min_{k \leq K-1} \|G^{\beta}(\boldsymbol{p}_k)\|$. Note that, Lemma B.12 implies that

$$\|G^{\beta}(\boldsymbol{p}_{\hat{k}-1})\| \leq \sqrt{\frac{2\ell_{\boldsymbol{p}}(J(\boldsymbol{\pi}, \boldsymbol{p}^{\boldsymbol{\pi}}) - J(\boldsymbol{\pi}, \boldsymbol{p}_0))}{K}} \leq \sqrt{\frac{2\ell_{\boldsymbol{p}}}{K}},$$

where the last inequality holds due to $|J(\boldsymbol{\pi}, \boldsymbol{p})| \leq 1$. While we set that

$$\sqrt{\frac{2\ell_{\boldsymbol{p}}}{K}} \leq \frac{\delta_{\boldsymbol{\pi}}}{4M\sqrt{SA}} \iff K \geq \frac{32\ell_{\boldsymbol{p}}M^2 SA}{\delta_{\boldsymbol{\pi}}^2} = \mathcal{O}(\delta_{\boldsymbol{\pi}}^{-2}),$$

then

$$\|G^{\beta}(\boldsymbol{p}_{\hat{k}-1})\| \leq \frac{\delta_{\boldsymbol{\pi}}}{4M\sqrt{SA}}.$$

Hence, by applying the equation (26), we have

$$(27) \leq M \cdot \max_{\bar{\boldsymbol{p}} \in \mathcal{P}} \|\bar{\boldsymbol{p}} - \boldsymbol{p}_{\hat{k}}\| \cdot 2 \cdot \frac{\epsilon_{\boldsymbol{\pi}}}{4M\sqrt{SA}} = \delta_{\boldsymbol{\pi}},$$

where for any $\boldsymbol{p}_1, \boldsymbol{p}_2 \in \mathcal{P}$,

$$\|\boldsymbol{p}_1 - \boldsymbol{p}_2\| \leq \|\boldsymbol{p}_1\| + \|\boldsymbol{p}_2\| \leq 2\sqrt{SA}. \tag{28}$$

$\square$

**Theorem D.2.** *(Lamperski, 2021, Theorem 1) Assume that $\eta \leq \frac{1}{2}$. There are positive constants $h, c_4, c_5$ such that for all integers $k \geq 4$, the following bound holds:*

$$W_1(\mathcal{L}(\boldsymbol{p}_k), \nu_{\lambda J}) \leq c_4 e^{-\eta h k} + c_5 (\eta \log k)^{\frac{1}{4}}.$$

*In particular, if $\eta = \frac{\log K}{4hK}$ and $K \geq 4$, then:*

$$W_1(\mathcal{L}(\boldsymbol{p}_K), \nu_{\lambda J}) \leq \left(c_4 + \frac{c_5}{(4h)^{\frac{1}{4}}}\right) K^{-\frac{1}{4}} (\log K)^{\frac{1}{2}}$$

**Proposition D.3.** *(Lamperski, 2021, Proposition 2) The constant $c_1$ and $c_2$ grows linearly with $n$. If $D^2 \ell \lambda < 8$, then we can set $h = \frac{4}{D^2 \ell \lambda} \geq \frac{\ell}{2}$, while $c_4$ and $c_5$ grows polynominally with respect to $\left(1 - \frac{D^2 \ell \lambda}{8}\right)^{-2}$ and $\lambda^{-\frac{1}{4}}$. In general, we have a positive constant $c_3$ and a monotonically increasing polynomial $p$ (independent of $\eta$ and $\lambda$) such that for all $\lambda > 0$, the following bounds hold:*

$$h \geq c_6 \lambda \exp\left\{-\frac{D^2 \ell \lambda}{4}\right\}, \quad \max\{c_4, c_5\} \leq p(\lambda^{-\frac{1}{4}}) \exp\left\{\frac{3D^2 \ell \lambda}{4}\right\}.$$

We note that only the notation has been adapted to our setting; the results and their proof remain unchanged from (Lamperski, 2021).

**Lemma D.4.** *For any function $J : \mathcal{P} \to \mathbb{R}$, let $\nu_J$ be the probability measure defined by $\nu_J(\mathcal{B}) = \frac{\int_{\mathcal{B}} \exp\{J(\boldsymbol{\pi}, \boldsymbol{p})\} d\boldsymbol{p}}{\int_{\mathcal{P}} \exp\{J(\boldsymbol{\pi}, \tilde{\boldsymbol{p}})\} d\tilde{\boldsymbol{p}}}$. Particularly, $\nu_0$ represents uniform measure. If $J$ is $L_{\boldsymbol{p}}$-Lipschitz, then the KL divergence of $\nu_J$ from the uniform measure $\nu_0$ is bounded by:*

$$0 \leq \mathrm{KL}(\nu_J, \nu_0) \leq \mathbb{E}_{\nu_J}[J(\boldsymbol{\pi}, \boldsymbol{p})] - \max_{\boldsymbol{p} \in \mathcal{P}} J(\boldsymbol{\pi}, \boldsymbol{p}) + n \log \left( \max \left\{ \frac{2}{r}, \frac{(r + \sqrt{r^2 + D^2}) L_{\boldsymbol{p}}}{r \log(2)} \right\} \right) + \log(2D^n).$$

*Proof of D.4.* The KL divergence is bounded below by $0$ as a standard result in (Cover & Thomas, 2012). Then we only need to prove the upper bound.

Denote $\boldsymbol{p}^\star$ as the optimal solution of $\max_{\boldsymbol{p} \in \mathcal{P}} J(\boldsymbol{\pi}, \boldsymbol{p})$ for a given policy $\boldsymbol{\pi}$. It exists as $J$ is Lipschitz continuos and $\mathcal{P}$ is compact. Then multiply both numerator and denominator of $\nu_J$ by $\exp(-J(\boldsymbol{\pi}, \boldsymbol{p}^\star))$, we have

$$\nu_J(\mathcal{B}) = \frac{\int_{\mathcal{B}} e^{J(\boldsymbol{\pi}, \boldsymbol{p}) - J(\boldsymbol{\pi}, \boldsymbol{p}^\star)} d\boldsymbol{p}}{\int_{\mathcal{P}} e^{J(\boldsymbol{\pi}, \tilde{\boldsymbol{p}}) - J(\boldsymbol{\pi}, \boldsymbol{p}^\star)} d\tilde{\boldsymbol{p}}}.$$

Noted that by definition of uniform distribution, $\nu_0(d\boldsymbol{p}) = \frac{d\boldsymbol{p}}{\mathrm{vol}(\mathcal{P})}$. So the definition of KL divergence implies

$$\mathrm{KL}(\nu_J, \nu_0) = \mathbb{E}_{\boldsymbol{p} \sim \nu_J}[J(\boldsymbol{\pi}, \boldsymbol{p}) - J(\boldsymbol{\pi}, \boldsymbol{p}^\star)] + \log(\mathrm{vol}(\mathcal{P})) - \log\left(\int_{\mathcal{P}} e^{J(\boldsymbol{\pi}, \tilde{\boldsymbol{p}}) - J(\boldsymbol{\pi}, \boldsymbol{p}^\star)} d\tilde{\boldsymbol{p}}\right).$$

Note that the set $\mathcal{P}$ is contained in a ball of radius $D$ (for example, for an ellipsoidal ambiguity set of size $\theta$, we have $D = \sqrt{\theta / \lambda_{\min}}$, where $\lambda_{\min}$ is the smallest eigenvalue of the shape matrix $Q$). Hence, the volume satisfies $\mathrm{vol}(\mathcal{P}) \leq D^n \frac{\pi^{n/2}}{\Gamma(n/2+1)} \leq 2D^n$, where $\pi$ denotes the circular constant. The second inequality holds for $n > 10$.

Therefore, to upper bound the denominator, it suffices to obtain a lower bound on $\int_{\mathcal{P}} e^{J(\boldsymbol{\pi}, \tilde{\boldsymbol{p}}) - J(\boldsymbol{\pi}, \boldsymbol{p}^\star)} d\tilde{\boldsymbol{p}}$. Since the function $J$ is $L_{\boldsymbol{p}}$-Lipschitz continuous, it follows that

$$0 \geq J(\boldsymbol{\pi}, \tilde{\boldsymbol{p}}) - J(\boldsymbol{\pi}, \boldsymbol{p}^\star) \geq -L_{\boldsymbol{p}} \|\tilde{\boldsymbol{p}} - \boldsymbol{p}^\star\|.$$

Besides, $e^{-L_{\boldsymbol{p}} \|\tilde{\boldsymbol{p}} - \boldsymbol{p}^\star\|} \geq 1/2$ if and only if $\|\tilde{\boldsymbol{p}} - \boldsymbol{p}^\star\| \leq \frac{\log 2}{L_{\boldsymbol{p}}}$.

Set $\epsilon = \frac{\log 2}{L_{\boldsymbol{p}}}$ and let $\mathcal{B}_{\boldsymbol{p}^\star}(\epsilon)$ be the ball of radius $\epsilon$ centred at $\boldsymbol{p}^\star$. Then for any $\mathcal{C} \subset \mathcal{P} \cap \mathcal{B}_{\boldsymbol{p}^\star}(\epsilon)$, we have

$$\int_{\mathcal{P}} e^{J(\boldsymbol{\pi}, \tilde{\boldsymbol{p}}) - J(\boldsymbol{\pi}, \boldsymbol{p}^\star)} d\tilde{\boldsymbol{p}} \geq \frac{1}{2} \mathrm{vol}(\mathcal{P} \cap \mathcal{B}_{\boldsymbol{p}^\star}(\epsilon)) \geq \frac{1}{2} \mathrm{vol}(\mathcal{C}).$$

As proved in (Lamperski, 2021, Lemma 15), the $\mathcal{C}$ contains a ball with radius $\min\{\frac{r}{2}, \frac{r\epsilon}{r + \sqrt{r^2 + D^2}}\}$, where $r$ denotes the radius of a ball contained in $\mathcal{P}$. Then lemma follows by using the fact that a ball of radius $\hat{r}$ has volume given by $\frac{\pi^{\frac{n}{2}}}{\Gamma(\frac{n}{2}+1)} \hat{r}^n$. $\qquad\square$

*Proof of 4.6.* Recall that $J(\boldsymbol{\pi}, \boldsymbol{p})$ is $L_{\boldsymbol{p}}$-Lipschitz, so that $\lambda J(\boldsymbol{\pi}, \boldsymbol{p})$ is $\lambda L_{\boldsymbol{p}}$-Lipschitz. Assume that $\tilde{\boldsymbol{p}}$ follows distribution $\nu_{\lambda J}$, then applying Lemma D.4 we can obtain

$$\mathbb{E}_{\tilde{\boldsymbol{p}} \sim \nu_{\lambda J}}[J(\boldsymbol{\pi}, \tilde{\boldsymbol{p}})] \geq \max_{\boldsymbol{p} \in \mathcal{P}} J(\boldsymbol{\pi}, \boldsymbol{p}) - \frac{n}{\lambda} \log \left( 2D \max \left\{ \frac{2}{r}, \frac{(r + \sqrt{r^2 + D^2}) L_{\boldsymbol{p}} \lambda}{r \log 2} \right\} \right). \tag{29}$$

Let $\boldsymbol{x}_k$ be the $k$-th iterate of the algorithm 3, then

$$\mathbb{E}[J(\boldsymbol{\pi}, \boldsymbol{p}_k)] \overset{\text{Kantorovich Duality}}{\geq} \mathbb{E}_{\nu_J}[J(\boldsymbol{\pi}, \boldsymbol{p})] - L_{\boldsymbol{p}} W_1(\mathcal{L}(\boldsymbol{p}_k), \nu_J)$$

$$\overset{(29)}{\geq} \max_{\boldsymbol{p} \in \mathcal{P}} J(\boldsymbol{\pi}, \boldsymbol{p}) - \frac{n \log(c_1 \max\{1, \lambda\})}{\lambda} - L_{\boldsymbol{p}} W_1(\mathcal{L}(\boldsymbol{p}_k), \nu_J),$$

where $c_1 = 2D \max \left\{ \frac{2}{r}, \frac{(r + \sqrt{r^2 + D^2})\lambda}{r \log(2)} \right\}$

Then we will show how to tune the parameters to achieve an average suboptimality of $\delta_{\boldsymbol{\pi}}$.

First, we choose $\lambda$ so that $\frac{n \log(c_1 \max\{1, \lambda\})}{\lambda} \leq \frac{\delta_{\boldsymbol{\pi}}}{2}$. Without loss of generality, we assume $\lambda > 1$. Set $x = \log(c_1 \lambda)$, so that $\lambda = c_1^{-1} e^x$ and the required bound becomes

$$x e^{-x} \leq \frac{c_1 \delta_{\boldsymbol{\pi}}}{2n}.$$

For any $\kappa \in (0, 1)$, the maximum value of $x e^{-(1-\kappa)x}$ occurs at $x = (1 - \kappa)^{-1}$, so that for all $x \in \mathbb{R}$:

$$x e^{-x} = x e^{-(1-\kappa)x} e^{-\kappa x} \leq \frac{1}{(1-\kappa)e} e^{-\kappa x}. \tag{30}$$

So it is sufficient to set $e^{-\kappa x} \leq \frac{c_1 \delta_{\boldsymbol{\pi}}(1-\kappa)e}{2n}$ to achieve the bound. Then plugging back, it shows that a sufficient condition for $\frac{n \log(c_1 \lambda)}{\lambda} \leq \frac{\delta_{\boldsymbol{\pi}}}{2}$ is given by

$$\lambda \geq c_1^{-1} \left( \frac{2n}{c_1(1-\kappa)\delta_{\boldsymbol{\pi}} e} \right)^{\frac{1}{\kappa}} \tag{31}$$

Now for a fixed $\lambda \geq 1$, the bounds from Theorem D.2 and Proposition D.3 to give that

$$W_1(\mathcal{L}(\boldsymbol{p}_k), \nu_J) \leq \left( c_4 + \frac{c_5}{(4h)^{\frac{1}{4}}} \right) K^{-\frac{1}{4}} (\log K)^{\frac{1}{2}}$$

$$\leq p(1) e^{\frac{3D^2 \ell \lambda}{4}} \left( 1 + \frac{e^{\frac{D^2 \ell \lambda}{16}}}{4^{\frac{1}{4}} c_3} \right) K^{-\frac{1}{4}} (\log K)^{\frac{1}{2}}$$

$$\leq p(1) \left( 1 + \frac{1}{4^{\frac{1}{4}} c_3} \right) e^{\frac{13D^2 \ell \lambda}{16}} K^{-\frac{1}{4}} (\log K)^{\frac{1}{2}}.$$

Similar to (30), we can derive the following inequality for all $\rho \in (0, 1/2)$ and all $K > 0$:

$$K^{-\frac{1}{4}} (\log K)^{\frac{1}{2}} = \left( K^{-\frac{1}{2}+\rho} K^{-\rho} \log K \right)^{\frac{1}{2}} \leq \sqrt{\frac{K^{-\frac{1}{2}+\rho}}{e\rho}}.$$

Thus, to achieve $L_{\boldsymbol{p}} W_1(\mathcal{L}(\boldsymbol{p}_k), \nu_J) \leq \frac{\delta_{\boldsymbol{\pi}}}{2}$, it is sufficient to have

$$K^{-\frac{1}{2}+\rho} \leq e\rho \left( \frac{\delta_{\boldsymbol{\pi}}}{2} \right)^2 \left( p(1) \left( 1 + \frac{1}{4^{\frac{1}{4}} c_3} \right) e^{\frac{13D^2 \ell \lambda}{16}} \right)^{-2} := \epsilon$$

which is equivalent to have

$$K \geq \frac{1}{\epsilon^{\frac{2}{1-2\rho}}}.$$

To separate the parameters, we can define a constant $c_6$ that is independent to $\eta, \lambda, \delta_{\boldsymbol{\pi}}, \alpha$ and $\rho$, such that the bound above holds whenever

$$K \geq \frac{c_6^{\frac{2}{1-2\rho}}}{\delta_{\boldsymbol{\pi}}^{\frac{4}{1-2\rho}}} \exp\left( \frac{13D^2 \ell \lambda}{4(1-2\rho)} \right),$$

Combining the results above with (31), we obtain the desired conclusion by introducing $a = \frac{4}{1-2\rho} > 4$ and $b = \frac{1}{\alpha} > 1$, and by substituting the dimension of the transition kernel as $n = S^2 A$. $\qquad \square$

## E. Experiment Details

In this section, we provide the implementation details and experimental setup. All results were generated on an Apple M2 Max with 32 GB LPDDR5 memory. The algorithms are implemented in Python 3.11.5, and we use Gurobi 11.0.3 to solve any linear optimization problems involved.

### E.1. Environment Setting

A GARNET MDP $\mathcal{G}(|\mathcal{S}|, |\mathcal{A}|, b)$ is defined by three parameters: $|\mathcal{S}|$, the size of the state space; $|\mathcal{A}|$, the size of the action space; and $b$, the branching factor, which specifies the number of accessible next states for each state. The cost is generated randomly following a uniform distribution within $[0, 10]$.

### E.2. Rectangular Ambiguous Case

We validate the convergence of RP2G on three different sizes of GARNET MDPs with $(s, a)$-rectangular ambiguity sets. Specifically, we use the $\ell_1$ norm to measure the size of the ambiguity set.

$$\mathcal{P}_{sa} = \{\boldsymbol{p}_{sa} \in \Delta^S \mid \|\boldsymbol{p}_{sa} - \bar{\boldsymbol{p}}_{sa}\|_1 \leq \kappa_{sa}\},$$

where $\bar{\boldsymbol{p}}_{sa}$ is the nominal transition kernel and $\kappa_{sa}$ is randomly generated from a uniform distribution over the interval $[0, 0.3]$.

We run 50 sample instances with 250 iterations of RP2G for each GARNET problem. At each iteration, we record the relative error between the objective values of RP2G and the optimal value $J^\star$, calculated as $(|J(\boldsymbol{\pi}_t, \boldsymbol{p}_t) - J^\star|)/J^\star$. For the optimal value $J^\star$, we use the robust value iteration method from (Wang et al., 2023c) as our benchmark, with the stopping criterion $\|\boldsymbol{v}_t - \boldsymbol{v}_{t-1}\|_2 \leq 5 \times 10^{-4}$.

The relative error values are plotted in Figure 1. The line represents the average relative error across the 50 instances for each problem at each iteration. The upper and lower envelopes of the lines correspond to the 95 and 5 percentiles of the 50 samples, respectively. These results demonstrate the convergence and optimality of our algorithm.

### E.3. Runtime

This subsection compares the computational efficiency of RP2G with the only existing gradient-based method to highlight the advantage of adopting a decreasing tolerance sequence $\{\delta_t\}_{t \geq 0}$. Specifically, we consider the robust policy mirror descent algorithm (Sun et al., 2024) as a benchmark, which assumes the inner worst-case evaluation problem is solved exactly. For computational considerations, we set the inner worst-case evaluation problem in the benchmark method with a fixed tolerance of $\delta = 10^{-5}$ at each iteration, while RP2G adopts a decreasing tolerance sequence initialized with $\delta_0 = 1$ and reduced at a rate of $\tau = 0.95$.

We use the same environment and ambiguity settings as described in the Section E.2. Table 1 reports the runtime for the two methods, with termination determined by minimal changes in the objective value, i.e., $|J(\boldsymbol{\pi}_t, \boldsymbol{p}_t) - J(\boldsymbol{\pi}_{t-1}, \boldsymbol{p}_{t-1})| \leq 10^{-4}$. The results demonstrate the effectiveness of adopting a decreasing tolerance sequence in improving runtime efficiency.

### E.4. Non-Rectangular Ambiguous Case

We adopt the ellipsoid form (Wiesemann et al., 2013; Li et al., 2023) for constructing the non-rectangular ambiguity set, defined as

$$\mathcal{P} = \left\{ \boldsymbol{p} : (\boldsymbol{p} - \bar{\boldsymbol{p}})^\top \boldsymbol{\Sigma} (\boldsymbol{p} - \bar{\boldsymbol{p}}) \leq r \right\}.$$

We set size parameter $r = 1$. The Hessian matrix $\boldsymbol{\Sigma}$ is generated as $\boldsymbol{\Sigma} = \boldsymbol{\sigma}\boldsymbol{\sigma}^\top$, with $\boldsymbol{\sigma} \in \mathbb{R}^{S \times A \times S}$ being a column vector whose elements are independently sampled from a uniform distribution over $[0, 0.1]$. The nominal transition kernel is denoted as $\bar{\boldsymbol{p}}$.

We validate the robustness of RP2G on the non-rectangular ambiguity set by comparing it against the non-robust policy gradient method in $\mathcal{G}(5, 3, 4)$. At each iteration, we evaluate and compare the values of $\Psi(\boldsymbol{\pi}_t) = \max_{\boldsymbol{p} \in \mathcal{P}} J(\boldsymbol{\pi}_t, \boldsymbol{p})$ for both methods. The results are plotted in Figure 2. The line represents the mean values across 20 instances, while the shaded area indicates the range between the 5 and 95 percentiles over the 20 samples. This figure demonstrates the robustness of RP2G compared to the non-robust policy gradient and also shows the convergence of the RP2G algorithm when using Algorithm 3 for inner worst-case evaluation.

### E.5. Discount Factor Discussion

In this section, we perform experiments with different choices of the discount factor $\gamma$. For robust discounted MDPs, we use the Double-Loop Robust Policy Gradient (DRPG) algorithm proposed in (Wang et al., 2023a). The experiment is also

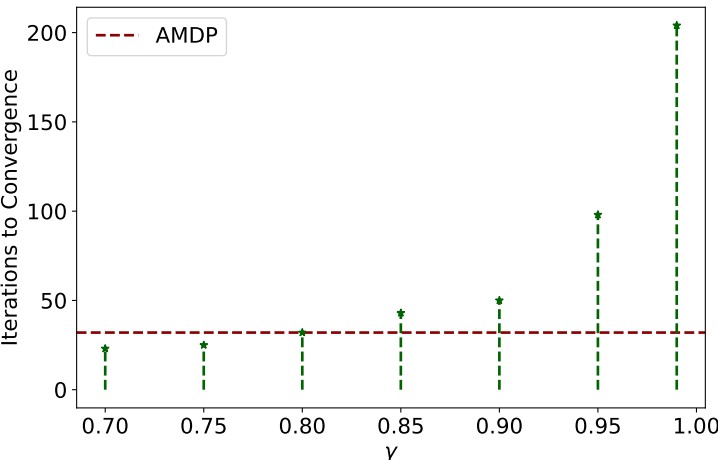

*Figure 3.* Iterations to convergence under different discount factors and AMDP.

conducted on GARNET MDPs $\mathcal{G}(5, 3, 5)$

We set the step size $\alpha = (1 - \gamma)^2$ for each robust discounted MDP, consistent with the theoretical convergence analysis in the reference. For RP2G, we use a step size of $\beta = 0.05$. Figure 3 illustrates the number of iterations required for convergence under different values of the discount factor, compared to the iteration count for convergence in robust average-reward MDPs. Convergence is determined when $|(J_{t+1} - J_t)/J_t| \leq 10^{-4}$. It is evident that as $\gamma$ increases, the number of iterations for convergence grows significantly, showing a clear upward trend.

### E.6. Rectangular and Non-rectangular Ambiguity Comparison

In this section, we conduct experiment with two ambiguity sets under same size to show the superiority of non-rectangularity in application. In particular, we compare the put-of-sample performance of $(s, a)$-rectangular ambiguity set and ellipsoid ambiguity in a classical inventory control problem (Zipkin, 2000).

In the inventory control environment, the agent decides how many items to order ($a$) based on the current inventory level ($s$). Each ordered item incurs a cost of 1, and we assume that there is no delay in the ordering process.

After the customer demand $d \in [0, m]$ is realized, the agent observes the updated inventory level. If the resulting inventory level $s + a - d$ falls below the allowed backlog limit (set to $-m$ in our experiments), the agent can only fulfill $s + a$ units of demand. In this case, the effective demand is truncated to $d = s + a$. On the other hand, if the updated inventory level exceeds the maximum inventory capacity $m$, it is reset to $m$. At the end of each period, the agent incurs a holding cost of 1 for each item in inventory and a backlog cost of 1 for each unit back-ordered.

To estimate transition probabilities for policy training, we simulate 1000 trajectories using uniformly random actions and demands. Each trajectory consists of $100m$ time steps. Table 3 reports the average long-term cost under two ambiguity sets. The results show that the policy derived from the non-rectangular RAMDP is less conservative, as indicated by its lower average cost.

*Table 3.* Average performance under different inventory levels over 2,000 out-of-sample trajectories, comparing the $(s, a)$-rectangular and non-rectangular (ellipsoidal) ambiguity sets with set size equal to 0.1.

| Inventory Level($m$) | 3 | 4 | 5 | 6 | 7 | 8 |
|---|---|---|---|---|---|---|
| $(s, a)$-rectangular | 2.906 | 3.980 | 4.360 | 5.246 | 6.014 | 6.620 |
| Non-rectangular | 2.850 | 3.659 | 4.289 | 5.187 | 5.931 | 6.582 |

