# OpenReview forum: "Provable Policy Gradient for Robust Average-Reward MDPs  Beyond Rectangularity"
_ICML.cc/2025/Conference — ICML 2025 poster_

### Official Review · Reviewer_zvi6 · 2025-03-07

**Overall Recommendation:** 3

**Summary:**

The studies very sparsely studied topic of average reward robust MDPs. Specifically, the paper establishes global convergence of robust policy gradient (RPG) for average reward MDPs with an iteration complexity of $O(\epsilon^{-4})$ given oracle access to the robust gradient. This paper combines the techniques of RPG in discounted reward MDP [1] and convergence policy gradient for average reward MDPs [2].

In addition, this paper improves the smoothness coefficient for average reward MDPs over existing result in [2].  The paper is well written.




[1] Wang, Q., Xu, S., Ho, C. P., and Petrick, M. Policy gradient for robust markov decision processes. arXiv preprint
arXiv:2410.22114, 2024a.

[2] Kumar, N., Murthy, Y., Shufaro, I., Levy, K. Y., Srikant, R., and Mannor, S. On the global convergence of policy gradient in average reward markov decision processes.arXiv preprint arXiv:2403.06806, 2024b.

**Claims And Evidence:**

Yes, seems good.

**Essential References Not Discussed:**

All seems good.

**Experimental Designs Or Analyses:**

No.

**Methods And Evaluation Criteria:**

Yes, seems good.

**Other Comments Or Suggestions:**

Suggestion: A more readable proofs would be beneficial to the readers.

**Other Strengths And Weaknesses:**

Strengths: Good theoretical results of significance.

Weakness: The proofs are well presented (above average), however a little more clarification on the maths proofs would be beneficial to the readers.

**Questions For Authors:**

Question: This paper is direct combination of [1] and [2]. Please elaborate on the technical novelty.

**Relation To Broader Scientific Literature:**

In RL, average reward MDPs has its own independent significance yet the most of the results are of discounted reward MDPs, due to the complexity of the setting. This work makes this gap little narrower.

**Theoretical Claims:**

I went over all the proofs but not in great details. All seems good.

---

> ### Author Rebuttal · Authors · 2025-04-01
>
> Thank you for your insightful questions and taking the time to assess our manuscript.
>
> 1. **_Comment 1: Additional clarification in proofs could be helpful ._**
>
> Thank you very much for providing helpful suggestions on writing style and missing clarification! We will clarify the wording in the next version of our manuscript, including more detailed explanations, to improve readability.
>
> 2. **_Comment 2: Technical novelty need to be further elaborated._**
>
> We'd like to emphasis that the extension from the discounted setting to the average-reward setting
> is non-trivial, even the average-reward MDP can be seen as a special case of discounted MDPs ($\gamma=1$). Please see our reply to **Reviewer iYjz, Comment 2** and the introduction of [2] for a detailed explanation.
>
> For the outer loop, we establish Lipschitz continuity via a novel sensitivity analysis and introduce a gradient dominance condition specifically tailored to the average-reward setting, leading to an outer-loop convergence guarantee. Compared to [2], we provide improved smoothness coefficients. While our approach adopts a double-loop structure, which is widely used in various fields such as game theory [3], min-max optimization [4,5], and robust MDPs [1], our analysis relies on standard non-convex minimax optimization techniques, which are significantly different from the mirror descent analysis used in [1].
>
> For the inner loop, we propose two tailored algorithms for worst-case transition evaluation under both rectangular and non-rectangular settings, whereas [1] only addresses the rectangular case. To establish the inner-loop convergence guarantee, we derive the first form of the adversarial transition gradient (Lemma 4.1), establish first adversarial smoothness conditions using sensitivity analysis techniques (Lemma 4.3), and prove the first adversarial gradient dominance condition (Theorem 4.4), along with corresponding inner convergence guarantee.
>
> Therefore, our work differs substantially from both references [1,2]. Building on these theoretical advancements, our approach offers a comprehensive framework for addressing robust average-reward MDPs beyond existing methods. We will further clarify these contributions in the next version of our paper.
>
> [1] Wang, Q., Xu, S., Ho, C. P., and Petrick, M. Policy gradient for robust markov decision processes.
>
> [2] Kumar, N., Murthy, Y., Shufaro, I., Levy, K. Y., Srikant, R., and Mannor, S. On the global convergence of policy gradient in average reward markov decision processes.
>
> [3] Ding D., Wei C. Y., Zhang K., \& Jovanovic M. 2022. Independent policy gradient for large-scale markov potential games: Sharper rates, function approximation, and game-agnostic convergence.
>
> [4] Jin C., Netrapalli P., \& Jordan M. 2020. What is local optimality in nonconvex-nonconcave minimax optimization?
>
> [5] Davis D., \& Drusvyatskiy D. 2019. Stochastic model-based minimization of weakly convex functions.

---

> > ### Comment · Reviewer_zvi6 · 2025-04-01
> >
> > Thanks for response. I have few additional minor commnets.
> >
> > The paper establishes iteration complexity of $O(\epsilon^{-4})$ for average reward robust policy gradient with non-agressive learning rate. The work [1] ,  [2], and Theorem 4.7 of [3] establishes $O(\epsilon^{-1})$ convergence rate for smooth-discounted reward robust MDP,  softmax discounted reward robust MDP, and sa-rectangular average reward robust MDPs respectively. Could authors comment on these? Could the results from [1] and [2] be translated to average reward too? If the result of this paper could be improved to $O(\epsilon^{-1})$ similar to [3]?
> >
> >
> > ---
> >
> >
> >
> >
> > [1] @inproceedings{
> > kumar2023towards,
> > title={Towards Faster Global Convergence of Robust Policy Gradient Methods},
> > author={Navdeep Kumar and Ilnura Usmanova and Kfir Yehuda Levy and Shie Mannor},
> > booktitle={Sixteenth European Workshop on Reinforcement Learning},
> > year={2023},
> > url={https://openreview.net/forum?id=cWrwdbEBx5}
> > }
> >
> > [2] @misc{wang2024policygradientrobustmarkov,
> >       title={Policy Gradient for Robust Markov Decision Processes},
> >       author={Qiuhao Wang and Shaohang Xu and Chin Pang Ho and Marek Petrik},
> >       year={2024},
> >       eprint={2410.22114},
> >       archivePrefix={arXiv},
> >       primaryClass={cs.LG},
> >       url={https://arxiv.org/abs/2410.22114},
> > }
> >
> > [3] @inproceedings{
> > sun2024policy,
> > title={Policy Optimization for Robust Average Reward {MDP}s},
> > author={Zhongchang Sun and Sihong He and Fei Miao and Shaofeng Zou},
> > booktitle={The Thirty-eighth Annual Conference on Neural Information Processing Systems},
> > year={2024},
> > url={https://openreview.net/forum?id=6FPZLnp1Zn}
> > }

---

> > > ### Author Response · Authors · 2025-04-02
> > >
> > > Thank you very much for providing helpful comments on enhancing the significance of our work!
> > >
> > > First of all, we'd like to clarify that [2] and [3] are well discussed in our manuscript as concurrent works (both were made public around November). The contribution of [2] as the inspiration for the idea of decreasing tolerance is clarified and highlighted in Section 3.1 (Line 219). The distinction between our work and [3] is discussed in Section 1.1 (Line 71-79), and we also adopt the robust mirror descent policy gradient from [3] as a numerical benchmark method. We appreciate the reviewer for pointing out another relevant work [1] on robust policy gradient methods, which will help us further refine our literature review.
> > >
> > > On the technical side, while [1,2,3] all establish an $\mathcal{O}(\epsilon^{-1})$ convergence rate, [1,2] focus on the discounted setting, whereas [3] studies the average-reward setting. However, the fundamental analysis techniques differ. The result in [1] relies on standard optimization tools with a restrictive smoothness assumption, which does not hold for general ambiguity sets. In contrast, the analyses in [2] and [3] are based on standard mirror descent techniques, where smoothness of the objective is not necessarily required under direct parameterization.
> > >
> > > Therefore, we believe that adopting the analysis of [2] in the discounted setting to robust MDPs with the average-reward criterion is a promising direction for future research. At this stage, [3] serves as a pioneering work in this direction, establishing a faster $\mathcal{O}(\epsilon^{-1})$ convergence rate under a more restrictive $(s,a)$-rectangularity assumption compared to our general convergence guarantee. However, whether such an extension would be effective for RAMDPs with general ambiguity sets remains an open question that warrants further investigation.

---

### Official Review · Reviewer_WW4Y · 2025-03-10

**Overall Recommendation:** 4

**Summary:**

This paper studies robust *average-reward* MDPs (RAMDPs) with general ambiguity sets. It proposes a policy-gradient-based algorithm, RP2G, that leverages an exponentially decaying adaptive tolerance mechanism $\{ \delta_t \}$ to enable provably efficient policy updates, assuming an oracle that solves the inner problem $\Psi(\pi)$, and coming with global convergence guarantees. Further, it also proposes an optimization algorithm (also leveraging projected gradient update) to solve $\Psi(\pi)$ for the worst-case kernel that also provably converges (in some sense). The performance of the proposed algorithms are supported by simulation results.

**Claims And Evidence:**

All claims are supported by concrete results.

**Essential References Not Discussed:**

N/A

**Experimental Designs Or Analyses:**

Since this is mostly a theory-oriented paper, the numerical simulations only act as a supporting evidence. For this purpose, the experimental design and results look good to me.

**Methods And Evaluation Criteria:**

The algorithm makes intuitive sense, and the evaluation method is reasonable.

**Other Comments Or Suggestions:**

1. There are a few typos observed during reading.
    * On line 715: does "DFunctions" mean "diffusion value functions"?
    * In Lemma 4.2: "is" should be "are".

**Other Strengths And Weaknesses:**

Strengths:
1. The paper is overall well-written. Algorithms are sufficiently motivated and introduced with intuitive ideas.
2. Theoretical results presented in the main text, though lacking quite a few technical details, are largely self-explanatory and convincing.
    * It's delightful to see the nice symmetry between the sensitivity in $\boldsymbol{\pi}$ and that in $\boldsymbol{p}$.

Weaknesses:
1. The discussion regarding the projected Langevin dynamics that solves $\Psi(\pi)$ for general ambiguity sets is a little too hand-wavy. Basically no technical details are provided in the paper (including the appendix). The current Section 4.3 also does not fit into the flow very well.
    * After a quick look, (Lamperski, 2021) is a very technical paper that deals with generic non-convex learning. Hence technical details are definitely needed here to show how it fits into the average-reward MDP setting.
    * It is a little weird to see "... is the first ..." after results cited from other papers, without further explanations.
2. The numerical experiments can preferably be extended to include more common benchmarks.

**Questions For Authors:**

1. Currently the theoretical guarantee for RP2G (Theorem 3.5) is isolated from that for the adversary gradient ascent (Theorem 4.6), in the sense that the former does not take the estimation error of the inner problem into consideration. Is it possible to incorporate this estimation error into the overall bound, probably using some quick fixes?
2. Honestly I'm not very familiar with average-reward MDPs. What makes average-reward MDPs so different from discounted MDPs? Is it the mixing property and the upper bounds involving mixing time that is key to the theoretical analysis? Is this approach potentially adaptable to discounted MDPs or episodic MDPs?

**Relation To Broader Scientific Literature:**

N/A

**Theoretical Claims:**

The theoretical results are supported by proofs that are checked to be correct.

---

> ### Author Rebuttal · Authors · 2025-04-01
>
> Thank you very much for taking the time to review our paper and your insightful comments as well as suggestions.
>
> **_Comment 1: Technical details in Section 4.3 should be added._**
>
> Thank you so much for pointing out the missing details in this section! That would be definitely helpful if full technical details are provided. We will include a detailed discussion of the proof and analysis in the next version of our paper.
>
> **_Comment 2: Statement with "... is the first..." could be misleading._**
>
> Thank you for your suggestion. We would like to emphasize that while the technical tools for solving generic non-convex problems are well established [1], no existing literature has applied such analysis to robust average-reward MDPs. To the best of our knowledge, the most recent policy gradient method for robust average-reward MDPs [2] is limited to the more conservative $(s,a)$-rectangular case. We will clarify the wording in the manuscript in our revision to improve readability.
>
> **_Comment 3: Considering additional benchmarks in the experiment could be helpful._**
>
> Thank you for this insightful suggestion. We do totally agree that incorporating additional benchmarks would strengthen our experimental evaluation. As suggested by **Reviewer sWdH**, we conducted an experiment in an inventory control setting to demonstrate RP2G's superior performance (see https://drive.google.com/file/d/1VKnmT5_Wzpj6PwH_UbHImimhrhKBpVgq/view?usp=sharing). Our results show that the policy obtained by solving the non-rectangular RAMDP is less conservative (see our reply to **Reviewer sWdH, Comment 1** for a detailed explanation). We will evaluate our algorithms on other benchmarks in the next version of this paper.
>
> **_Comment 4: Typos._**
>
> Thank you for pointing out the typos. We will update our paper according to these suggestions!
>
> **_Question 5: Could you incorporate the estimation error of Algorithm 3 into the overall convergence bound in Theorem 3.5?_**
>
> Thank you for raising this important question. We would like to emphasize that the theoretical guarantee for RP2G (Theorem 3.5) accounts for the inner output error ($\delta_{t}$ at $t$-th iteration). However, unlike Algorithm 2 for the inner problem, which features a simple and effective structure for updating the inner transition kernel, Algorithm 3 employs Monte Carlo sampling to achieve a probabilistic convergence guarantee, which introduces additional estimation error. Therefore, incorporating this estimation error from Algorithm 3 (as detailed in Theorem 4.6) into a deterministic convergence guarantee (shown in Theorem 3.5) remains challenging. We believe that investigating the overall convergence bound while accounting for the inner worst-case evaluation error is an interesting direction for future work that could further enhance the significance of our results.
>
> **_Question 6: What makes average-reward MDPs so different from discounted MDPs?_**
>
> This difference arises in both theory and practice. While the RAMDP be seen as a special case of the RMDP ($\gamma=1$), extending the theoretical framework from the discounted setting to the average-reward setting is non-trivial (see our reply to **Reviewer iYjz, Comment 2** and [3] for a detailed explanation). In practice, many real-world systems prioritize steady-state or long-term behaviour, where policies derived from discounted MDPs may perform poorly (see Introduction line 37-48).
>
> **_Question 7: Is the mixing time important to the theoretical analysis?_**
>
> At this stage, the mixing time is crucial for bounding the differential (action) value function, which plays a key role in establishing Lipschitz continuity and the gradient dominance condition.
>
> **_Question 8: Is this approach potentially adaptable to discounted MDPs or episodic MDPs?_**
>
> Thank you for this insightful question. Regarding the discounted MDPs, [4] already proposed methods with a double-loop structure for solving RMDPs. However, extending our approach to episodic MDPs seems challenging, as the ergodicity assumption does not hold in this setting, preventing a direct application of our theoretical framework. Developing a suitable policy gradient framework for episodic MDPs remains an important direction for future research.
>
> [1] Lamperski, A. 2021. Projected stochastic gradient langevin algorithms for constrained sampling and non-convex learning.
>
> [2] Sun, Z., He, S., Miao, F., \& Zou, S. 2024. Policy optimization for robust average reward mdps.
>
> [3] Kumar N, Murthy Y, Shufaro I, et al. 2024. On the global convergence of policy gradient in average reward markov decision processes.
>
> [4] Li M., Kuhn, D., \& Sutter T. 2023. Policy gradient algorithms for robust mdps with non-rectangular uncertainty sets.

---

### Official Review · Reviewer_iYjz · 2025-03-13

**Overall Recommendation:** 3

**Summary:**

The paper investigates methods for solving robust Markov Decision Processes (MDPs) under the average-reward criterion. Building on existing approaches developed for the discounted-reward setting, the authors extend these ideas to the average-reward framework. The study presents multiple algorithms tailored to different structures of the ambiguity set, with a particular focus on whether the set exhibits a rectangular structure. Convergence results for these algorithms are provided, demonstrating their theoretical validity and practical implications.

**Claims And Evidence:**

The paper explores methods for solving robust Markov Decision Processes (MDPs) under the average-reward criterion, extending ideas originally developed for the discounted-reward setting. The authors introduce a robust projected policy gradient algorithm (RP2G) and establish its global convergence, despite the challenges posed by non-rectangular ambiguity sets. This convergence analysis relies on an oracle that efficiently solves the so-called inner problem. To address this, the paper presents two specialized algorithms for solving the inner problem: one based on projected Langevin dynamics and another using projected gradient ascent, specifically for rectangular ambiguity sets.

The presented results are interesting, meaningful, and convincingly supported. However, one key question arises: Based on my understanding, Li et al. (2023) have already obtained similar results in the discounted-cost setting. Could the authors clarify what distinguishes their work from Li et al. (2023), beyond the shift from the discounted-reward to the average-reward criterion?

**Essential References Not Discussed:**

none

**Experimental Designs Or Analyses:**

I don't fully understand Figure 2. It would be helpful to explain better why this plot is interesting.

**Methods And Evaluation Criteria:**

The examples and evaluation criteria considered are meaningful and interesting, however, again almost identical to Li et al. (2023), with the only noticeable difference being the averge-reward criterion.

**Other Comments Or Suggestions:**

-

**Other Strengths And Weaknesses:**

The paper is well written and easy to follow. The only concern I have is the novelty compared to Li et al. (2023), see points raised above.

**Questions For Authors:**

-

**Relation To Broader Scientific Literature:**

The literature is well cited and discussed.

**Theoretical Claims:**

The theoretical results look surprisingly similar to those achieved for the discounted-reward setting in Li et al. (2023).

Could you state clearer where the differences are?

What happens if in the discounted cost setting you choose the discount factor $\gamma\to 1$? Can you recover the average-reward results?

---

> ### Author Rebuttal · Authors · 2025-04-01
>
> We are sincerely grateful to the reviewer for the insightful comments and valuable questions.
>
> **_Comment 1. Additional clarification on the difference between our work and Li et al. (2023) is needed._**
>
> We apologize for the insufficient explanation of the contributions of this work. It is important to clarify that the extension from the discounted-reward setting to the average-reward setting is non-trivial (see our reply to **Comment 2** for a detailed discussion).
> While both our work and that of Li et al. (2023) rely on a double-loop structure, along with similar properties such as the smoothness of the objective function $J$ and the gradient-dominance condition, the technical foundations used to establish these properties are quite different. Compared to their work, our contributions span both loops. For the outer loop, we establish Lipschitz continuity via a novel sensitivity analysis and introduce a gradient dominance condition specifically tailored to the average-reward setting, leading to the convergence guarantee. For the inner loop, We propose a tailored projected gradient ascent algorithm for worst-case transition evaluation under rectangularity, with a convergence guarantee. This approach is structurally different from Algorithm 3.2 in Li et al. (2023). Although our inner convergence results may appear similar, the underlying analytical tools are fundamentally different.
>
> We hope this clarifies the key differences between our work and that of Li et al. (2023), both in theoretical foundations and algorithmic contributions.
>
> **_Comment 2: Recovering average-reward results from the discounted setting could be discussed._**
>
> We thank the reviewer for raising this point. While it is theoretically possible to recover average-reward results by taking the limit as $\gamma \to 1$ in the discounted-reward setting, this approach is often infeasible in practice. As shown in Li et al. (2023), the number of iterations is $\mathcal{O}(\frac{1}{(1 - \gamma)^4})$ (hidden within their constants), which increases rapidly as $\gamma\to 1$. This makes the computational cost prohibitively high for large $\gamma$. We also provide empirical evidence in Appendix E.5, where we observe a significant increase in iteration count as $\gamma$ increases. These results highlight the practical limitations of using discounted-reward methods for solving robust average-reward MDPs, and motivate the need for a tailored algorithm, as proposed in our work.
>
> **_Comment 3: Additional explanation for Figure 2 could be helpful._**
>
> Thank you for pointing this out. Regarding the experimental setup, we obtain and record the policies from both robust and non-robust AMDPs at each iteration. To assess policy robustness, we evaluate their performance under the worst-case transition scenario, i.e., computing $\max_{\boldsymbol{ p} \in \mathcal{P}} J(\boldsymbol{\pi}, \boldsymbol{ p})$ with given $\boldsymbol{\pi}$, and then record these values for plotting. This comparison is wildly adopted in robust MDPs literature [8,9] as a standard approach to demonstrate robustness.
>
> In terms of result interpretation, the RAMDP policies consistently achieve lower costs under worst-case transitions, highlighting their effectiveness against adversarial kernels. Moreover, as the number of iterations increases, the worst-case evaluation cost stabilizes, indicating convergence. These findings are in line with our theoretical results in Section 4.3.
>
> We hope these additional explanations could help clarify our numerical experiment.
>
> [1] Lamperski, A. 2021. Projected stochastic gradient langevin algorithms for constrained sampling and non-convex learning.
>
> [2] Wang, Y., Velasquez, A., Atia, G., Prater-Bennette, A., \& Zou, S. 2024. Robust Average-Reward Reinforcement Learning.
>
> [3] Riemer, M., Khetarpal, K., Rajendran, J., \& Chandar, S. 2024. Balancing context length and mixing times for reinforcement learning at scale.
>
> [4] Kearns, M., Mansour, Y., \& Ng, A. 1999. Approximate planning in large POMDPs via reusable trajectories.
>
> [5] Jin, Y., \& Sidford, A. 2020. Efficiently solving MDPs with stochastic mirror descent.
>
> [6] Puterman, M. L. 2014. Markov decision processes: discrete stochastic dynamic programming.
>
> [7] Xiao, L. 2022. On the convergence rates of policy gradient methods.
>
> [8] Sun, Z., He, S., Miao, F., \& Zou, S. 2024. Policy optimization for robust average reward mdps.
>
> [9] Tamar, A., Mannor, S., \& Xu, H. 2014. Scaling up robust MDPs using function approximation.

---

### Official Review · Reviewer_sWdH · 2025-03-14

**Overall Recommendation:** 3

**Summary:**

This paper extends the work of Li et al. 2023 from solving robust MDPs in discounted setting to average reward setting. Numerical results are also provided.

## update after rebuttal
I thank the authors for their efforts in writing the rebuttal. I agree with that letting the discount factor to 1 would of course be a too simple adaptation, but still the essential difference seems to be an alternative dynamic equation -- the same technique from discount factor setting still applies. I will keep my score.

**Claims And Evidence:**

The claims are well-evidenced.

**Essential References Not Discussed:**

The paper is well-positioned and relevant literature is discussed.

**Experimental Designs Or Analyses:**

The methods are evaluated in random synthetic MDPs instance GARNET, which is a standard benchmark in robust MDPs. The methods are evaluated from runtime and compared to non-robust method. However, the paper only compares the worst-case policy value of the one output by a non-robust PG method and the proposed method, which seemed incomplete because the power of non-rectangularity is really statistical efficiency compared to rectangular uncertainty sets, and the key contribution is extension to average reward criteria. Please refer to my comment for possible improvements.

**Methods And Evaluation Criteria:**

Methods (PG) and evaluation criteria (average reward robust MDP) all make sense.

**Other Comments Or Suggestions:**

Please consider comparing to a data-driven MDP instance and compare the out-of-sample performance of robust MDP with non-rectangular uncertainty sets vs. the one with a rectangular uncertainty set. In addition, it would be enhancing the paper's significance if the authors can show the essentialty of using average-reward criteria (e.g., comparing against work of Li et al. 2023) where we can see the practical advantage of using average reward robust MDP with non-rectangular uncertainty set.

**Other Strengths And Weaknesses:**

Weakness: On a technical level, the essential thing that changes from solving discounted to average reward setting is the gradient formula and the magnitude of Lipschitz constant. Thus, I am unsure about the novelty on the technical side.

**Questions For Authors:**

How does the proof work when only access to stochastic policy gradients is available? What are the major changes required then?

**Relation To Broader Scientific Literature:**

The paper is solving an important instance of robust MDPs -- the one with average reward criteria. In this sense, the paper complements the work of Li et al. 2023 well.

**Theoretical Claims:**

No, but I believe the plausibleness of the theoretical claims because of correctness in the discounted setting.

---

> ### Author Rebuttal · Authors · 2025-04-01
>
> Thank you for your insightful questions and constructive suggestions!
>
> **_Comment 1: Additional comparison between rectangular and non-rectangular RAMDPs would be helpful._**
>
> Thanks for the suggestion! As the (non-)rectangularity only affects the ambiguity structure and appears to be independent of the chosen reward criterion [1], we adopt the non-rectangularity to the average-reward case without further justification; thus we did not design an experiment specifically to show the advantages of non-rectangularity over rectangularity under this criterion. We do agree that it would be helpful if the superiority of the non-rectangularity could be better demonstrated in this setting. To address this, we compare the performance of $(s,a)$-rectangular and non-rectangular RAMDPs in an inventory control setting with the same set size (see https://drive.google.com/file/d/1VKnmT5_Wzpj6PwH_UbHImimhrhKBpVgq/view?usp=sharing). The results, presented in the table, show that the policy obtained from the non-rectangular RAMDP is less conservative, as evidenced by its lower average cost. We will provide a more detailed discussion in the next version of our paper and highlight the advantage and superiority of non-rectangularity.
>
> **_Comment 2: Further clarification of the technical novelty could be useful._**
>
> We appreciate the reviewer's concerns regarding our technical novelty and would like to emphasize that the extension from the discounted setting to the average-reward setting is non-trivial (see the example in **Reviewer iYjz, Comment 2**)
>
> As our proposed algorithm follows a double-loop structure, widely used in various fields, including game theory [2], min-max optimization [3,4], and robust MDPs [5], our analysis builds on standard non-convex optimization techniques, which were also adopted by [5]. Compared to [5], our novel contributions are twofold: (1) for the outer loop, we establish Lipschitz continuity via a novel sensitivity analysis and introduce a gradient dominance condition with an convergence guarantee tailored to the average-reward setting; (2) for the inner loop, we provide an effective algorithm for worst-case transition evaluation under rectangularity, along with which is structurally different from Algorithm 3.2 in [5]. While achieving a similar convergence rate, our analysis differs from [5], relying on standard non-convex optimization techniques [6].
>
>
> **_Comment 3. Essentiality of using average-reward instead of discounted reward should be highlighted._**
>
> Thanks for your suggestion. As our work is theoretically oriented, our contribution focus on developing efficient algorithms with theoretical convergence guarantees for RAMDPs. We do understand average-reward setting is well-suited for agents concerned with long-term or steady-state policy behaviour, such as resource allocation, portfolio management, and healthcare [7,8]. In this sense, exploring the practical advantages of RAMDPs in these applications represents a promising direction for our future research.
>
> Moreover, as shown in Appendix E.5, when $\gamma$ approaches $1$, the computational cost increases significantly when using discounted-reward MDPs to approximate average-reward solutions. This underscores the necessity of methods specifically designed for the average-reward setting.
>
> **_Comment 4. Adaptation to stochastic policy gradients could be considered._**
>
> Thank you for this valuable question. Our analysis assumes a model-based setting in which the MDP structure is known except for the transition kernel, so the policy gradient can be computed exactly. When only stochastic policy gradients are available, additional challenges arise in modeling and in ensuring robustness under broader uncertainty. Extending our results to this case is a promising direction for future work, for example by incorporating robust temporal-difference methods such as [9].
>
> [1] Wiesemann W., Kuhn D., & Rustem B. 2013. Robust Markov decision processes.
>
> [2] Ding D., Wei C. Y., Zhang K., & Jovanovic M. 2022. Independent policy gradient for large-scale markov potential games: Sharper rates, function approximation, and game-agnostic convergence.
>
> [3] Jin C., Netrapalli P., & Jordan M. 2020. What is local optimality in nonconvex-nonconcave minimax optimization?
>
> [4] Davis D., & Drusvyatskiy D. 2019. Stochastic model-based minimization of weakly convex functions.
>
> [5] Li M., Kuhn, D., & Sutter T. 2023. Policy gradient algorithms for robust mdps with non-rectangular uncertainty sets.
>
> [6] Beck A. 2017. First-order methods in optimization.
>
> [7] Ghalme G., Nair V., Patil V., & Zhou Y. 2021. Long-term resource allocation fairness in average markov decision process (amdp) environment.
>
> [8] Patrick J., & Begen M. A. 2011. Markov decision processes and its applications in healthcare.
>
> [9] Wang Y., & Zou S. 2022. Policy gradient method for robust reinforcement learning.

---

### Decision · Program_Chairs · 2025-05-01

**Decision:**

Accept (poster)

**Comment:**

The paper studied robust MDP in the average reward setting. The  paper proposes a policy gradient based algorithm for robust average-reward MDPs (RAMDPs) that is applicable beyond the typical rectangularity assumption on transition ambiguity. Convergence analysis of the proposed algorithm is presented, and numerical experiments are conducted to support some of the claims of the paper.

After rebuttal, the reviewers are leaning to the positive side. But there are still several concerns. One concern is on the novelty side of the work when comparing to prior works. Particularly, multiple reviewerrs think the technical novelty possessed in the proof seems marginal based on existing tools in average-reward MDPs and the generic method proposed for robust MDPs with non-rectangular ambiguity sets. During the rebuttal, one reviewer also mentioned a few related work. The authors probably should include the detailed discussion to these prior works into the revised version.